# An Adaptive Approach for Infinitely Many-armed Bandits under Generalized Rotting Constraints

**Jung-hun Kim**
Seoul National University
Seoul, South Korea
junghunkim@snu.ac.kr

**Milan Vojnović**
London School of Economics
London, United Kingdom
m.vojnovic@lse.ac.uk

**Se-Young Yun**
KAIST AI
Seoul, South Korea
yunseyoung@kaist.ac.kr

## Abstract

In this study, we consider the infinitely many-armed bandit problems in a rested rotting setting, where the mean reward of an arm may decrease with each pull, while otherwise, it remains unchanged. We explore two scenarios regarding the rotting of rewards: one in which the cumulative amount of rotting is bounded by $V_T$, referred to as the slow-rotting case, and the other in which the cumulative number of rotting instances is bounded by $S_T$, referred to as the abrupt-rotting case. To address the challenge posed by rotting rewards, we introduce an algorithm that utilizes UCB with an adaptive sliding window, designed to manage the bias and variance trade-off arising due to rotting rewards. Our proposed algorithm achieves tight regret bounds for both slow and abrupt rotting scenarios. Lastly, we demonstrate the performance of our algorithm using numerical experiments.

## 1 Introduction

We consider multi-armed bandit problems [15], which are fundamental sequential learning problems where an agent plays an arm at each time and receives a corresponding reward. The core challenge lies in balancing the exploration-exploitation trade-off. Bandit problems have significant implications across diverse real-world applications, such as recommendation systems [17] and clinical trials [23]. In a recommendation system, each arm could represent an item, and the objective is to maximize the click-through rate by making effective recommendations.

In practice, the mean rewards associated with arms may decrease over repeated plays. For instance, in content recommendation systems, the click rates for each item (arm) may diminish due to user boredom with repeated exposure to the same content. Another example is evident in clinical trials, where the efficacy of a medication can decline over time due to drug tolerance induced by repeated administration. The decline in mean rewards resulting from playing arms, referred to as *(rested) rotting bandits*, has been studied by Levine et al. [16], Seznec et al. [20, 21]. The previous work focuses on finite $K$ arms, in which Seznec et al. [20] proposed algorithms achieving $\tilde{O}(\sqrt{KT})$ regret. This suggests that rotting bandits with a finite number of arms are no harder than the stationary case.

However, in real-world scenarios like recommendation systems, where the content items such as movies or articles are numerous, prior methods encounter limitations as the parameter $K$ becomes large, resulting in trivial regret. This emphasizes the necessity of studying rotting scenarios with *infinitely* many arms, particularly when there is a lack of information about the features of each item. The consideration of infinitely many arms for rested rotting bandits fundamentally distinguishes these problems from those with a finite number of arms, as we will explain later.

The study of multi-armed bandit problems with an infinite number of arms has been extensively conducted in the context of *stationary* rewards [6, 24, 8, 10, 5], where the agent has no chance to play all the arms at least once until horizon time $T$. Initially, the distribution of the mean rewards for

38th Conference on Neural Information Processing Systems (NeurIPS 2024).

Table 1: Summary of our regret bounds.

| Type | Regret upper bounds for $\beta \geq 1$ | Regret upper bounds for $0 < \beta < 1$ | Regret lower bounds for $\beta > 0$ |
|---|---|---|---|
| Slow rotting ($V_T$) | Theorem 3.1: $\tilde{O}\left(\max\left\{V_T^{\frac{1}{\beta+2}}T^{\frac{\beta+1}{\beta+2}}, T^{\frac{\beta}{\beta+1}}\right\}\right)$ | Theorem 3.1: $\tilde{O}\left(\max\left\{V_T^{\frac{1}{3}}T^{\frac{2}{3}}, \sqrt{T}\right\}\right)$ | Theorem 4.1: $\Omega\left(\max\left\{V_T^{\frac{1}{\beta+2}}T^{\frac{\beta+1}{\beta+2}}, T^{\frac{\beta}{\beta+1}}\right\}\right)$ |
| Abrupt rotting ($S_T$) | Theorem 3.3: $\tilde{O}\left(\max\left\{S_T^{\frac{1}{\beta+1}}T^{\frac{\beta}{\beta+1}}, \bar{V}_T\right\}\right)$ | Theorem 3.3: $\tilde{O}\left(\max\left\{\sqrt{S_T T}, \bar{V}_T\right\}\right)$ | Theorem 4.2: $\Omega\left(\max\left\{S_T^{\frac{1}{\beta+1}}T^{\frac{\beta}{\beta+1}}, \bar{V}_T\right\}\right)$ |

the arms was assumed to be uniform over the interval $[0, 1]$ [6, 8]. This assumption was expanded to include a much wider range of distributions satisfying $\mathbb{P}(\mu(a) > \mu^* - x) = \Theta(x^\beta)$, for a parameter $\beta > 0$, where $\mu(a)$ represents the mean reward of arm $a$ and $\mu^*$ is the mean reward of the best-performing arm [24, 10, 5]. Additionally, feature information for each arm is not required for multi-armed bandit problems with infinitely many arms, which differs from linear bandits [1] or continuum-armed bandits [3, 14], where feature information for each arm, either for the Lipschitz or linear structure, is involved. While Kim et al. [13], as the closest work, explores the concept of diminishing rewards in the context of bandits with infinitely many arms, their focus is restricted to the case of the maximum rotting rate constraint, where the amount of rotting at each time step is bounded by $\rho$ ($= o(1)$). This naturally directs focus towards regret regarding the maximum rotting rate rather than the total rotting rate over the time horizon. Furthermore, their focus is limited to the case where the initial mean rewards are uniformly distributed ($\beta = 1$).

In our study, we explore rotting bandits with infinitely many arms, subject to generalized initial mean reward distribution with $\beta > 0$ and, importantly, generalized constraints on the rate at which the mean reward of an arm declines. Our investigation into diminishing, or 'rotting,' rewards encompasses two scenarios: one with the total amount of rotting bounded by $V_T$, and the other with the total number of rotting instances bounded by $S_T$. This allows us to capture characteristics of entire rotting rates over the time horizon. Similar constraints of $V_T$ or $S_T$ regarding nonstationarity have been explored in the context of nonstationary finite $K$-armed bandit problems [7, 4, 19], where the reward distribution changes over time independently of the agent. Following established terminology for nonstationary bandits, we denote the environment with a bounded total amount of rotting as the *slow rotting* ($V_T$) case and the one with a bounded total number of rotting instances as the *abrupt rotting* ($S_T$) case.

Here we discuss why (rested) rotting bandits for infinitely many arms are fundamentally different from those for finite arms. In the case of finite arms, rested rotting is known to be no harder than stationary case [20, 21]. This result arises from the confinement of mean rewards of optimal arms and played arms within confidence bounds, even under rested rotting (as demonstrated in Lemma 1 of Seznec et al. [20, 21]). However, in the case of infinite arms under distribution for initial mean reward that allows for an infinite number of near-optimal arms, there always exist near-optimal arms outside of explored arms. Therefore, the mean reward gap may not be confined within confidence bounds. This fundamental difference from finite-armed rotting bandits introduces additional challenges. In our setting of infinite arms, there exists an additional cost for exploring new (unexplored) arms to find near-optimal arms while eliminating explored suboptimal arms. If the total rotting effect on explored arms is significant, then the frequency at which new near-optimal arms must be sought increases substantially, resulting in a large regret. This is why the rested rotting significantly affects the exploration cost regarding $V_T$ or $S_T$ in our setting, which differs from the case of finite arms.

To solve our problem, we introduce algorithms that employ an adaptive sliding window mechanism, effectively managing the tradeoff between bias and variance stemming from rotting rewards. Notably, to the best of our knowledge, this is the first work to consider slow and abrupt rotting scenarios, in the context of infinitely many-armed bandits. Furthermore, it is the first work to consider the generalized initial mean reward distribution for rotting bandits with infinitely many arms.

**Summary of our Contributions.** The key contributions of this study are summarized in the following points. Please refer to Table 1 for a summary of our regret bounds.

• To address the slow and abrupt rotting scenarios, we propose a UCB-based algorithm using an adaptive sliding window and a threshold parameter. This algorithm allows for effectively managing the bias and variance trade-off arising from rotting rewards.

• In the context of slow rotting ($V_T$) or abrupt rotting ($S_T$), for any $\beta > 0$, we present regret upper bounds achieved by our algorithm with an appropriately tuned threshold parameter. It is noteworthy that $V_T$, $S_T$, and $\beta$ are being considered for the first time in the context of rotting bandits with infinitely many arms.

• We establish regret lower bounds for both slow rotting and abrupt rotting scenarios. These regret lower bounds imply the tightness of our upper bounds when $\beta \geq 1$. In the other case, when $0 < \beta < 1$, there is a gap between our upper bounds and the corresponding lower bounds, similar to what can be found in related literature, which is discussed in the paper.

• Lastly, we demonstrate the performance of our algorithm through numerical experiments on synthetic datasets, validating our theoretical results.

## 2 Problem Statement

We consider rotting bandits with infinitely many arms where the mean reward of an arm may decrease when the agent pulls the arm. Let $\mathcal{A}$ be the set of infinitely many arms and let $\mu_t(a)$ denote the unknown mean reward of arm $a \in \mathcal{A}$ at time $t$. At each time $t$, an agent pulls arm $a_t^\pi \in \mathcal{A}$ according to policy $\pi$ and observes stochastic reward $r_t$ given by $r_t = \mu_t(a_t^\pi) + \eta_t$, where $\eta_t$ is a noise term following a 1-sub-Gaussian distribution. To simplify, we use $a_t$ for $a_t^\pi$ when there is no confusion about the policy. We assume that initial mean rewards $\{\mu_1(a)\}_{a \in \mathcal{A}}$ are i.i.d. random variables on $[0, 1]$, a widely accepted assumption in the context of infinitely many-armed bandits [8, 6, 24, 10, 5, 13].

As in Wang et al. [24], Carpentier and Valko [10], Bayati et al. [5], we consider, to our best knowledge, the most general condition on the distribution of the initial mean reward of an arm, satisfying the following condition: there exists a constant $\beta > 0$ such that for every $a \in \mathcal{A}$ and all $x \in [0, 1]$,

$$\mathbb{P}(\mu_1(a) > 1 - x) = \mathbb{P}(\Delta_1(a) < x) = \Theta(x^\beta), \tag{1}$$

where $\Delta_1(a) = 1 - \mu_1(a)$ is the initial sub-optimality gap. As noted in [24, 10, 5], Eq.(1) is a non-trivial condition only when $x$ approaches 0, as for any constant $x \in (0, 1]$, it becomes $\mathbb{P}(\Delta_1(a) < x) = \Theta(1)$, which may accommodate a wide range of distributions. It is noteworthy that the larger the value of $\beta$, the smaller the probability of sampling a good arm. Furthermore, the uniform distribution is a special case when $\beta = 1$. Importantly, our work allows for a wider range of distributions satisfying (1) for any constant $\beta > 0$ than the uniform distribution ($\beta = 1$) considered in Kim et al. [13]. Additional discussion is deferred to Appendix A.2.

The rotting of arms is defined as follows. At each time $t \geq 1$, the mean rewards of arms are updated as

$$\mu_{t+1}(a) = \mu_t(a) - \rho_t(a),$$

where $\rho_t(a_t) \geq 0$ for the pulled arm $a_t$ and $\rho_t(a) = 0$ for every $a \in \mathcal{A}/\{a_t\}$, which implies that the rotting may occur only for the pulled arm at each time. Note that, for every $a \in \mathcal{A}$ and $t \geq 2$, it holds $\mu_t(a) = \mu_1(a) - \sum_{s=1}^{t-1} \rho_s(a)$, allowing $\mu_t(a)$ to take negative values. For notation simplicity, in what follows, we write $\rho_t$ for $\rho_t(a_t)$ when there is no confusion. We refer to $\rho_1, \rho_2, \ldots$ as rotting rates. We also use the notation $[m] := \{1, \ldots, m\}$, for any integer $m \geq 1$.

We consider two cases for rotting rates: (a) *slow rotting case* where, for given $V_T \geq 0$, the cumulative amount of rotting is required to satisfy the slow rotting constraint $\sum_{t=1}^{T-1} \rho_t \leq V_T$, and (b) *abrupt rotting case* where, for given $S_T \in [T]$, the cumulative number of rotting instances (plus one) is required to satisfy the abrupt rotting constraint $1 + \sum_{t=1}^{T-1} \mathbb{1}(\rho_t \neq 0) \leq S_T$. The values of rotting rates of pulled arms, $\{\rho_t\}_{t \in [T-1]}$, are assumed to be determined by an adversary, described as follows.

**Assumption 2.1** (Adaptive Adversary). At each time $t \in [T]$, the value of the rotting rate $\rho_t \geq 0$ is arbitrarily determined immediately after the agent pulls $a_t$, *subject to* the constraint of either slow rotting for a given $V_T$ or abrupt rotting for a given $S_T$.

*Remark* 2.2. The adaptive adversary under the slow rotting constraint ($V_T$) is more general than that in Kim et al. [13], in which the adversary is under *a maximum rotting rate constraint*; that is, for given $\rho = o(1)$, $\rho_t \leq \rho$ for all $t \in [T-1]$. This is because our adversary is under a weaker constraint bounding the total sum of the rotting rates rather than each individual rotting rate. Additionally, the abrupt rotting constraint ($S_T$) is fundamentally different from the maximum rotting constraint [13] because the adversary for abrupt rotting is under a constraint on the total number of rotting instances rather than the magnitude of rotting rates.

Our problem's objective is to find a policy that minimizes the expected cumulative regret over a time horizon of $T$ time steps. For a given policy $\pi$, the regret is defined as $\mathbb{E}[R^\pi(T)] = \mathbb{E}[\sum_{t=1}^{T}(1 - \mu_t(a_t^\pi))]$. The use of 1 in the regret definition for the optimal mean reward is justified because among the infinite arms with initial mean rewards following the distribution specified in (1), there always exists an arm whose mean reward is sufficiently close to 1.[1]

We note that while we have $S_T \leq T$ because the number of rotting instances is at most $T - 1$, the upper bound for $V_T$ may not exist due to the lack of a constraint on the values of $\rho_t$'s. Here we discuss an assumption for the cumulative amount of rotting. In the case of $\sum_{t=1}^{T-1} \rho_t > T$, the problem becomes trivial as shown in the following proposition.

**Proposition 2.3.** *In the case of $\sum_{t=1}^{T-1} \rho_t > T$, there always exists a rotting adversary that incurs regret of $\Omega(T)$ and a simple policy that samples a new arm every round achieves the optimal regret of $\Theta(T)$.*

*Proof.* The proof is provided in Appendix A.3 □

From the above proposition, when $\sum_{t=1}^{T-1} \rho_t > T$, the regret lower bound of this problem is $\Omega(T)$, which can be achieved by a simple policy. Therefore, we consider the following assumption for the region of non-trivial problems.

**Assumption 2.4.** $\sum_{t=1}^{T-1} \rho_t \leq T$.

Notably, from the above assumption, we consider $V_T \leq T$ for the slow rotting case. We also note that the assumption is not strong, as it frequently arises in real-world scenarios and is more general than the assumption made in prior work, as described in the following remarks.

*Remark* 2.5. The assumption of $\sum_{t=1}^{T-1} \rho_t \leq T$ is satisfied if mean rewards are under the constraint of $0 \leq \mu_t(a_t) \leq 1$ for all $t \in [T]$, because this condition implies $\rho_t \leq 1$ for all $t \in [T]$. Such a scenario is frequently encountered in real-world applications, where reward is represented by metrics like click rates or (normalized) ratings in content recommendation systems.

*Remark* 2.6. Our rotting scenario with $\sum_{t=1}^{T-1} \rho_t \leq T$ is more general in scope than the one with the maximum rotting rate constraint where $\rho_t \leq \rho = o(1)$ for all $t \in [T - 1]$, which was explored in Kim et al. [13]. This is because for our setting, $\rho_t$ is not necessarily bounded by $o(1)$, and under the maximum rotting constraint, the condition $\sum_{t=1}^{T-1} \rho_t \leq T$ is always satisfied.

## 3  Algorithms and Regret Analysis

We propose an algorithm (Algorithm 1) utilizing an *adaptive sliding window* for delicately controlling bias and variance tradeoff of the mean reward estimator from rotting rewards, drawing on insights from [4, 21]. This is why our algorithm can adapt to varying rotting rates $\rho_t$ and achieve tight regret bounds with respect to $V_T$ or even $S_T$. Furthermore, our algorithm accommodates the general mean reward distribution with $\beta > 0$ by employing a carefully optimized threshold parameter.

Here we describe our proposed algorithm in detail. We define $\widehat{\mu}_{[t_1,t_2]}(a) = \sum_{t=t_1}^{t_2} r_t \mathbb{1}(a_t = a)/n_{[t_1,t_2]}(a)$ where $n_{[t_1,t_2]}(a) = \sum_{t=t_1}^{t_2} \mathbb{1}(a_t = a)$ for $t_1 \leq t_2$. Then for window-UCB index of the algorithm, we define $WUCB(a, t_1, t_2, T) = \widehat{\mu}_{[t_1,t_2]}(a) + \sqrt{12 \log(T)/n_{[t_1,t_2]}(a)}$. In what follows, 'selecting an arm' means that a policy chooses an arm before pulling it. In Algorithm 1, we first select an arbitrary new arm $a \in \mathcal{A}'$ without prior knowledge regarding the arms in $\mathcal{A}'$, denoting the corresponding time as $t(a)$. We define $\mathcal{T}_t(a)$ as the set of starting times for sliding windows of doubling lengths, defined as $\mathcal{T}_t(a) = \{s \in [T] : t(a) \leq s \leq t - 1 \text{ and } s = t - 2^{i-1} \text{ for some } i \in \mathbb{N}\}$. Then the algorithm pulls the arm consecutively until the following threshold condition is satisfied: $\min_{s \in \mathcal{T}_t(a)} WUCB(a, s, t - 1, T) < 1 - \delta$, in which the sliding window having minimized window-UCB is utilized for adapting nonstationarity. If the threshold condition holds, then the algorithm considers the arm to be a sub-optimal (bad) arm and withdraws the arm. Then it selects a new arm and repeats this procedure.

---

[1]This assertion follows from the fact that for any $\epsilon > 0$, there exists an arm $a$ in $\mathcal{A}$ excluding rotted arms such that $\Delta_1(a) < \epsilon$ with probability 1, as $\lim_{n \to \infty}(1 - \mathbb{P}(\Delta_1(a) \geq \epsilon)^n) = 1$.

**Algorithm 1** UCB-Threshold with Adaptive Sliding Window
---
**Input:** $T, \delta, \mathcal{A}$; **Initialize:** $\mathcal{A}' \leftarrow \mathcal{A}$
Select a new arm $a \in \mathcal{A}'$; Pull arm $a$ and get reward $r_1$
$t(a) \leftarrow 1$
**for** $t = 2, \ldots, T$ **do**
    **if** $\min_{s \in \mathcal{T}_t(a)} WUCB(a, s, t-1, T) < 1 - \delta$ **then**
        $\mathcal{A}' \leftarrow \mathcal{A}'/\{a\}$
        Select a new arm $a \in \mathcal{A}'$; Pull arm $a$ and get reward $r_t$
        $t(a) \leftarrow t$
    **else**
        Pull arm $a$ and get reward $r_t$

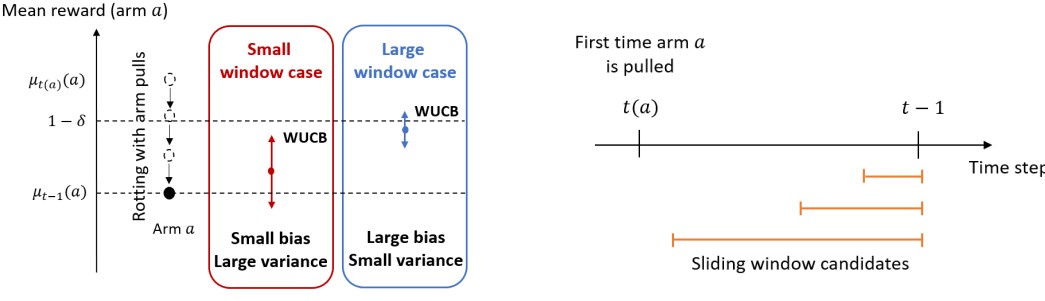

Figure 1: Illustrations for the adaptive sliding window: (left) the effect of the sliding window length on the mean reward estimation, (right) sliding window candidates with doubling lengths.

Utilizing the adaptive sliding window having minimized window UCB index enhances the algorithm's ability to dynamically identify poorly-performing arms across varying rotting rates. This adaptability is achieved by managing the tradeoff between bias and variance. The concept is depicted in Figure 1 (left), where an arm $a$ undergoes multiple rotting events. WUCB with a smaller window exhibits minimal bias with the arm's most recent mean reward but introduces higher variance. Conversely, WUCB with a larger window displays increased bias but reduced variance. In this visual representation, the value of WUCB with a small window reaches a minimum, enabling the algorithm to compare this value with $1 - \delta$ to identify the suboptimal arm. Moreover, as illustrated in Figure 1 (right), by taking into account the constraint of $s = t - 2^{i-1}$ for the size of the adaptive windows, we can reduce the computation time for determining the appropriate window and reduce the required memory from $O(t)$ to $O(\log t)$, respectively, for each time $t$.

Having introduced our algorithm, we compare it with the previously proposed algorithm UCB-TP [13], which is tailored for the maximum rotting rate constraint $\rho_t \leq \rho \ (= o(1))$ for all $t > 0$ and the uniform initial mean reward distribution ($\beta = 1$). The mean reward estimator in UCB-TP considers the worst-case scenario with the maximum rotting rate $\rho$ as $\tilde{\mu}_t^o(a) - \rho n_t(a)$ where $\tilde{\mu}_t^o$ is an estimator for the initial mean reward, $n_t(a)$ is the number of times arm $a$ is pulled until time $t - 1$, and $\rho n_t(a)$ is for reducing the bias from the worst-case rotting, which leads to achieving a regret bound of $\tilde{O}(\max\{\rho^{1/3}T, \sqrt{T}\})$. This estimator is not appropriate for dealing with our generalized rotting constraints because it aims to attain the regret bound regarding the maximum rotting rate $\rho$ without adequately addressing individual $\rho_t$ values. Our algorithm resolves this by using an adaptive sliding window estimator, which can handle rotting rates carefully. Furthermore, it can accommodate any constant $\beta > 0$ by using a carefully optimized $\delta$, as shown below.

**Slow Rotting ($V_T$).** Here we consider the case of slow rotting, where, recall, the adaptive adversary is constrained such that the total amount of rotting is bounded by $V_T$. We analyze the regret of Algorithm 1 with tuned $\delta$ using $\beta$ and $V_T$. We define $\delta_V(\beta) = c_1 \max\{(V_T/T)^{1/(\beta+2)}, 1/T^{1/(\beta+1)}\}$ when $\beta \geq 1$ and $\delta_V(\beta) = c_1 \max\{(V_T/T)^{1/3}, 1/\sqrt{T}\}$ when $0 < \beta < 1$ for some constant $0 < c_1 < 1$. The algorithm with $\delta_V(\beta)$ achieves a regret bound in the following theorem.

**Theorem 3.1.** *The policy $\pi$ of Algorithm 1 with $\delta = \delta_V(\beta)$ achieves:*

$$\mathbb{E}[R^\pi(T)] = \begin{cases} \tilde{O}(\max\{V_T^{\frac{1}{\beta+2}} T^{\frac{\beta+1}{\beta+2}}, T^{\frac{\beta}{\beta+1}}\}) & for \ \beta \geq 1, \\ \tilde{O}(\max\{V_T^{\frac{1}{3}} T^{\frac{2}{3}}, \sqrt{T}\}) & for \ 0 < \beta < 1. \end{cases}$$

We observe that when $\beta$ increases above 1, the regret bound becomes worse because the likelihood of sampling a good arm decreases. However, when $\beta$ decreases below 1, the regret bound remains the same due to the inability to avoid a certain level of regret arising from estimating the mean reward. Further discussion will be provided later. Also, we observe that when $V_T = O(\max\{1/T^{1/(\beta+1)}, 1/\sqrt{T}\})$ where the problem becomes near-stationary, the regret bound in Theorem 3.1 matches the previously known regret bound for stationary infinitely many-armed bandits, $\tilde{O}(\max\{T^{\beta/(\beta+1)}, \sqrt{T}\})$, as shown in Wang et al. [24], Bayati et al. [5].

*Proof sketch.* The full proof is provided in Appendix A.4. Here we outline the main ideas of the proof. There are several technical challenges involved in regret analysis, such as dealing with varying $\rho_t$ individually with respect to the total rotting budget of $V_T$, adaptive estimation in our algorithm, and the generalized distributions of initial mean rewards of arms with parameter $\beta > 0$, none of which appear in Kim et al. [13].

We separate the regret into two components: one associated with pulling initially good arms and another with pulling initially bad arms. An arm $a$ is said to be good if $\mu_1(a) \geq 1 - 2\delta$ and, otherwise, it is said to be bad. The reason why the separation is required is that our adaptive algorithm has different behaviors depending on the category of arms. Good arms may be pulled repeatedly when rotting rates are sufficiently small but bad arms are not. We write $R^\pi(T) = R^{\mathcal{G}}(T) + R^{\mathcal{B}}(T)$, where $R^{\mathcal{G}}(T)$ is the regret from good arms and $R^{\mathcal{B}}(T)$ is the regret from bad arms.

We first provide a bound for $\mathbb{E}[R^{\mathcal{G}}(T)]$. For analyzing regret from good arms, we analyze the cumulative amount of rotting while pulling a selected good arm before withdrawing the arm by the algorithm. Let $\mathcal{A}_T^{\mathcal{G}}$ be a set of distinct good arms selected until $T$, $t_1(a)$ be the initial time step at which arm $a$ is pulled, and $t_2(a)$ be the final time step at which the arm is pulled by the algorithm so that the threshold condition holds when $t = t_2(a) + 1$. For simplicity, we use $t_1$ and $t_2$ for $t_1(a)$ and $t_2(a)$, when there is no confusion. For any time steps $n \leq m$, we define $V_{[n,m]}(a) = \sum_{t=n}^{m} \rho_t(a)$ and $\overline{\rho}_{[n,m]}(a) = V_{[n,m]}(a)/n_{[n,m]}(a)$. We show that the regret is decomposed as

$$R^{\mathcal{G}}(T) = \sum_{a \in \mathcal{A}_T^{\mathcal{G}}} \left( \Delta_1(a) n_{[t_1, t_2]}(a) + \sum_{t=t_1+1}^{t_2} V_{[t_1, t-1]}(a) \right), \tag{2}$$

which consists of regret from the initial mean reward and the cumulative amount of rotting for each arm. For the first term of $\sum_{a \in \mathcal{A}_T^{\mathcal{G}}} \Delta_1(a) n_{[t_1, t_2]}(a)$ in (2), since $\Delta_1(a) = O(\delta)$ from the definition of good arms $a \in \mathcal{A}_T^{\mathcal{G}}$, we have $\mathbb{E}[\sum_{a \in \mathcal{A}_T^{\mathcal{G}}} \Delta_1(a) n_{[t_1, t_2]}(a)] = O(\delta T)$.

The main difficulty in (2) lies in dealing with the second term, $\sum_{a \in \mathcal{A}_T^{\mathcal{G}}} \sum_{t=t_1+1}^{t_2} V_{[t_1, t-1]}(a)$, where we need to analyze the amount of cumulative rotting until the arm is eliminated by using the adaptive threshold condition. A careful analysis of the adaptive threshold policy is required to limit the total variation of rotting. By examining the estimation errors arising from variance and bias due to the adaptive threshold condition, we can establish an upper bound for the cumulative amount of rotting as

$$\sum_{a \in \mathcal{A}_T^{\mathcal{G}}} \sum_{t=t_1+1}^{t_2} V_{[t_1, t-1]}(a) = \tilde{O}\left( T\delta + V_T + \sum_{a \in \mathcal{A}_T^{\mathcal{G}}} V_{[t_1, t_2-2]}(a)^{\frac{1}{3}} n_{[t_1, t_2-2]}(a)^{\frac{2}{3}} \right). \tag{3}$$

Therefore, from $\delta = \delta_V(\beta)$, $V_T \leq T$, and Eqs. (2) and (3), using Hölder's inequality, we have

$$\mathbb{E}[R^{\mathcal{G}}(T)] = \begin{cases} \tilde{O}(\max\{V_T^{\frac{1}{\beta+2}} T^{\frac{\beta+1}{\beta+2}}, T^{\frac{\beta}{\beta+1}}\}) & for \ \beta \geq 1, \\ \tilde{O}(\max\{V_T^{\frac{1}{3}} T^{\frac{2}{3}}, \sqrt{T}\}) & for \ 0 < \beta < 1. \end{cases} \tag{4}$$

Next, we provide a bound for $\mathbb{E}[R^{\mathcal{B}}(T)]$. We employ episodic regret analysis, defining an episode as the time steps between consecutively selected distinct good arms by the algorithm. By analyzing

bad arms within each episode, we can derive an upper bound for the overall regret arising from bad arms. We define the regret from bad arms over $m^{\mathcal{G}}$ episodes as $R^{\mathcal{B}}_{m^{\mathcal{G}}}$. We first consider the case of $V_T > \max\{1/\sqrt{T}, 1/T^{1/(\beta+1)}\}$. In this case, by setting $m^{\mathcal{G}} = \lceil 2V_T/\delta \rceil$, we can show that $R^{\mathcal{B}}(T) \leq R^{\mathcal{B}}_{m^{\mathcal{G}}}$ with a high probability. By analyzing $R^{\mathcal{B}}_{m^{\mathcal{G}}}$ with the episodic analysis, we can show that $\mathbb{E}[R^{\mathcal{B}}(T)] \leq \mathbb{E}[R^{\mathcal{B}}_{m^{\mathcal{G}}}] = \tilde{O}(\max\{T^{\frac{\beta+1}{\beta+2}}V_T^{\frac{1}{\beta+2}}, T^{\frac{2}{3}}V_T^{\frac{1}{3}}\})$. As in the similar manner, when $V_T \leq \max\{1/\sqrt{T}, 1/T^{1/(\beta+1)}\}$, by setting $m^{\mathcal{G}} = C_3$ for some constant $C_3 > 0$, we can show that $\mathbb{E}[R^{\mathcal{B}}(T)] \leq \mathbb{E}[R^{\mathcal{B}}_{m^{\mathcal{G}}}] = \tilde{O}(\max\{T^{\frac{\beta}{\beta+1}}, \sqrt{T}\})$. From the above two inequalities, we have

$$\mathbb{E}[R^{\mathcal{B}}(T)] = \begin{cases} \tilde{O}(\max\{V_T^{\frac{1}{\beta+2}}T^{\frac{\beta+1}{\beta+2}}, T^{\frac{\beta}{\beta+1}}\}) & for \ \beta \geq 1, \\ \tilde{O}(\max\{V_T^{\frac{1}{3}}T^{\frac{2}{3}}, \sqrt{T}\}) & for \ 0 < \beta < 1. \end{cases} \tag{5}$$

Finally, from (4) and (5), we can conclude the proof from $\mathbb{E}[R^{\pi}(T)] = \mathbb{E}[R^{\mathcal{G}}(T)] + \mathbb{E}[R^{\mathcal{B}}(T)]$. □

*Remark* 3.2. We compare our result in Theorem 3.1 with that in Kim et al. [13], which, recall, is under the maximum rotting rate constraint $\rho_t \leq \rho = o(1)$ for all $t$ and uniform distribution of initial mean rewards ($\beta = 1$). For a fair comparison, we consider an oblivious adversary for rotting rates where the values of $\rho_t$'s are determined before an algorithm is run, which may imply $V_T = \sum_{t=1}^{T-1} \rho_t$ and $\rho = \max_{t \in [T-1]} \rho_t$. Then with $\beta = 1$, from $V_T \leq T\rho$, we can observe that the regret bound of Algorithm 1 is tighter than that of UCB-TP [13] as $\tilde{O}(\max\{V_T^{\frac{1}{3}}T^{\frac{2}{3}}, \sqrt{T}\}) \leq \tilde{O}(\max\{\rho^{\frac{1}{3}}T, \sqrt{T}\})$, where the latter is the regret bound of UCB-TP. We will demonstrate this in our numerical results.

**Abrupt Rotting** ($S_T$). Here we consider abruptly rotting reward distribution under the constraint of $S_T$. We consider Algorithm 1 with $\delta$ newly tuned by $S_T$ and $\beta$. We define $\delta_S(\beta) = c_1(S_T/T)^{1/(\beta+1)}$ when $\beta \geq 1$ and $\delta_S(\beta) = c_1(S_T/T)^{1/2}$ when $0 < \beta \leq 1$ for some constant $0 < c_1 < 1$. We also define $\bar{V}_T = \sum_{t=1}^{T-1} \mathbb{E}[\rho_t]$. In the following theorem, we present a regret upper bound for Algorithm 1 with $\delta_S(\beta)$.

**Theorem 3.3.** *The policy $\pi$ of Algorithm 1 with $\delta = \delta_S(\beta)$ achieves:*

$$\mathbb{E}[R^{\pi}(T)] = \begin{cases} \tilde{O}(\max\{S_T^{\frac{1}{\beta+1}}T^{\frac{\beta}{\beta+1}}, \bar{V}_T\}) & for \ \beta \geq 1, \\ \tilde{O}(\max\{\sqrt{S_TT}, \bar{V}_T\}) & for \ 0 < \beta < 1. \end{cases}$$

As in the slow rotting case, for the abrupt rotting case ($S_T$), we observe that when $\beta$ increases above 1, the regret bound in the above theorem worsens as the likelihood of sampling a good arm decreases. When $\beta$ decreases below 1, the regret bound remains the same because we cannot avoid a certain level of regret arising from estimating the mean reward of an arm. Additionally, we observe that the regret bound is linearly bounded by $\bar{V}_T$, which is attributed to the algorithm's necessity to pull a rotted arm at least once to determine its status as bad. Later, in the analysis of regret lower bounds, we will establish the impossibility of avoiding $\bar{V}_T$ regret in the worst-case. Notably, in the typical cases where $0 \leq \rho_t \leq 1$ for all $t > 0$, as discussed in Remark 2.5, $\bar{V}_T$ is negligible in the regret bound from $\bar{V}_T \leq S_T \leq T$. Furthermore, we observe that for the case of $S_T = 1$, where the problem becomes stationary (implying $\bar{V}_T = 0$), the regret bound matches the previously known regret bound of $\tilde{O}(\max\{T^{\beta/(\beta+1)}, \sqrt{T}\})$ for the stationary infinitely many-armed bandits [24, 5].

*Proof sketch.* The full proof is provided in Appendix A.5. Here we provide a proof outline. We follow the proof framework of Theorem 3.1 but the main difference lies in carefully dealing with substantially rotted arms. For the ease of presentation, we consider each arm that experiences abrupt rotting as if it were newly selected by the algorithm, treating the arm before and after abrupt rotting as distinct arms. The definition of a good arm and a bad arm is based on the mean reward at the time when it is newly selected. Then we divide the regret into regret from good and bad arms as $R^{\pi}(T) = R^{\mathcal{G}}(T) + R^{\mathcal{B}}(T)$. From the definition of good arms, we can easily show that

$$\mathbb{E}[R^{\mathcal{G}}(T)] = O(\delta_S(\beta)T) = \begin{cases} \tilde{O}(S_T^{\frac{1}{\beta+1}}T^{\frac{\beta}{\beta+1}}) & for \ \beta \geq 1, \\ \tilde{O}(\sqrt{S_TT}) & for \ 0 < \beta < 1. \end{cases}$$

For dealing with $R^{\mathcal{B}}(T)$, we partition the regret into two scenarios: one where the bad arm is initially bad sampled from the distribution of (1) and another where it becomes bad after rotting. This can be

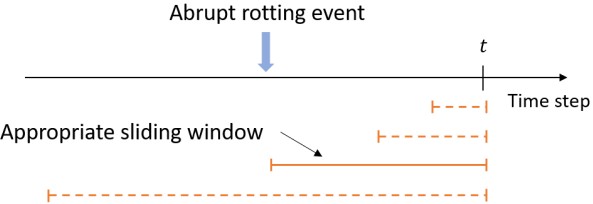

Figure 2: Adaptive sliding window for abrupt rotting.

expressed as $R^{\mathcal{B}}(T) = R^{\mathcal{B},1}(T) + R^{\mathcal{B},2}(T)$. Then for the former regret, $R^{\mathcal{B},1}(T)$, as in the proof of Theorem 3.1, by using the episodic analysis with $m^{\mathcal{G}} = S_T$, we can show that

$$\mathbb{E}[R^{\mathcal{B},1}(T)] \leq \mathbb{E}[R_{m^{\mathcal{G}}}^{\mathcal{B}}] = \begin{cases} \tilde{O}(S_T^{\frac{1}{\beta+1}} T^{\frac{\beta}{\beta+1}}) & for \ \beta \geq 1, \\ \tilde{O}(\sqrt{S_T T}) & for \ 0 < \beta < 1. \end{cases}$$

For the regret from rotted bad arms, $R^{\mathcal{B},2}(T)$, it is critical to analyze significant rotting instances to obtain a tight bound with respect to $S_T$, a factor not addressed in the regret analysis of slow rotting ($V_T$) in Theorem 3.1. We analyze that when there exists significant rotting, then the algorithm can efficiently detect it as a bad arm and eliminate it by pulling it at once. From this analysis, we have

$$\mathbb{E}[R^{\mathcal{B},2}(T)] = \begin{cases} \tilde{O}(\max\{S_T^{\frac{\beta}{\beta+1}} T^{\frac{1}{\beta+1}}, \bar{V}_T\}) & for \ \beta \geq 1, \\ \tilde{O}(\max\{\sqrt{S_T T}, \bar{V}_T\}) & for \ 0 < \beta < 1. \end{cases}$$

Putting all the results together with $\mathbb{E}[R^{\pi}(T)] = \mathbb{E}[R^{\mathcal{G}}(T)] + \mathbb{E}[R^{\mathcal{B},1}(T)] + \mathbb{E}[R^{\mathcal{B},2}(T)]$ and $S_T \leq T$, we can conclude the proof. □

Remarkably, our proposed method, utilizing an adaptive sliding window, yields a tight bound (lower bounds will be presented later) not only for slow rotting but also for abrupt rotting ($S_T$) scenarios characterized by a limited number of rotting instances. The rationale behind the effectiveness of the adaptive sliding window in controlling the bias and variance tradeoff with respect to abrupt rotting is as follows. It can be observed that the adaptive threshold condition of $\min_{s \in \mathcal{T}_t(a)} WUCB(a, s, t - 1, T) < 1 - \delta$ is equivalent to the condition of $WUCB(a, s, t - 1, T) < 1 - \delta$ for some $s$ such that $t_1(a) \leq s \leq t - 1$ (ignoring the computational reduction trick). The latter expression represents the threshold condition tested for every time step before $t$, encompassing the time step immediately following an abrupt rotting event. Consequently, as illustrated in Figure 2, this adaptive threshold condition can identify substantially rotted arms by mitigating bias and variance using the window starting from the time step following the occurrence of rotting.

**Slow rotting ($V_T$) and abrupt rotting ($S_T$).** In what follows, we study the case of rotting under both slow rotting and abrupt rotting constraints. In this case, Algorithm 1, with $\delta = \min\{\delta_V(\beta), \delta_S(\beta)\}$, can achieve a tighter regret bound as noted in the following corollary, which can be obtained from Theorems 3.1 and 3.3.

**Corollary 3.4.** *Let $R_V$ and $R_S$ be defined as*

$$R_V := \begin{cases} \max\{V_T^{\frac{1}{\beta+2}} T^{\frac{\beta+1}{\beta+2}}, T^{\frac{\beta}{\beta+1}}\} & for \ \beta \geq 1, \\ \max\{V_T^{1/3} T^{2/3}, \sqrt{T}\} & for \ 0 < \beta < 1 \end{cases} \quad and \ R_S := \begin{cases} \max\{S_T^{\frac{1}{\beta+1}} T^{\frac{\beta}{\beta+1}}, V_T\} & for \ \beta \geq 1, \\ \max\{\sqrt{S_T T}, V_T\} & for \ 0 < \beta < 1. \end{cases}$$

*The policy $\pi$ of Algorithm 1 with $\delta = \min\{\delta_V(\beta), \delta_S(\beta)\}$ achieves the regret bound of $\mathbb{E}[R^{\pi}(T)] = \tilde{O}\left(\min\{R_V, R_S\}\right)$.*

**Case without Prior Knowledge of $V_T$, $S_T$, and $\beta$.** Here we study the case when the algorithm does not have prior information about the values of $V_T$, $S_T$, and $\beta$ under the constraints of $V_T$ and $S_T$. These parameters play a crucial role in determining the optimal threshold parameter $\delta$ in Algorithm 1. We propose an algorithm based on estimating the optimal threshold parameter $\delta$ directly (Algorithm 2), rather than estimating each unknown parameter separately, employing the Bandit-over-Bandit (BoB) approach [11]. Under assumptions concerning the bounds for the cumulative amount of

rotting and a constrained version of the adaptive adversary for rotting rates, which are less general than Assumptions 2.1 and 2.4 but still more general than those in Kim et al. [13], the algorithm achieves a regret bound of $\mathbb{E}[R^\pi(T)] = \tilde{O}(\min\{R_V, R_S\} + \max\{T^{(2\beta+1)/(2\beta+2)}, T^{3/4}\})$. The additional cost arises from learning $\delta$ compared to the regret bound of Corollary 3.4. Further details of the algorithm and regret analysis are provided in Appendix A.6.

## 4 Regret Lower Bounds

In this section, we present regret lower bounds for our problem under Assumptions 2.1 and 2.4 to provide guidance on the tightness of our regret upper bounds. For the regret lower bounds, we consider worst-case instances of rotting rates. In the following theorems, we provide regret lower bounds for slow rotting ($V_T$) and abrupt rotting ($S_T$), respectively.

**Theorem 4.1.** *For the slow rotting case with the constraint $V_T$ and $\beta > 0$, for any policy $\pi$, there always exists a rotting rate adversary such that the regret of $\pi$ satisfies*

$$\mathbb{E}[R^\pi(T)] = \Omega\left(\max\left\{V_T^{\frac{1}{\beta+2}} T^{\frac{\beta+1}{\beta+2}}, T^{\frac{\beta}{\beta+1}}\right\}\right).$$

*Proof.* The proof is provided in Appendix A.8. $\qquad\square$

**Theorem 4.2.** *For the abrupt rotting case with the constraint $S_T$ and $\beta > 0$, for any policy $\pi$, there always exists a rotting rate adversary such that the regret of $\pi$ satisfies*

$$\mathbb{E}[R^\pi(T)] = \Omega\left(\max\left\{S_T^{\frac{1}{\beta+1}} T^{\frac{\beta}{\beta+1}}, \bar{V}_T\right\}\right).$$

*Proof.* The proof is provided in Appendix A.9. $\qquad\square$

For the abrupt rotting ($S_T$) case, it is unavoidable to incur a $\Omega(\bar{V}_T)$ regret because an arm may only be rotted once and any algorithm pulls this rotted arm at least once in the worst case. From Table 1, we can observe that Algorithm 1 achieves near-optimal regret when $\beta \geq 1$. The optimality proven only for $\beta \geq 1$ has also been observed for stationary infinitely many-armed bandits [5, 24]. We believe that our regret upper bounds are near-optimal across the entire range of $\beta$. Achieving tighter regret lower bounds when $\beta < 1$ is left for future research; see Appendix A.1 for further discussion.

## 5 Experiments

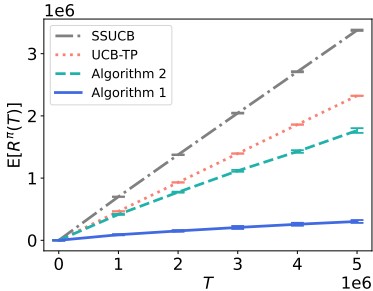

Figure 3: Regret Performance comparison between our algorithms and benchmarks.

In this section, we present numerical results validating some claims of our theoretical analysis.[2] We use randomly generated datasets under a uniform distribution for initial mean rewards ($\beta = 1$).

We first compare the performance of our Algorithms 1 and 2 with UCB-TP [13], the state-of-the-art algorithm for the rotting setting, and SSUCB [5], a near-optimal algorithm for stationary infinitely

---

[2]The source code is available at https://github.com/junghunkim7786/An-Adaptive-Approach-for-Infinitely-Many-armed-Bandits-under-Generalized-Rotting-Constraints

many-armed bandits. For comparison with `UCB-TP`, recall our discussion in Remark 3.2. We set the rotting rates such that $\rho_t = 1/(t \log(T))$ for all $t$, for which $\rho = \rho_1 = 1/\log(T) = o(1)$, $V_T = O(1)$, and $S_T = T$. In Figure 3, we can observe that Algorithms 1 and 2 perform better than `UCB-TP` and `SSUCB` (and Algorithm 1 outperforms Algorithm 2), which is in agreement with our theoretical analysis for the case $\beta = 1$. In this case, the regret bounds for Algorithms 1 and 2 are $\tilde{O}(T^{2/3})$ and $\tilde{O}(T^{3/4})$ from Corollary 3.4 and Theorem A.15, respectively, which are tighter than the regret bound of $\tilde{O}(T/\log(T)^{1/3})$ for `UCB-TP`. Additional experiments can be found in Appendix A.10.

## 6 Conclusion

We explore the challenges of infinitely many-armed bandit problems with rotting rewards, focusing on slow rotting ($V_T$) and abrupt rotting ($S_T$) scenarios. To address these challenges, we propose an algorithm incorporating an adaptive sliding window, which achieves tight regret bounds for both cases. We also provide regret lower bounds for both slow rotting and abrupt rotting cases. Lastly, we demonstrate our algorithm using synthetic datasets.

## 7 Acknowledgements

The authors thank Joe Suk and the anonymous reviewers for helpful discussions. JK was supported by the Global-LAMP Program of the National Research Foundation of Korea (NRF) grant funded by the Ministry of Education (No. RS-2023-00301976). SY was supported by Institute of Information & communications Technology Planning & Evaluation (IITP) grant funded by the Korea government(MSIT) (No. RS-2022-II220311, Development of Goal-Oriented Reinforcement Learning Techniques for Contact-Rich Robotic Manipulation of Everyday Objects)

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

# A  Appendix

## A.1  Limitations & Discussion

As we summarize our results in Table 1, Algorithm 1 achieves near-optimal regret only when $\beta \geq 1$. Here, we discuss the discrepancies between lower and upper bounds when $0 < \beta < 1$. From (1), we can observe that as $\beta$ decreases below 1, the probability to sample good arms may increase, which appears to be beneficial with respect to regret. However, the regret upper bounds for $0 < \beta < 1$ in Theorems 3.1 and 3.3 remain the same as the case when $\beta = 1$ while the regret lower bounds in Theorems 4.1 and 4.2 decrease as $\beta$ decreases, resulting in a gap between the regret upper and lower bounds. The phenomenon that the regret upper bound remains the same when $\beta$ decreases has also been observed in previous literature on infinitely many-armed bandits [5, 24, 10]. As mentioned in Carpentier and Valko [10], although there are likely to be many good arms when $\beta$ is small, it is not possible to avoid a certain amount of regret from estimating mean rewards to distinguish arms under sub-Gaussian reward noise. Therefore, we believe that our regret upper bounds are near-optimal across the entire range of $\beta$, and achieving tighter regret lower bounds when $\beta < 1$ is left for future research. Notably, the optimality proven only for $\beta \geq 1$ has also been observed for stationary infinitely many-armed bandits [5, 24].

## A.2  Additional Explanations for Eq. (1)

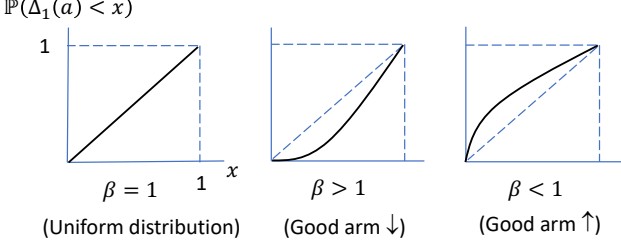

Figure 4: $\mathbb{P}(\Delta_1(a) < x) = x^\beta$ for different values of $\beta$.

To discuss the effect of $\beta$ on the distribution of $\Delta_1(a)$ and the probability of sampling a good arm (having small $\Delta_1(a)$), we consider the case when $\mathbb{P}(\Delta_1(a) < x) = x^\beta$, which is shown in Figure 4 for some values of $\beta$. It is noteworthy that the uniform distribution is a special case when $\beta = 1$. Importantly, the larger the value of $\beta$, the smaller the probability of sampling a good arm.

## A.3  Proof of Proposition 2.3

Recall $\Delta_1(a) = 1 - \mu_1(a)$. We first show that $\mathbb{E}[\mu_1(a)] = \Theta(1)$. For any randomly sampled $a \in \mathcal{A}$, we have $\mathbb{E}[\mu_1(a)] \geq y\mathbb{P}(\mu_1(a) \geq y) = y\mathbb{P}(\Delta_1(a) < 1 - y)$ for $y \in [0, 1]$. With $y = 1/2$, we have $\mathbb{E}[\mu_1(a)] \geq (1/2)\mathbb{P}(\Delta_1(a) < (1/2)) = \Theta(1)$ from constant $\beta > 0$ and (1). Then with $\mathbb{E}[\mu_1(a)] \leq 1$, we can conclude $\mathbb{E}[\mu_1(a)] = \Theta(1)$ (Especially when $\mathbb{P}(\Delta(a) < x) = x^\beta$, we have $\mathbb{E}[\Delta_1(a)] = \int_0^1 \mathbb{P}(\Delta_1(a) \geq x)dx = 1 - \int_0^1 \mathbb{P}(\Delta_1(a) < x)dx = 1 - \int_0^1 x^\beta dx = 1 - \frac{1}{\beta+1}$, which implies $\mathbb{E}[\mu_1(a)] = \Theta(1)$ with constant $\beta > 0$). We then think of a policy $\pi'$ that randomly samples a new arm and pulls it only once every round. Since $\mathbb{E}[\mu_1(a)] = \Theta(1)$ for any randomly sampled $a$, we have $\mathbb{E}[R^{\pi'}(T)] = \Theta(T)$.

Next we show that the policy $\pi'$ is optimal for the worst case of $\sum_{t=1}^{T-1} \rho_t > T$. We think of any policy $\pi''$ except $\pi'$. For any policy $\pi''$, there always exists an arm $a$ such that the policy must pull arm $a$ at least twice. Let $t'$ and $t''$ be the rounds when the policy pulls arm $a$. If we consider $\rho_{t'} > 0$ and $\rho_t = 0$ for $t \in [T-1]/\{t'\}$ such that $\rho_{t'} = \sum_{t=1}^{T-1} \rho_t$ then such policy has $\Omega(\sum_{t=1}^{T-1} \rho_t)$ regret bound. Since $\sum_{t=1}^{T-1} \rho_t > T$, for any algorithm $\pi''$ except $\pi'$, there always exist a rotting rate adversary such that $\mathbb{E}[R^{\pi''}(T)] = \Omega(\sum_{t=1}^{T-1} \mathbb{E}[\rho_t]) = \Omega(T)$. Therefore we can conclude that $\pi'$ is the optimal algorithm for achieving the optimal regret of $\Theta(T)$.

## A.4 Proof of Theorem 3.1: Regret Upper Bound of Algorithm 1 for Slow Rotting with $V_T$

Let $\Delta_t(a) = 1 - \mu_t(a)$. Using a threshold parameter $\delta$, we classify an arm $a$ as *good* if $\Delta_1(a) \leq \delta/2$, *near-good* if $\delta/2 < \Delta_1(a) \leq 2\delta$, and otherwise, we classify $a$ as a *bad* arm. In $\mathcal{A}$, let $\bar{a}_1, \bar{a}_2, \ldots$, be a sequence of arms, which have i.i.d. mean rewards with uniform distribution on $[0, 1]$. Without loss of generality, we assume that the policy samples arms, which are pulled at least once, according to the sequence of $\bar{a}_1, \bar{a}_2, \ldots,$. Let $\mathcal{A}_T$ be the set of sampled arms over the horizon of $T$ time steps, which satisfies $|\mathcal{A}_T| \leq T$. Let $\mathcal{A}_T^{\mathcal{G}}$ be a set of good or near good arms in $\mathcal{A}_T$. WLOG, the following proofs proceed under the given $\mathcal{A}_T$, since the proofs hold for any $\mathcal{A}_T$.

Let $\overline{\mu}_{[s_1,s_2]}(a) = \sum_{t=s_1}^{s_2} \mu_t(a)/n_{[s_1,s_2]}(a)$ for the time steps $0 < s_1 \leq s_2$. We define event $E_1 = \{|\widehat{\mu}_{[s_1,s_2]}(a) - \overline{\mu}_{[s_1,s_2]}(a)| \leq \sqrt{12\log(T)/n_{[s_1,s_2]}(a)}$ for all $1 \leq s_1 \leq s_2 \leq T, a \in \mathcal{A}_T\}$. By following the proof of Lemma 35 in Dylan J. Foster [12], from Lemma A.30 we have

$$P\left(\left|\widehat{\mu}_{[s_1,s_2]}(a) - \overline{\mu}_{[s_1,s_2]}(a)\right| \leq \sqrt{\frac{12\log T}{n_{[s_1,s_2]}(a)}}\right)$$

$$\leq \sum_{n=1}^{T} P\left(\left|\frac{1}{n}\sum_{i=1}^{n} X_i\right| \leq \sqrt{12\log(T)/n}\right)$$

$$\leq \frac{2}{T^5}, \tag{6}$$

where $X_i = r_{\tau_i} - \mu_{\tau_i}(a)$ and $\tau_i$ is the $i$-th time that the policy pulls arm $a$ starting from $s_1$. We note that even though $X_i$'s seem to depend on each other from $\tau_i$'s, each value of $X_i$ is independent of each other. Then using union bound for $s_1$, $s_2$, and $a \in \mathcal{A}_T$, we have $\mathbb{P}(E_1^c) \leq 2/T^2$. From the cumulative amount of rotting $V_T$, we note that $\Delta_t(a) = O(V_T + 1)$ for any $a$ and $t$, which implies $\mathbb{E}[R^\pi(T)|E_1^c] = O(T^2)$ from $V_T \leq T$. For the case where $E_1$ does not hold, the regret is $\mathbb{E}[R^\pi(T)|E_1^c]\mathbb{P}(E_1^c) = O(1)$, which is negligible compared to the regret when $E_1$ holds, which we show later. Therefore, for the rest of the proof, we assume that $E_1$ holds.

For regret analysis, we divide $R^\pi(T)$ into two parts, $R^{\mathcal{G}}(T)$ and $R^{\mathcal{B}}(T)$ corresponding to regret of good or near-good arms, and bad arms over time $T$, respectively, such that $R^\pi(T) = R^{\mathcal{G}}(T) + R^{\mathcal{B}}(T)$. We first provide a bound of $R^{\mathcal{G}}(T)$ in the following lemma.

**Lemma A.1.** *Under $E_1$ and policy $\pi$, we have*

$$\mathbb{E}[R^{\mathcal{G}}(T)] = \tilde{O}\left(T\delta + T^{2/3}V_T^{1/3}\right).$$

*Proof.* Here we consider arms $a \in \mathcal{A}_T^{\mathcal{G}}$. Let $V_{[n,m]}(a) = \sum_{l=n}^{m} \rho_l(a)$ and $\overline{\rho}_{[n,m]}(a) = \sum_{l=n}^{m} \rho_l(a)/n_{[n,m]}(a)$ for time steps $n \leq m$. For ease of presentation, for time steps $r > q$, we define $V_{[r,q]}(a) = n_{[r,q]}(a) = \overline{\rho}_{[r,q]}(a) = \sum_{t=r}^{q} x(t) = 0$ for $x(t) \in \mathbb{R}$ and $1/0 = \infty$. Then, for any $s$ such that $n \leq s \leq m$, under $E_1$ we have

$$\widehat{\mu}_{[s,m]}(a) \leq \overline{\mu}_{[s,m]}(a) + \sqrt{12\log(T)/n_{[s,m]}(a)}$$

$$\leq \mu_m(a) + \sum_{l=s}^{m-1}\rho_l \mathbb{1}(a_l = a) + \sqrt{12\log(T)/n_{[s,m]}(a)}$$

$$= \mu_n(a) - \sum_{l=n}^{m-1}\rho_l \mathbb{1}(a_l = a) + \sum_{l=s}^{m-1}\rho_l \mathbb{1}(a_l = a) + \sqrt{12\log(T)/n_{[s,m]}(a)}$$

$$\leq \mu_n(a) - V_{[n,m-1]}(a) + \overline{\rho}_{[s,m-1]}(a)n_{[s,m]}(a) + \sqrt{12\log(T)/n_{[s,m]}(a)}.$$

Therefore, from $\mu_n(a) \leq 1$ we obtain

$$\widehat{\mu}_{[s,m]}(a) + \sqrt{12\log(T)/n_{[s,m]}(a)}$$

$$\leq 1 - V_{[n,m-1]}(a) + \overline{\rho}_{[s,m-1]}(a)n_{[s,m]}(a) + 2\sqrt{12\log(T)/n_{[s,m]}(a)}. \tag{7}$$

Let $t_1(a)$ be the initial time when the arm $a$ is sampled and pulled and $t_2(a)$ be the final time when the policy pulls the arm. For simplicity, we use $t_1$ and $t_2$ instead of $t_1(a)$ and $t_2(a)$, respectively, when there is no confusion. We define $\mathcal{A}^0$ as a set of arms $a \in \mathcal{A}_T^{\mathcal{G}}$ such that $t_2(a) = t_1(a)$ and define $\mathcal{A}^1$ as a set of arms $a \in \mathcal{A}_T^{\mathcal{G}}$ such that $t_2(a) = t_1(a) + 1$. We also define a set of arms $\overline{\mathcal{A}}_T^{\mathcal{G}} = \{a \in \mathcal{A}_T^{\mathcal{G}}/\{\mathcal{A}^0 \cup \mathcal{A}^1\} : n_{[t_1,t_2-1]}(a) > \lceil(\log T)^{1/3}/\overline{\rho}_{[t_1,t_2-2]}(a)^{2/3}\rceil\}$. Let $w(a) = \lceil(\log T)^{1/3}/\overline{\rho}_{[t_1,t_2-2]}(a)^{2/3}\rceil$. For simplicity, we use $w$ for $w(a)$ when there is no confusion. Then with the fact that $\mu_t(a) = \mu_{t_1}(a) - \sum_{t=t_1(a)}^{t-1} \rho_t(a) = \mu_{t_1}(a) - V_{[t_1,t-1]}(a)$ for $t_1(a) \le t \le t_2(a)$, we have

$$
\begin{aligned}
\mathbb{E}[R^{\mathcal{G}}(T)] &= \mathbb{E}\left[\sum_{a \in \mathcal{A}_T^{\mathcal{G}}} \sum_{t=t_1(a)}^{t_2(a)} (1 - \mu_t(a))\right] \\
&= \mathbb{E}\left[\sum_{a \in \mathcal{A}_T^{\mathcal{G}}} \left(\Delta_1(a) n_{[t_1,t_2]}(a) + \sum_{t=t_1(a)+1}^{t_2(a)} V_{[t_1,t-1]}(a)\right)\right] \\
&\le \mathbb{E}\left[2T\delta + \sum_{a \in \mathcal{A}^1} \rho_{t_1(a)} + \sum_{a \in \mathcal{A}_T^{\mathcal{G}}/\{\overline{\mathcal{A}}_T^{\mathcal{G}} \cup \mathcal{A}^0 \cup \mathcal{A}^1\}} \sum_{t=t_1(a)+1}^{t_2(a)} V_{[t_1,t-1]}(a)\right. \\
&\quad \left. + \sum_{a \in \overline{\mathcal{A}}_T^{\mathcal{G}}} \left(\sum_{t=t_1(a)+1}^{t_1(a)+w(a)} V_{[t_1,t-1]}(a) + \sum_{t=t_1(a)+w(a)+1}^{t_2(a)} V_{[t_1,t-1]}(a)\right)\right],
\end{aligned}
\tag{8}
$$

where the first inequality comes from $\Delta_1(a) \le 2\delta$ for any $a \in \mathcal{A}_T^{\mathcal{G}}$. For the second term in the right hand side of the last inequality (8),

$$
\sum_{a \in \mathcal{A}^1} \rho_{t_1(a)} \le V_T.
\tag{9}
$$

For the third term in (8), from the fact that $n_{[t_1+1,t_2]}(a) = n_{[t_1,t_2-1]}(a) < w(a)$ for any $a \in \mathcal{A}_T^{\mathcal{G}}/\overline{\mathcal{A}}_T^{\mathcal{G}}$ from the definition of $\overline{\mathcal{A}}_T^{\mathcal{G}}$, we have

$$
\begin{aligned}
&\sum_{a \in \mathcal{A}_T^{\mathcal{G}}/\{\overline{\mathcal{A}}_T^{\mathcal{G}} \cup \mathcal{A}^0 \cup \mathcal{A}^1\}} \sum_{t=t_1(a)+1}^{t_2(a)} V_{[t_1,t-1]}(a) \\
&\le \sum_{a \in \mathcal{A}_T^{\mathcal{G}}/\{\overline{\mathcal{A}}_T^{\mathcal{G}} \cup \mathcal{A}^0 \cup \mathcal{A}^1\}} n_{[t_1+1,t_2]}(a) V_{[t_1,t_2-2]}(a) + \rho_{t_2(a)-1} \\
&= O\left(V_T + \sum_{a \in \mathcal{A}_T^{\mathcal{G}}/\{\overline{\mathcal{A}}_T^{\mathcal{G}} \cup \mathcal{A}^0 \cup \mathcal{A}^1\}} w(a) V_{[t_1,t_2-2]}(a)\right) \\
&= \tilde{O}\left(V_T + \sum_{a \in \mathcal{A}_T^{\mathcal{G}}/\{\overline{\mathcal{A}}_T^{\mathcal{G}} \cup \mathcal{A}^0 \cup \mathcal{A}^1\}} n_{[t_1,t_2-2]}(a)^{2/3} V_{[t_1,t_2-2]}(a)^{1/3}\right).
\end{aligned}
\tag{10}
$$

Now, we focus on the fourth term in (8). From $t_1(a) + w(a) + 1 \leq t_2(a)$ for $a \in \overline{\mathcal{A}}_T^{\mathcal{G}}$ from the definition of $\overline{\mathcal{A}}_T^{\mathcal{G}}$ and (10), we first have

$$
\begin{aligned}
\sum_{a \in \overline{\mathcal{A}}_T^{\mathcal{G}}} \sum_{t=t_1(a)+1}^{t_1(a)+w(a)} V_{[t_1,t-1]}(a) &= \sum_{a \in \overline{\mathcal{A}}_T^{\mathcal{G}}} \sum_{t=t_1(a)+1}^{t_1(a)+w(a)} \sum_{s=t_1}^{t-1} \rho_s \\
&\leq \sum_{a \in \overline{\mathcal{A}}_T^{\mathcal{G}}} \sum_{t=t_1(a)+1}^{t_1(a)+w(a)} \sum_{s=t_1(a)}^{t_2(a)-2} \rho_s \\
&\leq \sum_{a \in \overline{\mathcal{A}}_T^{\mathcal{G}}} w(a) V_{[t_1,t_2-2]}(a) \\
&= \tilde{O}\left( \sum_{a \in \overline{\mathcal{A}}_T^{\mathcal{G}}} n_{[t_1,t_2-2]}(a)^{2/3} V_{[t_1,t_2-2]}(a)^{1/3} \right). \qquad (11)
\end{aligned}
$$

Now we focus on $\sum_{a \in \overline{\mathcal{A}}_T^{\mathcal{G}}} \sum_{t=t_1(a)+w(a)+1}^{t_2(a)} V_{[t_1,t-1]}(a)$ in (8). From the definition of $t_2$ and the threshold condition in the algorithm with (7), for any $t_1 \leq t \leq t_2$ and any $t_1 \leq s \leq t-1$ s.t. $s = t - 2^{l-1}$ for $l \in \mathbb{Z}^+$, we have

$$
1 - V_{[t_1,t-2]}(a) + n_{[s,t-1]}(a)\overline{\rho}_{[s,t-2]}(a) + 2\sqrt{12\log(T)/n_{[s,t-1]}(a)} \geq 1 - \delta. \qquad (12)
$$

For $t \geq t_1 + w(a) + 1$, there always exists $t_1 \leq s(t) \leq t-1$ such that $w(a)/2 \leq n_{[s(t),t-1]}(a) \leq w(a)$ and $s(t) = t - 2^{l-1}$ for $l \in \mathbb{Z}^+$. Then from (12) with $s = s(t)$, we have

$$
V_{[t_1,t-2]}(a) = \tilde{O}\left( \delta + \overline{\rho}_{[s(t),t-2]}(a)/\overline{\rho}_{[t_1,t_2-2]}(a)^{2/3} + \overline{\rho}_{[t_1,t_2-2]}(a)^{1/3} \right). \qquad (13)
$$

Using the facts that $n_{[s(t),t-2]}(a) \geq n_{[s(t),t-1]}(a)/2 \geq w(a)/4$ and $t - s(t) \leq w(a)$ from $n_{[s(t),t-1]}(a) \leq w(a)$, we can obtain that

$$
\begin{aligned}
\sum_{t=t_1(a)+1+w(a)}^{t_2(a)} \overline{\rho}_{[s(t),t-2]}(a) &\leq \sum_{t=t_1(a)+1+w(a)}^{t_2(a)} \frac{\sum_{k=t-w(a)}^{t-2} \rho_k}{n_{[s(t),t-2]}(a)} \\
&\leq \sum_{t=t_1(a)+1}^{t_2(a)-2} \frac{w(a)\rho_t}{n_{[s(t),t-2]}(a)} \\
&\leq 4 \sum_{t=t_1(a)}^{t_2(a)-2} \rho_t, \qquad (14)
\end{aligned}
$$

where the second inequality is obtained from the fact that the number of times that $\rho_t$ is duplicated for each $t \in [t_1(a) + 1, t_2(a) - 2]$ in the expression $\sum_{t=t_1(a)+1+w(a)}^{t_2(a)} \sum_{k=t-w(a)}^{t-2} \rho_k$ is at most $w(a)$. Then with (13) and (14), using the fact that

$$
\sum_{t_1(a)+1+w(a)}^{t_2(a)} \overline{\rho}_{[t_1,t_2-2]}(a)^{1/3} \leq n_{[t_1,t_2-2]}(a)\overline{\rho}_{[t_1,t_2-2]}(a)^{1/3} = O(n_{[t_1,t_2-2]}(a)^{2/3} V_{[t_1,t_2-2]}(a)^{1/3}),
$$

we have

$$\sum_{a\in\overline{\mathcal{A}}_T^{\mathcal{G}}}\sum_{t=t_1(a)+1+w(a)}^{t_2(a)}V_{[t_1,t-1]}(a)$$

$$\leq \sum_{a\in\overline{\mathcal{A}}_T^{\mathcal{G}}}\sum_{t=t_1(a)+1+w(a)}^{t_2(a)}V_{[t_1,t-2]}(a)+\rho_{t_2(a)-1}$$

$$=\tilde{O}\left(\delta T+V_T+\sum_{a\in\overline{\mathcal{A}}_T^{\mathcal{G}}}\sum_{t=t_1(a)+1+w(a)}^{t_2(a)}\overline{\rho}_{[s(t),t-2]}(a)/\overline{\rho}_{[t_1,t_2-2]}(a)^{2/3}+\sum_{a\in\overline{\mathcal{A}}_T^{\mathcal{G}}}\sum_{t=t_1(a)+1+w(a)}^{t_2(a)}\overline{\rho}_{[t_1,t_2-2]}(a)^{1/3}\right)$$

$$=\tilde{O}\left(\delta T+V_T+\sum_{a\in\overline{\mathcal{A}}_T^{\mathcal{G}}}\sum_{t=t_1(a)+1+w(a)}^{t_2(a)}\overline{\rho}_{[s(t),t-2]}(a)/\overline{\rho}_{[t_1,t_2-2]}(a)^{2/3}+\sum_{a\in\overline{\mathcal{A}}_T^{\mathcal{G}}}n_{[t_1,t_2-2]}(a)^{2/3}V_{[t_1,t_2-2]}(a)^{1/3}\right)$$

$$=\tilde{O}\left(\delta T+V_T+\sum_{a\in\overline{\mathcal{A}}_T^{\mathcal{G}}}\sum_{t=t_1(a)}^{t_2(a)-2}\rho_t/\overline{\rho}_{[t_1,t_2-2]}(a)^{2/3}+\sum_{a\in\overline{\mathcal{A}}_T^{\mathcal{G}}}n_{[t_1,t_2-2]}(a)^{2/3}V_{[t_1,t_2-2]}(a)^{1/3}\right)$$

$$=\tilde{O}\left(\delta T+V_T+\sum_{a\in\overline{\mathcal{A}}_T^{\mathcal{G}}}n_{[t_1,t_2-2]}(a)^{2/3}V_{[t_1,t_2-2]}(a)^{1/3}\right). \tag{15}$$

Then putting the results from (8),(10),(11), and (15) altogether, we have

$$\mathbb{E}[R^{\mathcal{G}}(T)]$$

$$\leq \mathbb{E}\left[\sum_{a\in\mathcal{A}_T^{\mathcal{G}}}\left(\Delta_1(a)n_{[t_1,t_2]}(a)+\sum_{t=t_1(a)+1}^{t_2(a)}V_{[t_1,t-1]}(a)\right)\right]$$

$$=\tilde{O}\left(T\delta+V_T+\mathbb{E}\left[\sum_{a\in\mathcal{A}_T^{\mathcal{G}}/\{\mathcal{A}^0\cup\mathcal{A}^1\}}V_{[t_1,t_2-2]}(a)^{1/3}n_{[t_1,t_2-2]}(a)^{2/3}\right]\right)$$

$$=\tilde{O}\left(T\delta+V_T^{1/3}T^{2/3}\right), \tag{16}$$

where the last equality comes from Hölder's inequality and $V_T\leq T$. This concludes the proof. $\quad\square$

Now, we provide a bound for $R^{\mathcal{B}}(T)$. We note that the initially bad arms can be defined only when $2\delta<1$. Otherwise when $2\delta\geq 1$, we have $R(T)=R^{\mathcal{G}}(T)$, which completes the proof. Therefore, for the regret from bad arms, we consider the case of $2\delta<1$. We adopt the episodic approach in Kim et al. [13] for the remaining regret analysis. The episodic approach is reformulated using the cumulative amount of rotting instead of the maximum rotting rate. In the following, we define some notation.

Given a policy sampling arms in the sequence order, let $m^{\mathcal{G}}$ be the number of samples of distinct good arms and $m_i^{\mathcal{B}}$ be the number of consecutive samples of distinct bad arms between the $i-1$-st and $i$-th sample of a good arm among $m^{\mathcal{G}}$ good arms. We refer to the period starting from sampling the $i-1$-st good arm before sampling the $i$-th good arm as the $i$-th *episode*. Observe that $m_1^{\mathcal{B}},\ldots,m_{m^{\mathcal{G}}}^{\mathcal{B}}$ are i.i.d. random variables with geometric distribution with parameter $2\delta$, given a fixed value of $m^{\mathcal{G}}$. Therefore, with some constant $C>0$, for non-negative integer $k$ we have $\mathbb{P}(m_i^{\mathcal{B}}=k)=(1-C(2\delta)^{\beta})^kC(2\delta)^{\beta}$, for $i=1,\ldots,m^{\mathcal{G}}$. Define $\tilde{m}_T$ to be the number of episodes from the policy $\pi$ over the horizon $T$, $\tilde{m}_T^{\mathcal{G}}$ to be the total number of samples of a good arm by the policy $\pi$ over the horizon $T$ such that $\tilde{m}_T^{\mathcal{G}}=\tilde{m}_T$ or $\tilde{m}_T^{\mathcal{G}}=\tilde{m}-1$, and $\tilde{m}_{i,T}^{\mathcal{B}}$ to be the number of samples of a bad arm in the $i$-th episode by the policy $\pi$ over the horizon $T$.

Under a policy $\pi$, let $R_{i,j}^{\mathcal{B}}$ be the regret (summation of mean reward gaps) contributed by pulling the $j$-th bad arm in the $i$-th episode. Then let $R_{m^{\mathcal{G}}}^{\mathcal{B}} = \sum_{i=1}^{m^{\mathcal{G}}} \sum_{j \in [m_i^{\mathcal{B}}]} R_{i,j}^{\mathcal{B}}$, which is the regret from initially bad arms over the period of $m^{\mathcal{G}}$ episodes.

Let $a(i)$ be a good arm in the $i$-th episode and $a(i,j)$ be a $j$-th bad arm in the $i$-th episode. We define $V_T(a) = \sum_{t=1}^{T} \rho_t \mathbb{1}(a_t = a)$. Then excluding the last episode $\tilde{m}_T$ over $T$, we provide lower bounds of the total rotting variation over $T$ for $a(i)$, denoted by $V_T(a(i))$, in the following lemma.

**Lemma A.2.** *Under $E_1$, given $\tilde{m}_T$, for any $i \in [\tilde{m}_T^{\mathcal{G}}]/\{\tilde{m}_T\}$ we have*

$$V_T(a(i)) \geq \delta/2.$$

*Proof.* Suppose that $V_T(a(i)) < \delta/2$, then we have

$$\min_{t_1(a(i)) \leq s \leq t_2(a(i))} \left\{ \widehat{\mu}_{[s,t_2(a(i))]}(a(i)) + \sqrt{12 \log(T)/n_{[s,t_2(a(i))]}(a(i))} \right\}$$
$$\geq \min_{t_1(a(i)) \leq s \leq t_2(a(i))} \left\{ \overline{\mu}_{[s,t_2(a(i))]}(a(i)) \right\}$$
$$\geq \mu_{t_2(a(i))}(a(i))$$
$$\geq \mu_1(a(i)) - V_T(a(i))$$
$$> 1 - \delta,$$

where the first inequality is obtained from $E_1$, and the last inequality is from $V_T(a(i)) < \delta/2$ and $\mu_1(a(i)) \geq 1 - \delta/2$. Therefore, from the threshold condition, policy $\pi$ must pull arm $a(i)$ until its total rotting amount is greater than (or equal to) $\delta/2$, which implies $V_T(a(i)) \geq \delta/2$. $\square$

In the following, we consider two different cases with respect to $V_T$; large and small $V_T$.

**Case 1:** We consider $V_T > \max\{1/\sqrt{T}, 1/T^{1/(\beta+1)}\}$ in the following.

In this case, we have $\delta = \delta_V(\beta) = c_1 \max\{(V_T/T)^{1/(\beta+2)}, (V_T/T)^{1/3}\}$. Here, we define the policy $\pi$ after time $T$ such that it pulls a good arm until its total rotting variation is equal to or greater than $\delta/2$ and does not pull a sampled bad arm. We note that defining how $\pi$ works after $T$ is only for the proof to get a regret bound over time horizon $T$. For the last arm $\tilde{a}$ over the horizon $T$, it pulls the arm until its total variation becomes $\max\{\delta/2, V_T(\tilde{a})\}$ if $\tilde{a}$ is a good arm. For $i \in [m^{\mathcal{G}}], j \in [m_i^{\mathcal{B}}]$ let $V_i^{\mathcal{G}}$ and $V_{i,j}^{\mathcal{B}}$ be the total rotting variation of pulling the good arm in $i$-th episode and $j$-th bad arm in $i$-th episode from the policy, respectively. Here we define $V_i^{\mathcal{G}}$'s and $V_{i,j}^{\mathcal{B}}$'s as follows:

If $\tilde{a}$ is a good arm,

$$V_i^{\mathcal{G}} = \begin{cases} V_T(a(i)) & \text{for } i \in [\tilde{m}_T^{\mathcal{G}} - 1] \\ \max\{\delta/2, V_T(a(i))\} & \text{for } i \in [m^{\mathcal{G}}]/[\tilde{m}_T^{\mathcal{G}} - 1] \end{cases}, \quad V_{i,j}^{\mathcal{B}} = \begin{cases} V_T(a(i,j)) & \text{for } i \in [\tilde{m}_T^{\mathcal{G}}], j \in [\tilde{m}_{i,T}^{\mathcal{B}}] \\ 0 & \text{for } i \in [m^{\mathcal{G}}]/[\tilde{m}_T^{\mathcal{G}}], j \in [m_i^{\mathcal{B}}]. \end{cases}$$

Otherwise,

$$V_i^{\mathcal{G}} = \begin{cases} V_T(a(i)) & \text{for } i \in [\tilde{m}_T^{\mathcal{G}}] \\ \delta/2 & \text{for } i \in [m^{\mathcal{G}}]/[\tilde{m}_T^{\mathcal{G}}] \end{cases}, \quad V_{i,j}^{\mathcal{B}} = \begin{cases} V_T(a(i,j)) & \text{for } i \in [\tilde{m}_T^{\mathcal{G}}], j \in [\tilde{m}_{i,T}^{\mathcal{B}}] \\ 0 & \text{for } i \in [m^{\mathcal{G}}]/[\tilde{m}_T^{\mathcal{G}} - 1], j \in [m_i^{\mathcal{B}}]/[\tilde{m}_{i,T}^{\mathcal{B}}]. \end{cases}$$

For $i \in [m^{\mathcal{G}}], j \in [m_i^{\mathcal{B}}]$ let $n_{i,j}^{\mathcal{B}}$ be the number of pulling the $j$-th bad arm in $i$-th episode from the policy. We define $n_T(a)$ be the total amount of pulling arm $a$ over $T$. Here we define $n_{i,j}^{\mathcal{B}}$'s as follows:

$$n_{i,j}^{\mathcal{B}} = \begin{cases} n_T(a(i,j)) & \text{for } i \in [\tilde{m}_T^{\mathcal{G}}], j \in [\tilde{m}_{i,T}^{\mathcal{B}}] \\ 0 & \text{for } i \in [m^{\mathcal{G}}]/[\tilde{m}_T^{\mathcal{G}}], j \in [m_i^{\mathcal{B}}]. \end{cases}$$

Then we provide $m^{\mathcal{G}}$ such that $R^{\mathcal{B}}(T) \leq R_{m^{\mathcal{G}}}^{\mathcal{B}}$ in the following lemma.

**Lemma A.3.** *Under $E_1$, when $m^{\mathcal{G}} = \lceil 2V_T/\delta \rceil$ we have*

$$R^{\mathcal{B}}(T) \leq R_{m^{\mathcal{G}}}^{\mathcal{B}}.$$

*Proof.* From Lemma A.2, we have

$$\sum_{i \in [m^{\mathcal{G}}]} V_i^{\mathcal{G}} \geq m^{\mathcal{G}} \frac{\delta}{2} \geq V_T,$$

which implies that $R^{\mathcal{B}}(T) \leq R^{\mathcal{B}}_{m^{\mathcal{G}}}$. $\qquad\square$

From the result of Lemma A.3, we set $m^{\mathcal{G}} = \lceil 2V_T/\delta \rceil$. We analyze $R^{\mathcal{B}}_{m^{\mathcal{G}}}$ for obtaining a bound for $R^{\mathcal{B}}(T)$ in the following.

**Lemma A.4.** *Under $E_1$ and policy $\pi$, we have*

$$\mathbb{E}[R^{\mathcal{B}}_{m^{\mathcal{G}}}] = \tilde{O}\left(\max\{T^{(\beta+1)/(\beta+2)}V_T^{1/(\beta+2)}, T^{2/3}V_T^{1/3}\}\right).$$

*Proof.* Let $a(i,j)$ be a sampled arm for $j$-th bad arm in the $i$-th episode and $\tilde{m}_T$ be the number of episodes from the policy $\pi$ over the horizon $T$. Suppose that the algorithm samples arm $a(i,j)$ at time $t_1(a(i,j))$. Then the algorithm stops pulling arm $a(i,j)$ at time $t_2(a(i,j)) + 1$ if $\widehat{\mu}_{[s,t_2(a(i,j))]}(a) + \sqrt{12\log(T)/n_{[s,t_2(a(i,j))]}(a)} < 1 - \delta$ for some $s$ such that $t_1(a(i,j)) \leq s \leq t_2(a(i,j))$ and $s = t_2(a(i,j)) + 1 - 2^{l-1}$ for $l \in \mathbb{Z}^+$. For simplicity, we use $t_1$ and $t_2$ instead of $t_1(a(i,j))$ and $t_2(a(i,j))$ when there is no confusion. We first consider the case where the algorithm stops pulling arm $a(i,j)$ because the threshold condition is satisfied. For the regret analysis, we consider that for $t > t_2$, arm $a$ is virtually pulled. We note that under $E_1$, we have

$$\widehat{\mu}_{[s,t_2]}(a(i,j)) + \sqrt{12\log(T)/n_{[s,t_2]}(a(i,j))} \leq \overline{\mu}_{[s,t_2]}(a(i,j)) + 2\sqrt{12\log(T)/n_{[s,t_2]}(a(i,j))}$$

$$\leq \mu_1(a(i,j)) + 2\sqrt{12\log(T)/n_{[s,t_2]}(a(i,j))}.$$

Then we assume that $\tilde{t}_2(\geq t_2)$ is the smallest time that there exists $t_1 \leq s \leq \tilde{t}_2$ with $s = \tilde{t}_2 + 1 - 2^{l-1}$ for $l \in \mathbb{Z}^+$ such that the following threshold condition is met:

$$\mu_1(a(i,j)) + 2\sqrt{12\log(T)/n_{[s,\tilde{t}_2]}(a(i,j))} < 1 - \delta. \tag{17}$$

From the definition of $\tilde{t}_2$, we observe that for given $\tilde{t}_2$, the time step $s = s'$ which satisfying (17) equals to $t_1$ (i.e. $s' = t_1$). Then, we can observe that $n_{[s',\tilde{t}_2]}(a(i,j)) = n_{[t_1,\tilde{t}_2]}(a(i,j)) = \lceil C_2 \log(T)/(\Delta_{t_1}(a(i,j)) - \delta)^2 \rceil$ for some constant $C_2 > 0$, which satisfies (17). Then from $n_{[t_1,t_2]}(a(i,j)) \leq n_{[t_1,\tilde{t}_2]}(a(i,j))$, for all $i \in [\tilde{m}_T], j \in [\tilde{m}^{\mathcal{B}}_{i,T}]$ we have $n^{\mathcal{B}}_{i,j} = \tilde{O}(1/(\Delta_1(a(i,j)) - \delta)^2)$. Then with the facts that $n^{\mathcal{B}}_{i,j} = 0$ for $i \in [m^{\mathcal{G}}]/[\tilde{m}^{\mathcal{G}}_T], j \in [m^{\mathcal{B}}_i]/[\tilde{m}^{\mathcal{B}}_{i,T}]$, we have, for any $i \in [m^{\mathcal{G}}]$ and $j \in [m^{\mathcal{B}}_i]$,

$$n^{\mathcal{B}}_{i,j} = \tilde{O}(1/(\Delta_{t_1}(a(i,j)) - \delta)^2).$$

For $2\delta < x \leq 1$, let $b(x) = \mathbb{P}(\Delta_1(a) = x | a \text{ is a bad arm})$. Then we have

$$b(x) = \mathbb{P}(\Delta_1(a) = x | \Delta_1(a) > 2\delta)$$
$$= \mathbb{P}(\Delta_1(a) = x)/\mathbb{P}(\Delta_1(a) > 2\delta)$$
$$= \mathbb{P}(\Delta_1(a) = x)/(1 - C(2\delta)^{\beta}),$$

where $C(2\delta)^{\beta} < 1$ with small enough positive constant $c_1 < 1$ for $\delta$. We note that $2\delta < \Delta_{t_1}(a(i,j)) = \Delta_1(a(i,j)) \leq 1$. Since $n^{\mathcal{B}}_{i,j} = \tilde{O}(1/(\Delta_{t_1}(a(i,j)) - \delta)^2) = \tilde{O}(1/\delta^2)$, we have

$$\mathbb{E}[R^{\mathcal{B}}_{i,j}] = \mathbb{E}\left[\sum_{t=t_1(a(i,j))}^{t_2(a(i,j))} \Delta_{t_1}(a(i,j)) + \sum_{t=t_1(a(i,j))}^{t_2(a(i,j))-1} \sum_{s=t_1(a(i,j))}^{t} \rho_s\right]$$
$$\leq \mathbb{E}[\Delta_1(a(i,j))n^{\mathcal{B}}_{i,j} + V^{\mathcal{B}}_{i,j}n^{\mathcal{B}}_{i,j}]$$
$$\leq \mathbb{E}[\Delta_1(a(i,j))n^{\mathcal{B}}_{i,j} + V^{\mathcal{B}}_{i,j}(1/\delta^2)]$$
$$= \tilde{O}\left(\int_{2\delta}^{1} \frac{1}{(x-\delta)^2} x b(x) dx + \mathbb{E}[V^{\mathcal{B}}_{i,j}(1/\delta^2)]\right). \tag{18}$$

Recall that we consider $2\delta < 1$ for regret from bad arms. We adopt some techniques introduced in Appendix D of Bayati et al. [5] to deal with the generalized mean reward distribution with $\beta$. Let $K = (1 - 2\delta)/\delta$, $a_j = \frac{2}{j\delta}$, and $p_j = \int_{j\delta}^{(j+1)\delta} b(t + \delta)dt$. Then for obtaining a bound of the last equality in (18) we have

$$
\begin{aligned}
\int_{2\delta}^{1} \left( \frac{1}{(x - \delta)^2} x \right) b(x)dx &= \int_{\delta}^{1-\delta} \left( \frac{1}{t} + \frac{\delta}{t^2} \right) b(t + \delta)dt \\
&= \sum_{j=1}^{K} \int_{j\delta}^{(j+1)\delta} \left( \frac{1}{t} + \frac{\delta}{t^2} \right) b(t + \delta)dt \\
&\leq \sum_{j=1}^{K} \frac{2}{j\delta} \int_{j\delta}^{(j+1)\delta} b(t + \delta)dt \\
&= \sum_{j=1}^{K} a_j p_j.
\end{aligned}
\tag{19}
$$

We note that $\sum_{i=1}^{j} p_i \leq C_0(j\delta)^\beta$ for all $j \in [K]$ for some constant $C_0 > 0$. Then for getting a bound of the last equality in (19), we have

$$
\begin{aligned}
\sum_{j=1}^{K} a_j p_j &= \sum_{j=1}^{K-1} (a_j - a_{j+1}) \left( \sum_{i=1}^{j} p_i \right) + a_K \sum_{i=1}^{K} p_i \\
&\leq \sum_{j=1}^{K-1} (a_j - a_{j+1}) C_0(j\delta)^\beta + a_K C_0(K\delta)^\beta \\
&= C_0 \delta^\beta a_1 + \sum_{j=2}^{K} C_0(j^\beta - (j - 1)^\beta)\delta^\beta a_j \\
&= O\left( \left( \frac{1}{\delta} \right) \delta^\beta + \sum_{j=2}^{K} \left( \frac{1}{j\delta} \right) \left( (j\delta)^\beta - ((j - 1)\delta)^\beta \right) \right) \\
&= O\left( \delta^{\beta-1} + \sum_{j=2}^{K} \left( \frac{1}{j} \delta^{\beta-1} \right) \left( j^\beta - (j - 1)^\beta \right) \right).
\end{aligned}
\tag{20}
$$

Now we analyze the term in the last equality in (20) according to the criteria for $\beta$. For $\beta = 1$, we can obtain

$$
O\left( \delta^{\beta-1} + \sum_{j=2}^{K} \left( \frac{1}{j} \delta^{\beta-1} \right) \left( j^\beta - (j - 1)^\beta \right) \right) = \tilde{O}(1).
\tag{21}
$$

For $\beta > 1$, we have $j^\beta - (j - 1)^\beta \leq \beta j^{\beta-1}$ using the mean value theorem. Therefore, we obtain the following.

$$
\begin{aligned}
O\left( \delta^{\beta-1} + \sum_{j=2}^{K} \left( \frac{1}{j} \delta^{\beta-1} \right) \left( j^\beta - (j - 1)^\beta \right) \right) &= O\left( \sum_{j=1}^{K} \left( \frac{1}{j} \delta^{\beta-1} \right) j^{\beta-1} \right) \\
&= O\left( \sum_{j=2}^{K} \delta^{\beta-1} j^{\beta-2} \right) \\
&= O\left( \delta^{\beta-1} \frac{1}{\beta - 1} \left( (K + 1)^{\beta-1} - 1 \right) \right) \\
&= O(1).
\end{aligned}
\tag{22}
$$

For $\beta < 1$, when $j > 1$ we have $j^\beta - (j-1)^\beta \leq \beta(j-1)^{\beta-1}$ using the mean value theorem. Therefore, we obtain

$$O\left(\delta^{\beta-1} + \sum_{j=1}^{K}\left(\frac{1}{j}\delta^{\beta-1}\right)(j^\beta - (j-1)^\beta)\right) = O\left(\delta^{\beta-1} + \sum_{j=2}^{K}\left(\frac{1}{j}\delta^{\beta-1}\right)(j-1)^{\beta-1}\right)$$

$$= O\left(\delta^{\beta-1} + \sum_{j=2}^{K}\delta^{\beta-1}(j-1)^{\beta-2}\right)$$

$$= O\left(\delta^{\beta-1} + \delta^{\beta-1}\frac{1}{\beta-1}\left((K+1)^{\beta-1} - 1\right)\right)$$

$$= O\left(\delta^{\beta-1} + \delta^{\beta-1}\frac{1 - ((1-\delta)/\delta)^{\beta-1}}{1-\beta}\right) = O(\delta^{\beta-1}). \tag{23}$$

From (19),(20),(21),(22), and (23), we have

$$\int_{2\delta}^{1}\left(\frac{1}{(x-\delta)^2}x\right)b(x)dx = \tilde{O}(\max\{1, \delta^{\beta-1}\}).$$

Then for any $i \in [m^{\mathcal{G}}], j \in [m_i^{\mathcal{B}}]$, we have

$$\mathbb{E}[R_{i,j}^{\mathcal{B}}] \leq \mathbb{E}\left[\Delta(a(i,j))n_{i,j}^{\mathcal{B}} + V_{i,j}^{\mathcal{B}}n_{i,j}^{\mathcal{B}}\right]$$

$$= \tilde{O}\left(\max\{1, \delta^{\beta-1}\} + \mathbb{E}[V_{i,j}^{\mathcal{B}}]/\delta^2\right). \tag{24}$$

Recall that $R_{m^{\mathcal{G}}}^{\mathcal{B}} = \sum_{i=1}^{m^{\mathcal{G}}}\sum_{j\in[m_i^{\mathcal{B}}]}R_{i,j}^{\mathcal{B}}$. With $\delta = c_1\max\{(V_T/T)^{1/(\beta+2)}, (V_T/T)^{1/3}\}$ and $m^{\mathcal{G}} = \lceil 2V_T/\delta \rceil$, from the fact that $m_i^{\mathcal{B}}$'s are i.i.d. random variables with geometric distribution with $\mathbb{E}[m_i^{\mathcal{B}}] = (1/C(2\delta)^\beta) - 1$ for some constant $C > 0$, we have

$$\mathbb{E}[R_{m^{\mathcal{G}}}^{\mathcal{B}}] = O\left(\mathbb{E}\left[\sum_{i=1}^{m^{\mathcal{G}}}\sum_{j\in[m_i^{\mathcal{B}}]}R_{i,j}^{\mathcal{B}}\right]\right)$$

$$= \tilde{O}\left((V_T/\delta)\frac{1}{\delta^\beta}\max\{1, \delta^{\beta-1}\} + V_T/\delta^2\right)$$

$$= \tilde{O}\left(\max\{T^{(\beta+1)/(\beta+2)}V_T^{1/(\beta+2)}, T^{2/3}V_T^{1/3}\}\right). \tag{25}$$

$\square$

From $R^\pi(T) = R^{\mathcal{G}}(T) + R^{\mathcal{B}}(T)$ and Lemmas A.1, A.3, A.4, with $\delta = \max\{(V_T/T)^{1/(\beta+2)}, (V_T/T)^{1/3}\}$ we have

$$\mathbb{E}[R^\pi(T)] = \tilde{O}\left(\max\{T^{(\beta+1)/(\beta+2)}V_T^{1/(\beta+2)}, T^{2/3}V_T^{1/3}\}\right). \tag{26}$$

**Case 2:** Now we consider $V_T \leq \max\{1/\sqrt{T}, 1/T^{1/(\beta+1)}\}$ in the following. In this case, we have $\delta = c_1\max\{1/T^{\frac{1}{\beta+1}}, 1/\sqrt{T}\}$. For getting $R_{m^{\mathcal{G}}}^{\mathcal{B}}$, here we define the policy $\pi$ after time $T$ such that it pulls $V_T$ amount of rotting variation for a good arm and 0 for a bad arm. We note that defining how $\pi$ works after $T$ is only for the proof to get a regret bound over time horizon $T$. For the last arm $\tilde{a}$ over the horizon $T$, it pulls the arm up to $V_T$ amount of rotting variation if $\tilde{a}$ is a good arm. For $i \in [m^{\mathcal{G}}], j \in [m_i^{\mathcal{B}}]$ let $V_i^{\mathcal{G}}$ and $V_{i,j}^{\mathcal{B}}$ be the amount of rotting variation from pulling the good arm in $i$-th episode and $j$-th bad arm in $i$-th episode from the policy, respectively. Here we define $V_i^{\mathcal{G}}$'s and $V_{i,j}^{\mathcal{B}}$'s as follows:

If $\tilde{a}$ is a good arm,

$$V_i^{\mathcal{G}} = \begin{cases} V_T(a(i)) & \text{for } i \in [\tilde{m}_T^{\mathcal{G}} - 1] \\ V_T & \text{for } i \in [m^{\mathcal{G}}]/[\tilde{m}_T^{\mathcal{G}} - 1] \end{cases}, V_{i,j}^{\mathcal{B}} = \begin{cases} V_T(a(i,j)) & \text{for } i \in [\tilde{m}_T^{\mathcal{G}}], j \in [\tilde{m}_{i,T}^{\mathcal{B}}] \\ 0 & \text{for } i \in [m^{\mathcal{G}}]/[\tilde{m}_T^{\mathcal{G}}], j \in [m_i^{\mathcal{B}}]. \end{cases}$$

Otherwise,

$$V_i^{\mathcal{G}} = \begin{cases} V_T(a(i)) & \text{for } i \in [\tilde{m}_T^{\mathcal{G}}] \\ V_T & \text{for } i \in [m^{\mathcal{G}}]/[\tilde{m}_T^{\mathcal{G}}] \end{cases}, V_{i,j}^{\mathcal{B}} = \begin{cases} V_T(a(i,j)) & \text{for } i \in [\tilde{m}_T^{\mathcal{G}}], j \in [\tilde{m}_{i,T}^{\mathcal{B}}] \\ 0 & \text{for } i \in [m^{\mathcal{G}}]/[\tilde{m}_T^{\mathcal{G}} - 1], j \in [m_i^{\mathcal{B}}]/[\tilde{m}_{i,T}^{\mathcal{B}}]. \end{cases}$$

For $i \in [m^{\mathcal{G}}]$, $j \in [m_i^{\mathcal{B}}]$ let $n_{i,j}^{\mathcal{B}}$ be the number of pulling the $j$-th bad arm in $i$-th episode from the policy. We define $n_T(a)$ be the total amount of pulling arm $a$ over $T$. Here we define $n_{i,j}^{\mathcal{B}}$'s as follows:

$$n_{i,j}^{\mathcal{B}} = \begin{cases} n_T(a(i,j)) & \text{for } i \in [\tilde{m}_T^{\mathcal{G}}], j \in [\tilde{m}_{i,T}^{\mathcal{B}}] \\ 0 & \text{for } i \in [m^{\mathcal{G}}]/[\tilde{m}_T^{\mathcal{G}}], j \in [m_i^{\mathcal{B}}]. \end{cases}$$

Then we provide $m^{\mathcal{G}}$ such that $R^{\mathcal{B}}(T) \leq R_{m^{\mathcal{G}}}^{\mathcal{B}}$ in the following lemma.

**Lemma A.5.** *Under $E_1$, when $m^{\mathcal{G}} = C_3$ for some constant $C_3 > 0$, we have*
$$R^{\mathcal{B}}(T) \leq R_{m^{\mathcal{G}}}^{\mathcal{B}}.$$

*Proof.* From Lemma A.2, under $E_1$ we can find that $V_i^{\mathcal{G}} \geq \min\{\delta/2, V_T\}$ for $i \in [m^{\mathcal{G}}]$. Then if $m^{\mathcal{G}} = C_3$ for large enough $C_3 > 0$, then with $\delta = c_1 \max\{1/T^{1/(\beta+1)}, 1/\sqrt{T}\}$ and $V_T \leq \max\{1/T^{1/(\beta+1)}, 1/\sqrt{T}\}$, we have

$$\sum_{i \in [m^{\mathcal{G}}]} V_i^{\mathcal{G}} \geq C_3 \min\{\delta/2, V_T\} > V_T,$$

which implies $R^{\mathcal{B}}(T) \leq R_{m^{\mathcal{G}}}^{\mathcal{B}}$.

$\square$

We analyze $R_{m^{\mathcal{G}}}^{\mathcal{B}}$ for obtaining a bound for $R^{\mathcal{B}}(T)$ in the following.

**Lemma A.6.** *Under $E_1$ and policy $\pi$, we have*
$$\mathbb{E}[R_{m^{\mathcal{G}}}^{\mathcal{B}}] = \tilde{O}\left(\max\{T^{\beta/(\beta+1)}, \sqrt{T}\}\right).$$

*Proof.* From (24), for any $i \in [m^{\mathcal{G}}]$, $j \in [m_i^{\mathcal{B}}]$, we have
$$\mathbb{E}[R_{i,j}^{\mathcal{B}}] \leq \mathbb{E}\left[\Delta(a(i,j))n_{i,j}^{\mathcal{B}} + V_{i,j}^{\mathcal{B}}n_{i,j}^{\mathcal{B}}\right]$$
$$= \tilde{O}\left(\max\{1, \delta^{\beta-1}\} + \mathbb{E}[V_{i,j}^{\mathcal{B}}]/\delta^2\right).$$

Recall that $R_{m^{\mathcal{G}}}^{\mathcal{B}} = \sum_{i=1}^{m^{\mathcal{G}}} \sum_{j \in [m_i^{\mathcal{B}}]} R_{i,j}^{\mathcal{B}}$. With $\delta = c_1 \max\{(1/T)^{1/(\beta+1)}, 1/T^{1/2}\}$ and $m^{\mathcal{G}} = C_3$, from the fact that $m_i^{\mathcal{B}}$'s are i.i.d. random variables with geometric distribution with $\mathbb{E}[m_i^{\mathcal{B}}] = (1/C(2\delta)^{\beta}) - 1$ for some constant $C > 0$, we have

$$\mathbb{E}[R_{m^{\mathcal{G}}}^{\mathcal{B}}] = O\left(\mathbb{E}\left[\sum_{i=1}^{m^{\mathcal{G}}} \sum_{j \in [m_i^{\mathcal{B}}]} R_{i,j}^{\mathcal{B}}\right]\right)$$
$$= \tilde{O}\left(\frac{1}{\delta^{\beta}} \max\{1, \delta^{\beta-1}\} + V_T/\delta^2\right)$$
$$= \tilde{O}\left(\max\{T^{\beta/(\beta+1)}, \sqrt{T}\}\right).$$

$\square$

From Lemma A.1, with $\delta = c_1 \max\{1/T^{\frac{1}{\beta+1}}, 1/\sqrt{T}\}$ we have
$$\mathbb{E}[R^{\mathcal{G}}(T)] = \tilde{O}\left(\max\{T^{\beta/(\beta+1)}, \sqrt{T}\}\right).$$

From $R^{\pi}(T) = R^{\mathcal{G}}(T) + R^{\mathcal{B}}(T)$ and Lemmas A.1, A.5, A.6 with $\delta = c_1 \max\{1/T^{\frac{1}{\beta+1}}, 1/\sqrt{T}\}$ we have

$$\mathbb{E}[R^{\pi}(T)] = \tilde{O}\left(\max\{T^{\beta/(\beta+1)}, \sqrt{T}\}\right). \tag{27}$$

**Conclusion:** Overall, from (26) and (27), we have

$$\mathbb{E}[R^\pi(T)] = \tilde{O}\left(\max\{V_T^{1/(\beta+2)}T^{(\beta+1)/(\beta+2)}, V_T^{1/3}T^{2/3}, T^{\beta/(\beta+1)}, \sqrt{T}\}\right).$$

## A.5 Proof of Theorem 3.3: Regret Upper Bound of Algorithm 1 for Abrupt Rotting ($S_T$)

Using the threshold parameter $\delta$ in the algorithm, we define an arm $a$ as a *good* arm if $\Delta_t(a) \leq \delta/2$, a *near-good* arm if $\delta/2 < \Delta_t(a) \leq 2\delta$, and otherwise, $a$ is a *bad* arm at time $t$. For analysis, *we consider abrupt change as sampling a new arm.* In other words, if a sudden change occurs to an arm $a$ by pulling the arm $a$, then the arm is considered to be two different arms; before and after the change. The type of abruptly rotted arms (good, near-good, or bad) after the change is determined by the current value of rotted mean reward. Without loss of generality, we assume that the policy samples arms, which are pulled at least once, in the sequence of $\bar{a}_1, \bar{a}_2, \ldots,$. Let $\mathcal{A}_T$ be the set of sampled arms, which are pulled at least once, over the horizon of $T$ time steps, which satisfies $|\mathcal{A}_T| \leq T$. We also define $\mathcal{A}_S$ as a set of arms that have been rotted and pulled at least once, which satisfies $|\mathcal{A}_S| \leq S_T$. To better understand the definitions, we provide an example. If an arm $a$ suffers abrupt rotting at first, then the arm $a$ is considered to be a different arm $a'$ after the rotting. If the arm $a'$ again suffers abrupt rotting, then it is considered to be $a''$ after the rotting. If arms $a, a', a''$ are pulled at least once, then $\{a, a', a''\} \in \mathcal{A}_T$ and $\{a', a''\} \in \mathcal{A}_S$ but $a \notin \mathcal{A}_S$. If arm $a''$ is not pulled at least once but $a$ and $a'$ are pulled at least once, then $\{a, a'\} \in \mathcal{A}_T$ and $a' \in \mathcal{A}_S$ but $a'' \notin \mathcal{A}_S$.

WLOG, the following proofs proceed under the given $\mathcal{A}_T$, since the proofs hold for any $\mathcal{A}_T$. Let $\bar{\mu}_{[t_1,t_2]}(a) = \sum_{t=t_1}^{t_2} \mu_t(a)\mathbb{1}(a_t = a)/n_{[t_1,t_2]}(a)$. We define the event $E_1 = \{|\widehat{\mu}_{[s_1,s_2]}(a) - \bar{\mu}_{[s_1,s_2]}(a)| \leq \sqrt{12\log(T)/n_{[s_1,s_2]}(a)}$ for all $1 \leq s_1 \leq s_2 \leq T, a \in \mathcal{A}_T\}$. By following the proof of Lemma 35 in Dylan J. Foster [12], from Lemma A.30 we have

$$P\left(\left|\widehat{\mu}_{[s_1,s_2]}(a) - \bar{\mu}_{[s_1,s_2]}(a)\right| \leq \sqrt{\frac{12\log T}{n_{[s_1,s_2]}(a)}}\right)$$
$$\leq \sum_{n=1}^{T} P\left(\left|\frac{1}{n}\sum_{i=1}^{n} X_i\right| \leq \sqrt{12\log(T)/n}\right)$$
$$\leq \frac{2}{T^5}, \tag{28}$$

where $X_i = r_{\tau_i} - \mu_{\tau_i}(a)$ and $\tau_i$ is the $i$-th time that the policy pulls arm $a$ starting from $s_1$. We note that even though $X_i$'s seem to depend on each other from $\tau_i$'s, each value of $X_i$ is independent of each other. Then using union bound for $s_1$, $s_2$, and $a \in \mathcal{A}_T$, we have

$$\mathbb{P}(E_1^c) \leq 2/T^2.$$

Let $t(s)$ be the time when $s$-th abrupt rotting occurs with $\rho_{t(s)}$ for $s \in [S_T]$. Then we have $\Delta_t(a) = O(1 + \sum_{s=1}^{S_T} \rho_{t(s)}) = O(1 + V_T)$ for any $a$ and $t$, which implies $\mathbb{E}[R^\pi(T)|E_1^c] = O(T + TV_T)$. For the case that $E_1$ does not hold, the regret is $\mathbb{E}[R^\pi(T)|E_1^c]\mathbb{P}(E_1^c) = O((1 + V_T)/T)$, which is negligible comparing with the regret when $E_1$ holds true which we show later. Therefore, in the rest of the proof we assume that $E_1$ holds true.

Recall that $R^\pi(T) = \sum_{t=1}^{T}(1 - \mu_t(a_t))$. For regret analysis, we divide $R^\pi(T)$ into two parts, $R^\mathcal{G}(T)$ and $R^\mathcal{B}(T)$ corresponding to regret of good or near-good arms, and bad arms over time $T$, respectively, such that $R^\pi(T) = R^\mathcal{G}(T) + R^\mathcal{B}(T)$. Recall that we consider abrupt change as sampling a new arm in this analysis. Then, from $\Delta_t(a) \leq 2\delta$ for any good or near-good arms $a$ at time $t$, we can easily obtain that

$$\mathbb{E}[R^\mathcal{G}(T)] = O(\delta T) = O(\max\{S_T^{1/(\beta+1)}T^{\beta/(\beta+1)}, \sqrt{S_T T}\}). \tag{29}$$

Now we analyze $R^\mathcal{B}(T)$. We divide regret $R^\mathcal{B}(T)$ into two regret from bad arms in $\mathcal{A}_T/\mathcal{A}_S$, denoted by $R^{\mathcal{B},1}(T)$, and regret from bad arms in $\mathcal{A}_S$, denoted by $R^{\mathcal{B},2}(T)$ such that $R^\mathcal{B}(T) = R^{\mathcal{B},1}(T) + R^{\mathcal{B},2}(T)$. We denote bad arms in $\mathcal{A}_S$ by $\mathcal{A}_S^\mathcal{B}$. We first analyze $R^{\mathcal{B},1}(T)$ in the following. For regret analysis, we adopt the episodic approach suggested in Kim et al. [13]. The main difference lies in analyzing our adaptive window UCB and a more generalized mean-reward distribution with

$\beta$. In the following, we introduce some notation. *Here we only consider arms in $\mathcal{A}_T/\mathcal{A}_S$ so that the following notation is defined without considering (rotted) arms in $\mathcal{A}_S$.* We note that from the definition of $\mathcal{A}_T$, arms $a$ before having undergone rotting are contained in $\mathcal{A}_T/\mathcal{A}_S$. Here we consider the case of $2\delta_S(\beta) < 1$ since otherwise when $2\delta_S(\beta) \geq 1$, bad arms are not defined in $\mathcal{A}_T/\mathcal{A}_S$.

Given a policy sampling arms in the sequence order, let $m^{\mathcal{G}}$ be the number of samples of distinct good arms and $m_i^{\mathcal{B}}$ be the number of consecutive samples of distinct bad arms between the $i-1$-st and $i$-th sample of a good arm among $m^{\mathcal{G}}$ good arms. We refer to the period starting from sampling the $i-1$-st good arm before sampling the $i$-th good arm as the $i$-th *episode*. Observe that $m_1^{\mathcal{B}}, \ldots, m_{m^{\mathcal{G}}}^{\mathcal{B}}$ are i.i.d. random variables with geometric distribution with parameter $C(2\delta)^\beta$ for some constant $C > 0$, given a fixed value of $m^{\mathcal{G}}$. Therefore, for non-negative integer $k$ we have $\mathbb{P}(m_i^{\mathcal{B}} = k) = (1 - C(2\delta)^\beta)^k C(2\delta)^\beta$, for $i = 1, \ldots, m^{\mathcal{G}}$.

Define $\tilde{m}_T^{\mathcal{G}}$ to be the total number of samples of a good arm by the policy $\pi$ over the horizon $T$ and $\tilde{m}_{i,T}^{\mathcal{B}}$ to be the number of samples of a bad arm in the $i$-th episode by the policy $\pi$ over the horizon $T$. For $i \in [\tilde{m}_T^{\mathcal{G}}]$, $j \in [\tilde{m}_{i,T}^{\mathcal{B}}]$, let $\tilde{n}_i^{\mathcal{G}}$ be the number of pulls of the good arm in the $i$-th episode and $\tilde{n}_{i,j}^{\mathcal{B}}$ be the number of pulls of the $j$-th bad arm in the $i$-th episode by the policy $\pi$ over the horizon $T$. Let $\tilde{a}$ be the last sampled arm over time horizon $T$ by $\pi$.

With a slight abuse of notation, we use $\pi$ for a modified strategy after $T$. Under a policy $\pi$, let $R_{i,j}^{\mathcal{B}}$ be the regret (summation of mean reward gaps) contributed by pulling the $j$-th bad arm in the $i$-th episode. Then let $R_{m^{\mathcal{G}}}^{\mathcal{B}} = \sum_{i=1}^{m^{\mathcal{G}}} \sum_{j \in [m_i^{\mathcal{B}}]} R_{i,j}^{\mathcal{B}}$, which is the regret from initially bad arms over the period of $m^{\mathcal{G}}$ episodes. For getting $R_{m^{\mathcal{G}}}^{\mathcal{B}}$, here we define the policy $\pi$ after $T$ such that it pulls $T$ amounts for a good arm and zero for a bad arm. After $T$ we can assume that there are no abrupt changes. For the last arm $\tilde{a}$ over the horizon $T$, it pulls the arm up to $T$ amounts if $\tilde{a}$ is a good arm and $\tilde{n}_{\tilde{m}_T^{\mathcal{G}}}^{\mathcal{G}} < T$. For $i \in [m^{\mathcal{G}}]$, $j \in [m_i^{\mathcal{B}}]$ let $n_i^{\mathcal{G}}$ and $n_{i,j}^{\mathcal{B}}$ be the number of pulling the good arm in $i$-th episode and $j$-th bad arm in $i$-th episode under $\pi$, respectively. Here we define $n_i^{\mathcal{G}}$'s and $n_{i,j}^{\mathcal{B}}$'s as follows:

If $\tilde{a}$ is a good arm,

$$n_i^{\mathcal{G}} = \begin{cases} \tilde{n}_i^{\mathcal{G}} & \text{for } i \in [\tilde{m}_T^{\mathcal{G}} - 1] \\ T & \text{for } i = \tilde{m}_T^{\mathcal{G}} \\ 0 & \text{for } i \in [m^{\mathcal{G}}]/[\tilde{m}_T^{\mathcal{G}}] \end{cases}, n_{i,j}^{\mathcal{B}} = \begin{cases} \tilde{n}_{i,j}^{\mathcal{B}} & \text{for } i \in [\tilde{m}_T^{\mathcal{G}}], j \in [\tilde{m}_{i,T}^{\mathcal{B}}] \\ 0 & \text{for } i \in [m^{\mathcal{G}}]/[\tilde{m}_T^{\mathcal{G}}], j \in [m_i^{\mathcal{B}}]/[\tilde{m}_{i,T}^{\mathcal{B}}]. \end{cases}$$

Otherwise,

$$n_i^{\mathcal{G}} = \begin{cases} \tilde{n}_i^{\mathcal{G}} & \text{for } i \in [\tilde{m}_T^{\mathcal{G}}] \\ T & \text{for } i = \tilde{m}_T^{\mathcal{G}} + 1 \\ 0 & \text{for } i \in [m^{\mathcal{G}}]/[\tilde{m}_T^{\mathcal{G}} + 1] \end{cases}, n_{i,j}^{\mathcal{B}} = \begin{cases} \tilde{n}_{i,j}^{\mathcal{B}} & \text{for } i \in [\tilde{m}_T^{\mathcal{G}}], j \in [\tilde{m}_{i,T}^{\mathcal{B}}] \\ 0 & \text{for } i \in [m^{\mathcal{G}}]/[\tilde{m}_T^{\mathcal{G}} - 1], j \in [m_i^{\mathcal{B}}]/[\tilde{m}_{i,T}^{\mathcal{B}}]. \end{cases}$$

Using the above notation and newly defined $\pi$ after $T$, we show that if $m^{\mathcal{G}} = S_T + 1$, then $R^{\mathcal{B}}(T) \leq R_{m^{\mathcal{G}}}^{\mathcal{B}}$ in the following.

**Lemma A.7.** *Under $E_1$, when $m^{\mathcal{G}} = S_T$ we have*

$$R^{\mathcal{B},1}(T) \leq R_{m^{\mathcal{G}}}^{\mathcal{B}}.$$

*Proof.* There are $S_T - 1$ number of abrupt changes over $T$. We consider two cases; there are $S_T$ abrupt changes before sampling $S_T$-th good arm or there are not. For the former case, if $\pi$ samples the $S_T$-th good arm and there are $S_T - 1$ number of abrupt changes before sampling the good arm, then it continues to pull the good arm until $T$. This is because when the algorithm samples a good arm $a$ at time $t'$, from $E_1$ and the stationary period, we have

$$\hat{\mu}_{[t',t]}(a) + \sqrt{12 \log(T)/n_{[t',t]}(a)} \geq \mu_{t'}(a) \geq 1 - \delta.$$

This implies that from the threshold condition, the algorithm does not stop pulling the good arm $a$. After $T$, from the definition of $\pi$ for the case when $\tilde{a}$ is a good arm, $n_{\tilde{m}_T^{\mathcal{G}}}^{\mathcal{G}} = T$. Therefore, the algorithm pulls the good arm for $T$ rounds.

Now we consider the latter case, such that $\pi$ samples the $S_T$-th good arm before the $S_T - 1$-st abrupt change over $T$. Before sampling the $S_T$-th good arm, there must exist two consecutive good arms such that there is no abrupt change between the two sampled good arms. This is a contraction because $\pi$ must pull the first good arm among the two up to $T$ under $E_1$ and $S_T - 1$-st abrupt change must occur after $T$.

Therefore, it is enough to consider the former case. When $m^{\mathcal{G}} = S_T$, we have

$$\sum_{i \in [m^{\mathcal{G}}]} n_i^{\mathcal{G}} \geq T,$$

which implies $R^{\mathcal{B},1}(T) \leq R_{m^{\mathcal{G}}}^{\mathcal{B}}$.  $\square$

From the above lemma, we set $m^{\mathcal{G}} = S_T$. We analyze $R_{m^{\mathcal{G}}}^{\mathcal{B}}$ to get a bound for $R^{\mathcal{B},1}(T)$ in the following lemma.

**Lemma A.8.** *Under $E_1$ and policy $\pi$, we have*

$$\mathbb{E}\left[R_{m^{\mathcal{G}}}^{\mathcal{B}}\right] = \tilde{O}\left(\max\{S_T^{1/(\beta+1)}T^{\beta/(\beta+1)}, \sqrt{S_T T}\}\right).$$

*Proof.* Recall that we consider arms in $\mathcal{A}_T/\mathcal{A}_S$. Let $a(i,j)$ be a sampled arm for $j$-th bad arm in the $i$-th episode and $\tilde{m}_T$ be the number of episodes from the policy $\pi$ over the horizon $T$. Suppose that the algorithm samples arm $a(i,j)$ at time $t_1(a(i,j))$. Then the algorithm stops pulling arm $a(i,j)$ at time $t_2(a(i,j)) + 1$ if $\hat{\mu}_{[s,t_2(a(i,j))]}(a) + \sqrt{12\log(T)/n_{[s,t_2(a(i,j))]}(a)} < 1 - \delta$ for some $s$ such that $t_1(a(i,j)) \leq s \leq t_2(a(i,j))$ and $s = t_2(a(i,j)) + 1 - 2^{l-1}$ for $l \in \mathbb{Z}^+$. For simplicity, we use $t_1$ and $t_2$ instead of $t_1(a(i,j))$ and $t_2(a(i,j))$ when there is no confusion. For the regret analysis, we consider that for $t > t_2$, arm $a$ is virtually pulled. With $E_1$, we assume that $\tilde{t}_2(\geq t_2)$ is the smallest time that there exists $t_1 \leq s \leq \tilde{t}_2$ with $s = \tilde{t}_2 + 1 - 2^{l-1}$ for $l \in \mathbb{Z}^+$ such that the following condition is met:

$$\mu_{t_1}(a(i,j)) + 2\sqrt{12\log(T)/n_{[s,\tilde{t}_2]}(a(i,j))} < 1 - \delta. \tag{30}$$

From the definition of $\tilde{t}_2$, we observe that for given $\tilde{t}_2$, the time step $s = s'$ satisfying (30) equals to $t_1$ (i.e. $s' = t_1$). Then, we can observe that $n_{[s',\tilde{t}_2]}(a(i,j)) = n_{[t_1,\tilde{t}_2]}(a(i,j)) = \lceil C_2\log(T)/(\Delta_{t_1}(a(i,j)) - \delta)^2 \rceil$ for some constant $C_2 > 0$, which satisfies (30). Then from $n_{[t_1,t_2]}(a(i,j)) \leq n_{[t_1,\tilde{t}_2]}(a(i,j))$, for all $i \in [\tilde{m}_T], j \in [\tilde{m}_{i,T}^{\mathcal{B}}]$ we have $n_{i,j}^{\mathcal{B}} = \tilde{O}(1/(\Delta_{t_1}(a(i,j)) - \delta)^2)$. We note that this bound for the number of pulling an arm holds for not only the case where the arm stops being pulled from the threshold condition but also the case where the arm stops being pulled from meeting an abrupt change (recall that abrupt changes are considered as sampling a new arm) or $T$. Then with the facts that $n_{i,j}^{\mathcal{B}} = 0$ for $i \in [m^{\mathcal{G}}]/[\tilde{m}_T], j \in [m_i^{\mathcal{B}}]/[\tilde{m}_{i,T}^{\mathcal{B}}]$, we have, for any $i \in [m^{\mathcal{G}}]$ and $j \in [m_i^{\mathcal{B}}]$,

$$n_{i,j}^{\mathcal{B}} = \tilde{O}(1/(\Delta_{t_1}(a(i,j)) - \delta)^2).$$

For $2\delta < x \leq 1$, let $b(x) = \mathbb{P}(\Delta_{t_1}(a) = x | a$ is a bad arm$)$. Then we have $\mathbb{P}(\Delta_{t_1}(a) = x | a$ is a bad arm$) = \mathbb{P}(\Delta_{t_1}(a) = x | \Delta_{t_1}(a) > 2\delta) = \mathbb{P}(\Delta_{t_1}(a) = x)/\mathbb{P}(\Delta_1(a) > 2\delta) = \mathbb{P}(\Delta_{t_1}(a) = x)/(1 - C(2\delta)^\beta) = O(\mathbb{P}(\Delta_{t_1}(a) = x))$, where the last equality comes from small $\delta$ with small enough $c_1 < 1$. For any $i \in [m^{\mathcal{G}}], j \in [m_i^{\mathcal{B}}]$, we have

$$\mathbb{E}[R_{i,j}^{\mathcal{B}}] \leq \mathbb{E}\left[\Delta_{t_1(a(i,j))}(a(i,j))n_{i,j}^{\mathcal{B}}\right]$$
$$= \tilde{O}\left(\int_{2\delta}^1 \frac{1}{(x-\delta)^2}xb(x)dx\right). \tag{31}$$

From the above results in (31),(19),(20),(21),(22),(23), for $\beta > 0$ we have

$$\mathbb{E}[R_{i,j}^{\mathcal{B}}] = \tilde{O}(\max\{1, \delta^{\beta-1}\}).$$

Recall that $R_{m^{\mathcal{G}}}^{\mathcal{B}} = \sum_{i=1}^{m^{\mathcal{G}}} \sum_{j \in [m_i^{\mathcal{B}}]} R_{i,j}^{\mathcal{B}}$. With $\delta = c_1 \max\{(S_T/T)^{1/(\beta+1)}, (S_T/T)^{1/2}\}$ and $m^{\mathcal{G}} = S_T$, from Lemma A.7 and the fact that $m_i^{\mathcal{B}}$'s are i.i.d. random variables following geometric

distribution with $\mathbb{E}[m_i^{\mathcal{B}}] = (1/C(2\delta)^\beta) - 1$ for some constant $C > 0$, we have

$$\mathbb{E}[R_{m^{\mathcal{G}}}^{\mathcal{B}}] = O\left(\mathbb{E}\left[\sum_{i=1}^{m^{\mathcal{G}}} \sum_{j \in [m_i^{\mathcal{B}}]} R_{i,j}^{\mathcal{B}}\right]\right)$$

$$= \tilde{O}\left(S_T \frac{1}{\delta^\beta} \max\{1, \delta^{\beta-1}\}\right)$$

$$= \tilde{O}\left(\max\{S_T^{1/(\beta+1)} T^{\beta/(\beta+1)}, \sqrt{S_T T}\}\right).$$

$\square$

From Lemma A.8, we have $\mathbb{E}[R^{\mathcal{B},1}(T)] = \mathbb{E}[R_{m^{\mathcal{G}}}^{\mathcal{B}}] = \tilde{O}\left(\max\{S_T^{1/(\beta+1)} T^{\beta/(\beta+1)}, \sqrt{S_T T}\}\right)$.

Now we analyze $R^{\mathcal{B},2}(T)$ in the following lemma. Here, we consider arms in $\mathcal{A}_S^{\mathcal{B}}$, which is allowed to have negative mean rewards.

**Lemma A.9.** *Under $E_1$ and policy $\pi$, we have*

$$\mathbb{E}\left[R^{\mathcal{B},2}(T)\right] = \tilde{O}\left(\max\{S_T/\delta, \bar{V}_T\}\right).$$

*Proof.* Recall that we consider arms $a \in \mathcal{A}_S^{\mathcal{B}}$ so that $\Delta_{t_1}(a) > 2\delta$ from definition. Suppose that the arm $a$ is sampled and pulled for the first time at time $t_1(a)$. Then the algorithm stops pulling arm $a$ at time $t_2(a) + 1$ if $\hat{\mu}_{[s,t_2(a)]}(a) + \sqrt{12\log(T)/n_{[s,t_2(a)]}(a)} < 1 - \delta$ for some $s$ such that $s \leq t_2(a)$ and $s = t_2(a) + 1 - 2^{l-1}$ for $l \in \mathbb{Z}^+$. For simplicity, we use $t_1$ and $t_2$ instead of $t_1(a)$ and $t_2(a)$ when there is no confusion. For regret analysis, we consider that for $t > t_2$, arm $a$ is virtually pulled. With $E_1$, we assume that $\tilde{t}_2(\geq t_2)$ is the smallest time that there exists $t_1 \leq s \leq \tilde{t}_2$ with $s = \tilde{t}_2 + 1 - 2^{l-1}$ for $l \in \mathbb{Z}^+$ such that the following condition is met:

$$\mu_{t_1}(a) + 2\sqrt{12\log(T)/n_{[s,\tilde{t}_2]}(a)} < 1 - \delta. \tag{32}$$

From the definition of $\tilde{t}_2$, we observe that for given $\tilde{t}_2$, the time step $s$, which satisfies (32), equals to $t_1$. Then, we can observe that $n_{[t_1,\tilde{t}_2]}(a) = \max\{\lceil C_2 \log(T)/(\Delta_{t_1}(a) - \delta)^2\rceil, 1\}$ for some constant $C_2 > 0$, which satisfies (32). From the above, for any $a \in \mathcal{A}_S^{\mathcal{B}}$ satifying $\Delta_{t_1}(a) \geq \sqrt{C_2 \log(T)} + \delta$, we have $n_{[t_1,\tilde{t}_2]}(a) = 1$. This implies that after pulling the arm $a$ once, the arm is eliminated and after that, the arm is not pulled anymore. Therefore, for any arm $a'$ which was rotted to $a$, we have $\Delta_{t_1(a')}(a') < \sqrt{C_2 \log(T)} + \delta$. This is because otherwise such that $\Delta_{t_1(a')}(a') \geq \sqrt{C_2 \log(T)} + \delta$, the arm $a'$ is eliminated and $a$ cannot be pulled which means $a \notin \mathcal{A}_S^{\mathcal{B}}$, which is a contradiction. Then for any arm $a \in \mathcal{A}_S^{\mathcal{B}}$, we have $\Delta_{t_1}(a) \leq \sqrt{C_2 \log(T)} + \delta + \rho_{t_1(a)-1}$. Recall that we consider abrupt rotting of an arm as sampling a new arm. Let $t(s)$ be the time step when the $s$-th abrupt rotting occurs. Then we note that $\rho_{t_1(a)-1} = \rho_{t(s)}$ when arm $a$ is a sampled arm from $s$-th abrupt rotting for $s \in [S_T]$.

From $n_{[t_1,t_2]}(a) \leq n_{[t_1,\tilde{t}_2]}(a)$, we have $n_{[t_1,t_2]}(a) = \tilde{O}(\max\{1/(\Delta_{t_1}(a) - \delta)^2, 1\})$. We note that this bound for number of pulling an arm holds for not only the case where the arm stops to be pulled from the threshold condition, but also the case where the arm stops to be pulled from meeting an abrupt change (recall that abrupt changes are considered as sampling a new arm) or $T$. From the definition of bad arms, we have $\Delta_{t_1}(a) \geq 2\delta$. Then the regret from arm $a$, denoted by $R(a)$, is bounded as follows: $R(a) = \Delta_{t_1}(a) n_{[t_1,t_2]}(a) = \tilde{O}(\max\{\Delta_{t_1}(a)/(\Delta_{t_1}(a) - \delta)^2, \Delta_{t_1}(a)\})$. Since $x/(x-\delta)^2 \leq 2/\delta$ for any $x \geq 2\delta$, we have $R(a) = \tilde{O}(\max\{1/\delta, \Delta_{t_1}(a)\})$. Therefore, with the fact that $\Delta_{t_1}(a) \leq \sqrt{C_2 \log(T)} + \delta + \rho_{t(s)}$ for the corresponding $s \in [S_T]$ such that $\rho_{t_1(a)-1} = \rho_{t(s)}$,

we have

$$\mathbb{E}\left[\sum_{a\in\mathcal{A}_S^B} R(a)\right] = \tilde{O}\left(\max\left\{S_T/\delta, \mathbb{E}\left[\sum_{a\in\mathcal{A}_S^\mathcal{B}} \Delta_{t_1}(a)\right]\right\}\right)$$

$$= \tilde{O}(\max\{S_T/\delta, S_T + \sum_{s=1}^{S_T}\mathbb{E}[\rho_{t(s)}]\})$$

$$= \tilde{O}(\max\{S_T/\delta, \sum_{s=1}^{S_T}\mathbb{E}[\rho_{t(s)}]\})$$

$$= \tilde{O}(\max\{S_T/\delta, \bar{V}_T\}),$$

where the second last equality comes from $S_T/\delta \geq S_T$. □

Finally, from $R^\pi(T) = R^\mathcal{G}(T) + R^\mathcal{B}(T)$, (29), and Lemmas A.8, A.9, we have

$$\mathbb{E}[R^\pi(T)] = \tilde{O}\left(\max\{S_T^{1/(\beta+1)}T^{\beta/(\beta+1)}, \sqrt{S_T T}, \bar{V}_T\}\right).$$

## A.6   Details for the Case of Unknown Parameters

---

**Algorithm 2** Adaptive UCB-Threshold with Adaptive Sliding Window

---

Given: $T, H, \mathcal{B}, \mathcal{A}, \alpha, \kappa, C$
Initialize: $\mathcal{A}' \leftarrow \mathcal{A}, w(\delta') \leftarrow 1$ for $\delta' \in \mathcal{B}$
**for** $i = 1, 2, \ldots, \lceil T/H \rceil$ **do**
  $t' \leftarrow (i-1)H + 1$
  Select a new arm $a \in \mathcal{A}'$
  Pull arm $a$ and get reward $r_{(i-1)H+1}$
  $p(\delta') \leftarrow (1-\alpha)\frac{w(\delta')}{\sum_{k\in\mathcal{B}} w(k)} + \alpha\frac{1}{B}$ for $\delta' \in \mathcal{B}$
  Select $\delta \leftarrow \delta'$ with probability $p(\delta')$ for $\delta' \in \mathcal{B}$
  **for** $t = (i-1)H + 2, \ldots, i\cdot H \wedge T$ **do**
    **if** $\min_{s\in\mathcal{T}_t(a)} WUCB(a, s, t-1, H) < 1 - \delta$ **then**
      $\mathcal{A}' \leftarrow \mathcal{A}'/\{a\}$
      Select a new arm $a \in \mathcal{A}'$
      Pull arm $a$ and get reward $r_t$
      $t' \leftarrow t$
    **else**
      Pull arm $a$ and get reward $r_t$
    **end if**
  **end for**
  $w(\delta) \leftarrow w(\delta)\exp\left(\frac{\alpha}{Bp(\delta)}\left(\frac{1}{2} + \frac{\sum_{t=(i-1)H}^{i\cdot H \wedge T} r_t}{CH\log(H)+4\sqrt{H\log T}}\right)\right)$
**end for**

---

Here, we consider the case where there are constraints for both $S_T$ and $V_T$, and parameters of $V_T$, $S_T$, and $\beta$ are unknown to the agent. We note that $\bar{V}_T \leq V_T$ from the constraint. The parameters of $\beta$, $V_T$, and $S_T$ are used to set the optimal threshold parameter $\delta$ in Algorithm 1. Therefore, when the parameters are not given, the procedure to find the optimal value $\delta$ is required. We adopt the Bandit-over-Bandit (BoB) approach in Cheung et al. [11], Kim et al. [13] by additionally considering adaptive window. In Algorithm 2, the algorithm consists of a master and several base algorithms with $\mathcal{B}$. For the master, we use EXP3 [2] to find a nearly best base in $\mathcal{B}$. Each base represents Algorithm 1 with a candidate threshold $\delta' \in \mathcal{B}$. The algorithm divides the time horizon into several blocks of length $H$. At each block, the algorithm samples a base in $\mathcal{B}$ from the EXP3 strategy and runs the base over the time steps of the block. Using the feedback from the block, the algorithm updates EXP3 and samples a new base for the next block. By block time passes, the master is likely to find an optimized $\delta$ in $\mathcal{B}$. Let $B = |\mathcal{B}|$. Then for Algorithm 2, we set $\alpha = \min\{1, \sqrt{B\log B/((e-1)\lceil T/H \rceil)}\}$ and $C > 0$ to be a large enough constant.

We define $\delta_V^\dagger = c_1 \max\{(V_T/T)^{1/(\beta+2)}, (V_T/T)^{1/3}, 1/H^{1/(\beta+1)}, 1/\sqrt{H}\}$ and $\delta_S^\dagger = c_1 \max\{(S_T/T)^{1/(\beta+1)}, (S_T/T)^{1/2}, 1/H^{1/(\beta+1)}, 1/\sqrt{H}\}$ for some constant $0 < c_1 < 1$. Then the optimized threshold parameter is $\delta_{VS}^\dagger = \min\{\delta_S^\dagger, \delta_V^\dagger\}$. The optimized threshold parameter can be derived from the theoretical analysis in Appendix A.7. The target of the master is to find the parameter. From the above, we can observe that $c_1/\sqrt{H} \leq \delta_{VS}^\dagger \leq 1$. Therefore, we set $\mathcal{B} = \{1/2, \ldots, 1/2^{\log_2 \sqrt{H}/c_1}\}$ which is the candidate values for unknown $\delta^\dagger$.

The regret is composed of two factors from the master and bases. To ensure that the regret bound from each base concerning $V_T$ and $S_T$ remains guaranteed irrespective of the bases chosen, we consider a constrained adaptive adversary. For the following, we consider that $\varrho_t$ for all $t > 0$ are arbitrarily determined before an algorithm is run, under the constraints of $V_T$ and $S_T$ where $\sum_{t=1}^{T-1} \varrho_t \leq V_T$ and $1 + \sum_{t=1}^{T-1} \mathbb{1}(\varrho_t \neq 0) \leq S_T$. Then under $\varrho_t$ for all $t > 0$, we consider the following adversary for rotting rates.

**Assumption A.10** (Constrained Adaptive Adversary). At each time $t > 0$, the value of rotting rate $\rho_t$ is determined arbitrarily immediately after the agent pulls an arm $a_t$ under the constraint of $0 \leq \rho_t \leq \varrho_t$ for given $\varrho_t$.

*Remark* A.11. Assumption A.10 is still more general than that for the maximum rotting rate constraint in Kim et al. [13] where $\varrho_t = \rho$ for all $t > 0$. We also observe that the constrained adaptive adversary in Assumption A.10 is milder than the adaptive adversary in Assumption 2.1. Additionally, we note that a special case of constraint $\rho_t = \varrho_t$ for all $t > 0$ in Assumption A.10 represents an oblivious adversary because $\varrho_t$'s are determined before an algorithm is run.

With a time block size of $H$ (where $H = \lceil\sqrt{T}\rceil$), the algorithm operates over $\lceil T/H \rceil$ blocks. Denote by $\mathcal{T}_i$ the set of time steps in the $i$-th block containing time steps of $(i-1)H + 1 \leq t \leq iH \wedge T$. It is possible to encounter large rotting for some $i$ block, potentially resulting in an arm's mean reward having a significantly low negative value, leading to suboptimal behavior by the master incurring large regret from the master. To address this, we introduce the assumption of equally distributed cumulative rotting for blocks, stated as follows:

**Assumption A.12.** $\sum_{t\in\mathcal{T}_i} \rho_t \leq H$ for all $i \in [\lceil T/H \rceil]$

*Remark* A.13. As similarly highlighted in Remark 2.5, this assumption is satisfied when mean rewards are under the constraint of $0 \leq \mu_t(a_t) \leq 1$ for all $t \in [T]$, which is frequently encountered in real-world applications where reward is represented by metrics like click rates or (normalized) ratings in content recommendation systems.

*Remark* A.14. Our rotting scenario with $\sum_{t\in\mathcal{T}_i} \rho_t \leq H$ for all $i \in [\lceil T/H \rceil]$ is more general in scope than the one with a maximum rotting rate constraint where $\rho_t \leq \rho = o(1)$ for all $t \in [T-1]$, which was explored in Kim et al. [13]. This is because for our setting, $\rho_t$ is not necessarily bounded by $o(1)$, and for the maximum rotting constraint setting with $\rho_t \leq \rho = o(1)$, the condition of $\sum_{t\in\mathcal{T}_i} \rho_t \leq H$ for all $i \in [\lceil T/H \rceil]$ is always satisfied. It is noteworthy that Assumption A.12 implies Assumption 2.4.

We provide a regret bound of Algorithm 2 under Assumption A.10 and Assumption A.12 in the following.

**Theorem A.15.** *Let $R_V'$ and $R_S'$ be defined as*

$$R_V' := \begin{cases} V_T^{\frac{1}{\beta+2}} T^{\frac{\beta+1}{\beta+2}} + T^{\frac{2\beta+1}{2\beta+2}} & for \ \beta \geq 1, \\ V_T^{\frac{1}{3}} T^{\frac{2}{3}} + T^{\frac{3}{4}} & for \ 0 < \beta < 1 \end{cases} \quad and \ R_S' := \begin{cases} \max\{S_T^{\frac{1}{\beta+1}} T^{\frac{\beta}{\beta+1}} + T^{\frac{2\beta+1}{2\beta+2}}, V_T\} & for \ \beta \geq 1, \\ \max\{\sqrt{S_T T} + T^{\frac{3}{4}}, V_T\} & for \ 0 < \beta < 1. \end{cases}$$

*Then, the policy $\pi$ of Algorithm 2 with $H = \lceil\sqrt{T}\rceil$ achieves the following regret bound:*

$$\mathbb{E}[R^\pi(T)] = \tilde{O}(\min\{R_V', R_S'\})$$

*Proof.* The proof is provided in Appendix A.7. $\square$

We can observe that there is the additional regret cost of $T^{(2\beta+1)/(2\beta+2)}$ for $\beta \geq 1$ or $T^{3/4}$ for $0 < \beta < 1$ compared to Algorithm 1. This additional cost originates from the additional procedure to learn the optimal value of $\delta$ in Algorithm 2, which is negligible when $V_T$ and $S_T$ are large enough.

*Remark* A.16. In the case where the value of $\beta$ is known, setting $H = \lceil \max\{T^{(\beta+1)/(\beta+3)}, \sqrt{T}\} \rceil$ reduces the additional regret cost of Algorithm 2 to $\max\{T^{(\beta+2)/(\beta+3)}, T^{3/4}\}$.

Attaining the optimal regret bound under a parameter-free algorithm remains an open problem.

## A.7 Proof of Theorem A.15: Regret Upper Bound of Algorithm 2

In the following, we deal with the cases of (a) $\delta_V^\dagger \leq \delta_S^\dagger$ so that $\delta_{VS}^\dagger = \delta_V^\dagger$ and (b) $\delta_V^\dagger > \delta_S^\dagger$ so that $\delta_{VS}^\dagger = \delta_S^\dagger$, separately.

### A.7.1 Case of $\delta_V^\dagger \leq \delta_S^\dagger$

Let $\pi_i(\delta')$ for $\delta' \in \mathcal{B}$ denote the base policy for time steps between $(i-1)H + 1$ and $i \cdot H \wedge T$ in Algorithm 2 using $1 - \delta'$ as a threshold. Denote by $a_t^{\pi_i(\delta')}$ the pulled arm at time step $t$ by policy $\pi_i(\delta')$. Then, for $\delta^\dagger \in \mathcal{B}$, which is set later for a near-optimal policy, we have

$$\mathbb{E}[R^\pi(T)] = \mathbb{E}\left[\sum_{t=1}^{T} 1 - \sum_{i=1}^{\lceil T/H \rceil} \sum_{t=(i-1)H+1}^{i \cdot H \wedge T} \mu_t(a_t^\pi)\right] = \mathbb{E}[R_1^\pi(T)] + \mathbb{E}[R_2^\pi(T)]. \quad (33)$$

where

$$R_1^\pi(T) = \sum_{t=1}^{T} 1 - \sum_{i=1}^{\lceil T/H \rceil} \sum_{t=(i-1)H+1}^{i \cdot H \wedge T} \mu_t(a_t^{\pi_i(\delta^\dagger)})$$

and

$$R_2^\pi(T) = \sum_{i=1}^{\lceil T/H \rceil} \sum_{t=(i-1)H+1}^{i \cdot H \wedge T} \mu_t(a_t^{\pi_i(\delta^\dagger)}) - \sum_{i=1}^{\lceil T/H \rceil} \sum_{t=(i-1)H+1}^{i \cdot H \wedge T} \mu_t(a_t^\pi).$$

Note that $R_1^\pi(T)$ accounts for the regret caused by the near-optimal base algorithm $\pi_i(\delta^\dagger)$'s against the optimal mean reward and $R_2^\pi(T)$ accounts for the regret caused by the master algorithm by selecting a base with $\delta \in \mathcal{B}$ at every block against the base with $\delta^\dagger$. In what follows, we provide upper bounds for each regret component. We first provide an upper bound for $\mathbb{E}[R_1^\pi(T)]$ by following the proof steps in Theorem 3.1. Then we provide an upper bound for $\mathbb{E}[R_2^\pi(T)]$. We set $H = \lceil T^{1/2} \rceil$ and $\delta^\dagger$ to be a smallest value in $\mathcal{B}$ which is larger than $\delta_V^\dagger = c_1 \max\{(V_T/T)^{1/(\beta+2)}, (V_T/T)^{1/3}, 1/H^{1/(\beta+1)}, 1/H^{1/2}\}$.

*Remark* A.17. One might wonder whether $R_1^\pi(T)$, regret from the near-optimal base of $\delta^\dagger$, satisfies the constraint of $V_T$ and $S_T$ even though the master may not select the near-optimal base in the algorithm for each block. From Assumption A.10, we can guarantee the constraints of $V_T$ and $S_T$ for rotting rates from each base regardless of the selected bases from the master for each block because the rotting upper bound $\varrho_t$'s are determined before staring the game regardless of the behavior of the master. Therefore, we can utilize $V_T$ and $S_T$ for bounding $R_1^\pi(T)$, which is the regret from the near-optimal base of $\delta^\dagger$.

**Upper Bounding $\mathbb{E}[R_1^\pi(T)]$.** We refer to the period starting from time step $(i-1)H + 1$ to time step $i \cdot H \wedge T$ as the $i$-th *block*. For any $i \in \lceil (T/H) - 1 \rceil$, policy $\pi_i(\delta^\dagger)$ runs over $H$ time steps independent to other blocks so that each block has the same expected regret and the last block has a smaller or equal expected regret than other blocks. Therefore, we focus on finding a bound on the regret from the first block equal to $\sum_{t=1}^{H} 1 - \mu_t(a_t^{\pi_1(\delta^\dagger)})$. We define an arm $a$ as a *good* arm if $\Delta(a) \leq \delta^\dagger/2$, a *near-good* arm if $\delta^\dagger/2 < \Delta(a) \leq 2\delta^\dagger$, and otherwise, $a$ is a *bad* arm. In $\mathcal{A}$, let $\bar{a}_1, \bar{a}_2, \ldots,$ be a sequence of arms, which have i.i.d. mean rewards following (1). Without loss of generality, we assume that the policy samples arms in the sequence of $\bar{a}_1, \bar{a}_2, \ldots,$.

Denote by $\mathcal{A}(i)$ the set of selected (explored) arms in the $i$-th block, which satisfies $|\mathcal{A}(i)| \leq H$. WLOG, we consider the case of given $\mathcal{A}(i)$ for the following because the proof can be applied to any given $\mathcal{A}(i)$. Let $\bar{\mu}_{[t_1, t_2]}(a) = \sum_{t=t_1}^{t_2} \mu_t(a)/n_{[t_1, t_2]}(a)$. We define the event $E_1 = \{|\widehat{\mu}_{[s_1, s_2]}(a) - \bar{\mu}_{[s_1, s_2]}(a)| \leq \sqrt{12 \log(H)/n_{[s_1, s_2]}(a)}$ for all $1 \leq s_1 \leq s_2 \leq H, a \in \mathcal{A}(i)\}$. As in (6), we have

$$\mathbb{P}(E_1^c) \leq 2/H^2.$$

We denote by $V_{H,i} = \sum_{t \in \mathcal{T}_i} \rho_t$ the cumulative amount of rotting in the time steps in the $i$-th block. From the cumulative amount of rotting, we note that $\Delta_t(a) = O(V_{H,i} + 1)$ for any $a$ and $t$ in $i$-th block, which implies $\mathbb{E}[R^\pi(T)|E_1^c] = O(H^2)$ from $V_{H,i} \leq H$ under Assumption A.12. For the case where $E_1$ does not hold, the regret is $\mathbb{E}[R^\pi(T)|E_1^c]\mathbb{P}(E_1^c) = O(1)$, which is negligible compared to the regret when $E_1$ holds, which we show later. For the case that $E_1$ does not hold, the regret is $\mathbb{E}[R^\pi(H)|E_1^c]\mathbb{P}(E_1^c) = O(1)$, which is negligible compared with the regret when $E_1$ holds true which we show later. Therefore, in the rest of the proof we assume that $E_1$ holds true.

In the following, we first provide a regret bound over the first block.

For regret analysis, we divide $R^{\pi_1(\delta^\dagger)}(H)$ into two parts, $R^{\mathcal{G}}(H)$ and $R^{\mathcal{B}}(H)$ corresponding to regret of good or near-good arms, and bad arms over time $H$, respectively, such that $R^{\pi_1(\delta^\dagger)}(H) = R^{\mathcal{G}}(H) + R^{\mathcal{B}}(H)$. We denote by $V_{H,i}$ the cumulative amount of rotting in the time steps in the $i$-th block. We first provide a bound of $R^{\mathcal{G}}(H)$ in the following lemma.

**Lemma A.18.** *Under $E_1$ and policy $\pi$, we have*

$$\mathbb{E}[R^{\mathcal{G}}(H)] = \tilde{O}\left(H\delta^\dagger + H^{2/3}\mathbb{E}[V_{H,1}^{1/3}]\right).$$

*Proof.* We can easily prove the theorem by following the proof steps in Lemma A.1 □

Now, we provide a regret bound for $R^{\mathcal{B}}(H)$. We note that the initially bad arms can be defined only when $2\delta^\dagger < 1$. Otherwise when $2\delta^\dagger \geq 1$, we have $R(T) = R^{\mathcal{G}}(T)$, which completes the proof. Therefore, for the regret from bad arms, we consider the case of $2\delta^\dagger < 1$. For the proof, we adopt the episodic approach in Kim et al. [13] for regret analysis.

Given a policy sampling arms in the sequence order, let $m^{\mathcal{G}}$ be the number of samples of distinct good arms and $m_i^{\mathcal{B}}$ be the number of consecutive samples of distinct bad arms between the $i-1$-st and $i$-th sample of a good arm among $m^{\mathcal{G}}$ good arms. We refer to the period starting from sampling the $i-1$-st good arm before sampling the $i$-th good arm as the $i$-th *episode*. Observe that $m_1^{\mathcal{B}}, \ldots, m_{m^{\mathcal{G}}}^{\mathcal{B}}$ are i.i.d. random variables with geometric distribution with parameter $2\delta$, given a fixed value of $m^{\mathcal{G}}$. Therefore, for non-negative integer $k$ we have $\mathbb{P}(m_i^{\mathcal{B}} = k) = (1 - C(2\delta^\dagger)^\beta)^k C(2\delta^\dagger)^\beta$ for some constant $C > 0$, for $i = 1, \ldots, m^{\mathcal{G}}$. Define $\tilde{m}_H$ to be the number of episodes from the policy $\pi$ over the horizon $H$, $\tilde{m}_H^{\mathcal{G}}$ to be the total number of samples of a good arm by the policy $\pi$ over the horizon $H$ such that $\tilde{m}_H^{\mathcal{G}} = \tilde{m}_H$ or $\tilde{m}_H^{\mathcal{G}} = \tilde{m} - 1$, and $\tilde{m}_{i,H}^{\mathcal{B}}$ to be the number of samples of a bad arm in the $i$-th episode by the policy $\pi_1(\delta^\dagger)$ over the horizon $H$.

Under a policy $\pi_1(\delta^\dagger)$, let $R_{i,j}^{\mathcal{B}}$ be the regret (summation of mean reward gaps) contributed by pulling the $j$-th bad arm in the $i$-th episode. Then let $R_{m^{\mathcal{G}}}^{\mathcal{B}} = \sum_{i=1}^{m^{\mathcal{G}}} \sum_{j \in [m_i^{\mathcal{B}}]} R_{i,j}^{\mathcal{B}}$, which is the regret from initially bad arms over the period of $m^{\mathcal{G}}$ episodes.

For obtaining a regret bound, we first focus on finding a required number of episodes, $m^{\mathcal{G}}$, such that $R^{\mathcal{B}}(T) \leq R_{m^{\mathcal{G}}}^{\mathcal{B}}$. Then we provide regret bounds for each bad arm and good arm in an episode. Lastly, we obtain a regret bound for $\mathbb{E}[R^{\mathcal{B}}(T)]$ using the episodic regret bound.

Let $a(i)$ be a good arm in the $i$-th episode and $a(i, j)$ be a $j$-th bad arm in the $i$-th episode. We define $V_H(a) = \sum_{t=1}^{H} \rho_t \mathbb{1}(a_t = a)$. Then excluding the last episode $\tilde{m}_H$ over $H$, we provide lower bounds of the total rotting variation over $H$ for $a(i)$, denoted by $V_H(a(i))$, in the following lemma.

**Lemma A.19.** *Under $E_1$, given $\tilde{m}_H$, for any $i \in [\tilde{m}_H^{\mathcal{G}}]/\{\tilde{m}_H\}$ we have*

$$V_H(a(i)) \geq \delta^\dagger/2.$$

*Proof.* We can easily prove the theorem by following the proof steps in Lemma A.2 □

**We first consider the case where $V_T > \max\{T/H^{3/2}, T/H^{(\beta+2)/(\beta+1)}\}$.** In this case, we have $\delta^\dagger = c_1 \max\{(V_T/T)^{1/(\beta+2)}, (V_T/T)^{1/3}\}$. Here, we define the policy $\pi$ after time $H$ such that it pulls a good arm until its total rotting variation is equal to or greater than $\delta^\dagger/2$ and does not pull a sampled bad arm. We note that defining how $\pi$ works after $H$ is only for the proof to get a regret

bound over time horizon $H$. For the last arm $\tilde{a}$ over the horizon $H$, it pulls the arm until its total variation becomes $\max\{\delta^\dagger/2, V_H(\tilde{a})\}$ if $\tilde{a}$ is a good arm. For $i \in [m^{\mathcal{G}}]$, $j \in [m_i^{\mathcal{B}}]$ let $V_i^{\mathcal{G}}$ and $V_{i,j}^{\mathcal{B}}$ be the total rotting variation of pulling the good arm in $i$-th episode and $j$-th bad arm in $i$-th episode from the policy, respectively. Here we define $V_i^{\mathcal{G}}$'s and $V_{i,j}^{\mathcal{B}}$'s as follows:

If $\tilde{a}$ is a good arm,

$$V_i^{\mathcal{G}} = \begin{cases} V_H(a(i)) & \text{for } i \in [\tilde{m}_H^{\mathcal{G}} - 1] \\ \max\{\delta^\dagger/2, V_H(a(i))\} & \text{for } i \in [m^{\mathcal{G}}]/[\tilde{m}_H^{\mathcal{G}} - 1] \end{cases}, V_{i,j}^{\mathcal{B}} = \begin{cases} V_H(a(i,j)) & \text{for } i \in [\tilde{m}_H^{\mathcal{G}}], j \in [\tilde{m}_{i,H}^{\mathcal{B}}] \\ 0 & \text{for } i \in [m^{\mathcal{G}}]/[\tilde{m}_H^{\mathcal{G}}], j \in [m_i^{\mathcal{B}}]. \end{cases}$$

Otherwise,

$$V_i^{\mathcal{G}} = \begin{cases} V_H(a(i)) & \text{for } i \in [\tilde{m}_H^{\mathcal{G}}] \\ \delta^\dagger/2 & \text{for } i \in [m^{\mathcal{G}}]/[\tilde{m}_H^{\mathcal{G}}] \end{cases}, V_{i,j}^{\mathcal{B}} = \begin{cases} V_H(a(i,j)) & \text{for } i \in [\tilde{m}_H^{\mathcal{G}}], j \in [\tilde{m}_{i,H}^{\mathcal{B}}] \\ 0 & \text{for } i \in [m^{\mathcal{G}}]/[\tilde{m}_H^{\mathcal{G}} - 1], j \in [m_i^{\mathcal{B}}]/[\tilde{m}_{i,H}^{\mathcal{B}}]. \end{cases}$$

For $i \in [m^{\mathcal{G}}]$, $j \in [m_i^{\mathcal{B}}]$ let $n_{i,j}^{\mathcal{B}}$ be the number of pulling the good arm in $i$-th episode and $j$-th bad arm in $i$-th episode from the policy, respectively. We define $n_H(a)$ be the total amount of pulling arm $a$ over $H$. Here we define $n_{i,j}^{\mathcal{B}}$'s as follows:

$$n_{i,j}^{\mathcal{B}} = \begin{cases} n_H(a(i,j)) & \text{for } i \in [\tilde{m}_H^{\mathcal{G}}], j \in [\tilde{m}_{i,H}^{\mathcal{B}}] \\ 0 & \text{for } i \in [m^{\mathcal{G}}]/[\tilde{m}_H^{\mathcal{G}}], j \in [m_i^{\mathcal{B}}]. \end{cases}$$

Then we provide $m^{\mathcal{G}}$ such that $R^{\mathcal{B}}(H) \leq R_{m^{\mathcal{G}}}^{\mathcal{B}}$ in the following lemma.

**Lemma A.20.** *Under $E_1$, when $m^{\mathcal{G}} = \lceil 2V_{H,1}/\delta^\dagger \rceil$ we have*

$$R^{\mathcal{B}}(H) \leq R_{m^{\mathcal{G}}}^{\mathcal{B}}.$$

*Proof.* We can easily show the theorem by following the proof steps of Lemma A.3 □

From the result of Lemma A.20, we set $m^{\mathcal{G}} = \lceil 2V_{H,1}/\delta^\dagger \rceil$. In the following, we anlayze $R_{m^{\mathcal{G}}}^{\mathcal{B}}$ for obtaining a regret bound for $R^{\mathcal{B}}(H)$.

**Lemma A.21.** *Under $E_1$ and policy $\pi$, we have*

$$\mathbb{E}[R_{m^{\mathcal{G}}}^{\mathcal{B}}] = \tilde{O}\left(\max\{V_{H,1}(T/V_T)^{(\beta+1)/(\beta+2)} + (T/V_T)^{\beta/(\beta+2)}, V_{H,1}(T/V_T)^{2/3} + (T/V_T)^{1/3}\}\right).$$

*Proof.* We can easily prove the theorem by following proof steps in Lemma A.4. From (24), for any $i \in [m^{\mathcal{G}}]$, $j \in [m_i^{\mathcal{B}}]$, we have

$$\mathbb{E}[R_{i,j}^{\mathcal{B}}] \leq \mathbb{E}\left[\Delta(a(i,j))n_{i,j}^{\mathcal{B}} + V_{i,j}^{\mathcal{B}}n_{i,j}^{\mathcal{B}}\right]$$
$$= \tilde{O}\left(\max\{1, (\delta^\dagger)^{\beta-1}\} + \mathbb{E}[V_{i,j}^{\mathcal{B}}]/(\delta^\dagger)^2\right).$$

Recall that $R_{m^{\mathcal{G}}}^{\mathcal{B}} = \sum_{i=1}^{m^{\mathcal{G}}} \sum_{j \in [m_i^{\mathcal{B}}]} R_{i,j}^{\mathcal{B}}$. With $\delta^\dagger = c_1 \max\{(V_T/T)^{1/(\beta+2)}, (V_T/T)^{1/3}\}$ and $m^{\mathcal{G}} = \lceil 2V_{H,1}/\delta^\dagger \rceil$, from the fact that $m_i^{\mathcal{B}}$'s are i.i.d. random variables with geometric distribution with $\mathbb{E}[m_i^{\mathcal{B}}] = 1/(2\delta^\dagger)^\beta - 1$, we have

$$\mathbb{E}[R_{m^{\mathcal{G}}}^{\mathcal{B}}] = O\left(\mathbb{E}\left[\sum_{i=1}^{m^{\mathcal{G}}} \sum_{j \in [m_i^{\mathcal{B}}]} R_{i,j}^{\mathcal{B}}\right]\right)$$

$$= \tilde{O}\left((\mathbb{E}[V_{H,1}]/\delta^\dagger + 1)\frac{1}{(\delta^\dagger)^\beta} \max\{1, (\delta^\dagger)^{\beta-1}\} + \mathbb{E}[V_{H,1}]/(\delta^\dagger)^2\right)$$

$$= \tilde{O}\left(\max\left\{\frac{\mathbb{E}[V_{H,1}]}{(\delta^\dagger)^{\beta+1}}, \frac{\mathbb{E}[V_{H,1}]}{(\delta^\dagger)^2}\right\} + \max\left\{\frac{1}{(\delta^\dagger)^\beta}, \frac{1}{\delta^\dagger}\right\}\right)$$

$$= \tilde{O}\left(\max\{\mathbb{E}[V_{H,1}](T/V_T)^{(\beta+1)/(\beta+2)} + (T/V_T)^{\beta/(\beta+2)}, \mathbb{E}[V_{H,1}](T/V_T)^{2/3} + (T/V_T)^{1/3}\}\right).$$

□

From $R^{\pi_1(\delta^\dagger)}(H) = R^{\mathcal{G}}(H) + R^{\mathcal{B}}(H)$ and Lemmas A.18, A.20, A.21, with $\delta^\dagger = \max\{(V_T/T)^{1/(\beta+2)}, (V_T/T)^{1/3}\}$ we have

$$
\begin{aligned}
&\mathbb{E}[R^{\pi_1(\delta^\dagger)}(H)] \\
&= \tilde{O}\left(\max\left\{\mathbb{E}[V_{H,1}](T/V_T)^{(\beta+1)/(\beta+2)} + H(V_T/T)^{1/(\beta+2)} + (T/V_T)^{\beta/(\beta+2)},\right.\right. \\
&\qquad\qquad \left.\left. \mathbb{E}[V_{H,1}](T/V_T)^{2/3} + H(V_T/T)^{1/3} + (T/V_T)^{1/3}\right\} + H^{2/3}\mathbb{E}[V_{H,1}^{1/3}]\right).
\end{aligned}
$$

The above regret bound is for the first block. Therefore, by summing regrets from $\lceil T/H \rceil$ number of blocks, from $V_T > \max\{T/H^{(\beta+2)/(\beta+1)}, T/H^{3/2}\}$, $H = \lceil T^{1/2} \rceil$ and the fact that $\mathbb{E}[\sum_{t=1}^{T-1} \rho_t] \le V_T$, using Hölder's inequality we have shown that

$$
\begin{aligned}
\mathbb{E}[R_1^\pi(T)] &= \tilde{O}\left(\max\{T^{(\beta+1)/(\beta+2)}V_T^{1/(\beta+2)}, T^{2/3}V_T^{1/3}\} + \frac{T}{H}\max\{(T/V_T)^{\beta/(\beta+2)}, (T/V_T)^{1/3}\}\right) \\
&= \tilde{O}\left(\max\{T^{(\beta+1)/(\beta+2)}V_T^{1/(\beta+2)}, T^{2/3}V_T^{1/3}\} + \max\{T^{(2\beta+1)/(2\beta+2)}, T^{3/4}\}\right).
\end{aligned}
$$
(34)

**Now, we consider the case where** $V_T \le \max\{T/H^{3/2}, T/H^{(\beta+2)/(\beta+1)}\}$**.** In this case, we have $\delta^\dagger = c_1 \max\{1/\sqrt{H}, 1/H^{\frac{1}{\beta+1}}\}$. From the result of Lemma A.20, by setting $m^{\mathcal{G}} = \lceil 2V_{H,1}/\delta^\dagger \rceil$ we have $R^{\mathcal{B}}(H) \le R_{m^{\mathcal{G}}}^{\mathcal{B}}$.

**Lemma A.22.** *Under $E_1$ and policy $\pi$, we have*

$$
\mathbb{E}[R_{m^{\mathcal{G}}}^{\mathcal{B}}] = \tilde{O}\left(\max\{V_{H,1}(T/V_T)^{(\beta+1)/(\beta+2)} + (T/V_T)^{\beta/(\beta+2)}, V_{H,1}(T/V_T)^{2/3} + (T/V_T)^{1/3}\}\right).
$$

*Proof.* We can easily prove the theorem by following proof steps in Lemma A.4. From (24), for any $i \in [m^{\mathcal{G}}]$, $j \in [m_i^{\mathcal{B}}]$, we have

$$
\begin{aligned}
\mathbb{E}[R_{i,j}^{\mathcal{B}}] &\le \mathbb{E}\left[\Delta(a(i,j))n_{i,j}^{\mathcal{B}} + V_{i,j}^{\mathcal{B}}n_{i,j}^{\mathcal{B}}\right] \\
&= \tilde{O}\left(\max\{1, \delta^{\beta-1}\} + \mathbb{E}[V_{i,j}^{\mathcal{B}}]/\delta^2\right).
\end{aligned}
$$

Recall that $R_{m^{\mathcal{G}}}^{\mathcal{B}} = \sum_{i=1}^{m^{\mathcal{G}}} \sum_{j \in [m_i^{\mathcal{B}}]} R_{i,j}^{\mathcal{B}}$. With $\delta^\dagger = c_1 \max\{1/H^{1/2}, 1/H^{1/(\beta+1)}\}$ and $m^{\mathcal{G}} = \lceil 2V_{H,1}/\delta^\dagger \rceil$, from the fact that $m_i^{\mathcal{B}}$'s are i.i.d. random variables with geometric distribution with $\mathbb{E}[m_i^{\mathcal{B}}] = (1/C(2\delta^\dagger)^\beta) - 1$, we have

$$
\begin{aligned}
\mathbb{E}[R_{m^{\mathcal{G}}}^{\mathcal{B}}] &= O\left(\mathbb{E}\left[\sum_{i=1}^{m^{\mathcal{G}}} \sum_{j \in [m_i^{\mathcal{B}}]} R_{i,j}^{\mathcal{B}}\right]\right) \\
&= \tilde{O}\left((\mathbb{E}[V_{H,1}]/\delta^\dagger + 1)\frac{1}{(\delta^\dagger)^\beta}\max\{1, (\delta^\dagger)^{\beta-1}\} + \mathbb{E}[V_{H,1}]/(\delta^\dagger)^2\right) \\
&= \tilde{O}\left(\max\left\{\frac{\mathbb{E}[V_{H,1}]}{(\delta^\dagger)^{\beta+1}}, \frac{\mathbb{E}[V_{H,1}]}{(\delta^\dagger)^2}\right\} + \max\left\{\frac{1}{(\delta^\dagger)^\beta}, \frac{1}{\delta^\dagger}\right\}\right) \\
&= \tilde{O}\left(\mathbb{E}[V_{H,1}]H + \max\{H^{\beta/(\beta+1)}, H^{1/2}\}\right).
\end{aligned}
$$

$\square$

From $R^{\pi_1(\delta^\dagger)}(H) = R^{\mathcal{G}}(H) + R^{\mathcal{B}}(H)$ and Lemmas A.18, A.20, A.22, with $\delta^\dagger = \Theta(\max\{1/H^{1/2}, 1/H^{1/(\beta+1)}\})$ we have

$$
\mathbb{E}[R^{\pi_1(\delta^\dagger)}(H)] = \tilde{O}\left(\max\{H^{\beta/(\beta+1)}, H^{1/2}\} + H^{2/3}\mathbb{E}[V_{H,1}^{1/3}] + \mathbb{E}[V_{H,1}]H\right).
$$

Therefore, by summing regrets from $\lceil T/H \rceil$ number of blocks and from $V_T = O(\max\{T/H^{3/2}, T/H^{(\beta+2)/(\beta+1)}\})$, $H = \lceil T^{1/2} \rceil$, and the fact that length of time steps in each block is bounded by $H$, we have

$$\mathbb{E}[R_1^\pi(T)] = \tilde{O}\left( \frac{T}{H} \max\{H^{\beta/(\beta+1)}, H^{1/2}\} + \sum_{i=1}^{\lceil T/H \rceil} H^{2/3}\mathbb{E}[V_{H,i}^{1/3}] + \sum_{i=1}^{\lceil T/H \rceil} \mathbb{E}[V_{H,i}]H \right)$$

$$= \tilde{O}\left( \frac{T}{H} \max\{H^{\beta/(\beta+1)}, H^{1/2}\} + T^{2/3}V_T^{1/3} + V_T H \right)$$

$$= \tilde{O}\left( \max\{T/H^{1/(\beta+1)}, T/H^{1/2}\} \right)$$

$$= \tilde{O}\left( \max\{T^{(2\beta+1)/(2\beta+2)}, T^{3/4}\} \right), \tag{35}$$

where the second equality comes from Hölder's inequality.

From (34) and (35), we have

$$\mathbb{E}[R_1^\pi(T)] = \tilde{O}(\max\{T^{(\beta+1)/(\beta+2)}V_T^{1/(\beta+2)} + T^{(2\beta+1)/(2\beta+2)}, T^{2/3}V_T^{1/3} + T^{3/4}\}). \tag{36}$$

**Upper Bounding** $\mathbb{E}[R_2^\pi(T)]$. We observe that the EXP3 is run for $\lceil T/H \rceil$ decision rounds and the number of policies (i.e. $\pi_i(\delta')$ for $\delta' \in \mathcal{B}$) is $B$. Denote the maximum absolute sum of rewards of any block with length $H$ by a random variable $Q'$. We first provide a bound for $Q'$ using concentration inequalities. For any block $i$, we have

$$\left| \sum_{t=(i-1)H+1}^{i \cdot H \wedge T} \mu_t(a_t^\pi) + \eta_t \right| \leq \left| \sum_{t=(i-1)H+1}^{i \cdot H \wedge T} \mu_t(a_t^\pi) \right| + \left| \sum_{t=(i-1)H+1}^{i \cdot H \wedge T} \eta_t \right|. \tag{37}$$

Denote by $\mathcal{T}_i$ the set of time steps in the $i$-th block. We define the event $E_2(i) = \{|\widehat{\mu}_{[s_1,s_2]}(a) - \overline{\mu}_{[s_1,s_2]}(a)| \leq \sqrt{14\log(H)/n_{[s_1,s_2]}(a)}$, for all $s_1, s_2 \in \mathcal{T}_i, s_1 \leq s_2, a \in \mathcal{A}(i)\}$ and $E_2 = \bigcap_{i \in [[T/H]]} E_2(i)$. From Lemma A.30, with $H = \lceil \sqrt{T} \rceil$ we have

$$\mathbb{P}(E_2^c) \leq \sum_{i \in [[T/H]]} \frac{2H^3}{H^6} \leq \frac{2}{T}.$$

By assuming that $E_2$ holds true, we can get a lower bound for $\mu_t(a_t^\pi)$, which may be a negative value from rotting, for getting an upper bound for $|\sum_{t=(i-1)H+1}^{i \cdot H \wedge T} \mu_t(a_t^\pi)|$. We can observe that $\sum_{t=(i-1)H+1}^{i \cdot H \wedge T} \mu_t(a_t^\pi) \leq H$. Therefore the remaining part is to get a lower bound for $\sum_{t=(i-1)H+1}^{i \cdot H \wedge T} \mu_t(a_t^\pi)$. For the proof simplicity, we consider that when an arm is rotted, then the arm is considered as a different arm after rotting. For instance, when arm $a$ is rotted at time $s$, then arm $a$ is considered as a different arm $a'$ after $s$. Therefore, each arm can be considered to be stationary. The set of arms is denoted by $\mathcal{L}$. We denote by $\mathcal{L}^+$ the set of arms having $\mu_t(a) \geq 0$ for $a \in \mathcal{L}$. We first focus on the arms in $\mathcal{L}/\mathcal{L}^+$.

Let $\delta_{\max}$ denote the maximum value in $\mathcal{B}$ so that $\delta_{\max} = 1/2$. With $E_2$ and $a \in \mathcal{L}/\mathcal{L}^+$, we assume that $\tilde{t}_2(\geq t_2)$ is the smallest time that there exists $t_1 \leq s \leq \tilde{t}_2$ with $s = \tilde{t}_2 + 1 - 2^{l-1}$ for $l \in \mathbb{Z}^+$ such that the following condition is met:

$$\mu_{t_1}(a) + \sqrt{12\log(H)/n_{[s,\tilde{t}_2]}(a)} + \sqrt{14\log(H)/n_{[s,\tilde{t}_2]}(a)} < 1 - \delta_{\max}. \tag{38}$$

From the definition of $\tilde{t}_2$, we observe that for given $\tilde{t}_2$, the time step $s$, which satisfies (38), equals to $t_1$. Then, we can observe that $n_{[t_1,\tilde{t}_2]}(a) = \max\{\lceil C_2 \log(H)/(\Delta_{t_1}(a) - \delta_{\max})^2 \rceil, 1\}$ for some constant $C_2 > 0$, which satisfies (38). From $n_{[t_1,t_2]}(a) \leq n_{[t_1,\tilde{t}_2]}(a)$, we have $n_{[t_1,t_2]}(a) \leq \max\{C_3 \log(H)/(\Delta_{t_1}(a) - \delta_{\max})^2, 1\}$ for some constant $C_3 > 0$. Then the regret from arm $a$, denoted by $R(a)$, is bounded as follows: $R(a) = \Delta_{t_1}(a)n_{[t_1,t_2]}(a) \leq \max\{C_3 \log(H)\Delta_{t_1}(a)/(\Delta_{t_1}(a) - \delta_{\max})^2, \Delta_{t_1}(a)\}$. Since $x/(x - \delta_{\max})^2 < 1/(1 - \delta_{\max})^2 = 4$ for any $x > 1$, we have $\Delta_{t_1}(a)/(\Delta_{t_1}(a) - \delta_{\max})^2 \leq 4$. Then we have $R(a) \leq$

$\max\{C_4 \log(H), \Delta_{t_1}(a)\}$ for some constant $C_4 > 0$. Then from $|\mathcal{L}| \leq H$, we have $\sum_{a \in \mathcal{L}/\{\mathcal{L}^+\}} R(a) \leq \max\{C_4 H \log(H), H + V_{H,i}\}$.

Since $\sum_{a \in \mathcal{L}^+} R(a) \leq H$, we have $\sum_{a \in \mathcal{L}} R(a) \leq H + \max\{C_4 H \log(H), H + V_{H,i}\}$. Therefore from $R(a) = \sum_{t=t_1(a)}^{t_2(a)} (1 - \mu_t(a))$, we have

$$\sum_{t=(i-1)H+1}^{iH \wedge T} \mu_t(a_t) \geq -\max\{C_4 H \log(H), H + V_{H,i}\},$$

which implies that from $V_{H,i} \leq H$ under Assumption A.12, for some $C_5 > 0$, we have

$$\left| \sum_{t=(i-1)H+1}^{i \cdot H \wedge T} \mu_t(a_t^\pi) \right| \leq \max\{C_4 H \log(H), H + V_{H,i}\} \leq C_5 H \log(H).$$

Next we provide a bound for $|\sum_{t=(i-1)H+1}^{i \cdot H \wedge T} \eta_t|$. We define the event $E_3(i) = \{|\sum_{t=(i-1)H+1}^{i \cdot H \wedge T} \eta_t| \leq 2\sqrt{H \log(T)}\}$ and $E_3 = \bigcap_{i \in [\lceil T/H \rceil]} E_3(i)$. From Lemma A.30, for any $i \in [\lceil T/H \rceil]$, we have

$$\mathbb{P}\left(E_3(i)^c\right) \leq \frac{2}{T^2}.$$

Then, under $E_2 \cap E_3$, with (37), we have

$$Q' \leq \max\{C_5 H \log H, H\} + 2\sqrt{H \log(T)} \leq C_5 H \log H + 2\sqrt{H \log(T)},$$

which implies $1/2 + \sum_{t=(i-1)H}^{i \cdot H \wedge T} r_t / (C_5 H \log H + 4\sqrt{H \log T}) \in [0, 1]$ or some large enough $C > 0$. With the rescaling and translation of rewards in Algorithm 2, from Corollary 3.2. in Auer et al. [2], we have

$$\mathbb{E}[R_2^\pi(T)|E_2 \cap E_3] = \tilde{O}\left((C_5 H \log H + 2\sqrt{H \log T})\sqrt{BT/H}\right) = \tilde{O}\left(\sqrt{HBT}\right). \tag{39}$$

*Remark* A.23. Regarding the utilization of the regret analysis of Corollary 3.2 (EXP3) in Auer et al. [2], we note that the reward for each base can be defined independently of the actual master's action. One might wonder whether the regret analysis for EXP3 can be utilized, considering the fact that the reward from a selected base may depend on the master's action due to the adaptive rotting rates. However, we highlight that the critical aspect of applying EXP3 analysis is whether the rewards from each base are defined independently of the actual action of the master, rather than whether the received (observed) reward from the selected base depends on the master's action. We can construct rewards for each base $\delta \in \mathcal{B}$ at time $t$ when a block starts, denoted as $x_t(\delta)$, as the reward obtained when the master selects base $\delta$ (even though base $\delta$ is not actually selected from the algorithm). Then, we can define $x_t(\delta)$ for each $\delta$ regardless of the master's actual action. In other words, irrespective of the actually selected base, we define $x_t(\delta)$ for all $\delta \in \mathcal{B}$ as the reward that the master can obtain by selecting $\delta$. In such a case, whatever the selected base by the master is at time $t$, $x_t(\delta)$'s remain the same, respectively. This construction is feasible because it's solely for analytical purposes and not necessary for the algorithm's functioning. With this construction of reward for each base, we can utilize EXP3 analysis to obtain a regret bound regarding the master ($R_2^\pi(T)$).

Note that the expected regret from EXP3 is trivially bounded by $o(H^2(T/H)) = o(TH)$ and $B = O(\log(T))$. Then, with (39), we have

$$\mathbb{E}[R_2^\pi(T)] = \mathbb{E}[R_2^\pi(T)|E_2 \cap E_3]\mathbb{P}(E_2 \cap E_3) + \mathbb{E}[R_2^\pi(T)|E_2^c \cup E_3^c]\mathbb{P}(E_2^c \cup E_3^c)$$
$$= \tilde{O}\left(\sqrt{HT}\right) + o(TH)(4/T^2)$$
$$= \tilde{O}\left(\sqrt{HT}\right). \tag{40}$$

Finally, from (33), (36), and (40), with $H = T^{1/2}$, we have

$$\mathbb{E}[R^\pi(T)] = \tilde{O}\left(\max\left\{V_T^{\frac{1}{\beta+2}} T^{\frac{\beta+1}{\beta+2}} + T^{\frac{2\beta+1}{2\beta+2}}, V_T^{\frac{1}{3}} T^{\frac{2}{3}} + T^{\frac{3}{4}}\right\}\right),$$

which concludes the proof.

### A.7.2 Case of $\delta_V^\dagger > \delta_S^\dagger$

Let $\pi_i(\delta')$ for $\delta' \in \mathcal{B}$ denote the base policy for time steps between $(i-1)H + 1$ and $i \cdot H \wedge T$ in Algorithm 2 using $1 - \delta'$ as a threshold. Denote by $a_t^{\pi_i(\delta')}$ the pulled arm at time step $t$ by policy $\pi_i(\delta')$. Then, for $\delta^\dagger \in \mathcal{B}$, which is set later for a near-optimal policy, we have

$$\mathbb{E}[R^\pi(T)] = \mathbb{E}\left[\sum_{t=1}^{T} 1 - \sum_{i=1}^{\lceil T/H \rceil} \sum_{t=(i-1)H+1}^{i \cdot H \wedge T} \mu_t(a_t^\pi)\right] = \mathbb{E}[R_1^\pi(T)] + \mathbb{E}[R_2^\pi(T)]. \qquad (41)$$

where

$$R_1^\pi(T) = \sum_{t=1}^{T} 1 - \sum_{i=1}^{\lceil T/H \rceil} \sum_{t=(i-1)H+1}^{i \cdot H \wedge T} \mu_t(a_t^{\pi_i(\delta^\dagger)})$$

and

$$R_2^\pi(T) = \sum_{i=1}^{\lceil T/H \rceil} \sum_{t=(i-1)H+1}^{i \cdot H \wedge T} \mu_t(a_t^{\pi_i(\delta^\dagger)}) - \sum_{i=1}^{\lceil T/H \rceil} \sum_{t=(i-1)H+1}^{i \cdot H \wedge T} \mu_t(a_t^\pi).$$

Note that $R_1^\pi(T)$ accounts for the regret caused by the near-optimal base algorithm $\pi_i(\delta^\dagger)$'s against the optimal mean reward and $R_2^\pi(T)$ accounts for the regret caused by the master algorithm by selecting a base with $\delta \in \mathcal{B}$ at every block against the base with $\delta^\dagger$. In what follows, we provide upper bounds for each regret component. We first provide an upper bound for $\mathbb{E}[R_1^\pi(T)]$ by following the proof steps in Theorem 3.3. Then we provide an upper bound for $\mathbb{E}[R_2^\pi(T)]$. We set $\delta^\dagger$ to be a smallest value in $\mathcal{B}$ which is larger than $\delta_S^\dagger = c_1 \max\{(S_T/T)^{1/(\beta+1)}, 1/H^{1/(\beta+1)}, (S_T/T)^{1/2}, 1/H^{1/2}\}$ such that we have $\delta^\dagger = \Theta(\max\{(S_T/T)^{1/(\beta+1)}, 1/H^{1/(\beta+1)}, (S_T/T)^{1/2}, 1/H^{1/2}\})$.

**Upper Bounding $\mathbb{E}[R_1^\pi(T)]$.** We refer to the period starting from time step $(i-1)H + 1$ to time step $i \cdot H \wedge T$ as the $i$-th *block*. For any $i \in \lceil T/H - 1 \rceil$, policy $\pi_i(\delta^\dagger)$ runs over $H$ time steps independent to other blocks so that each block has the same expected regret and the last block has a smaller or equal expected regret than other blocks. Therefore, we focus on finding a bound on the regret from the first block equal to $\sum_{t=1}^{H} 1 - \mu_t(a_t^{\pi_1(\delta^\dagger)})$. We define an arm $a$ as a *good* arm if $\Delta_t(a) \leq \delta^\dagger/2$, a *near-good* arm if $\delta^\dagger/2 < \Delta_t(a) \leq 2\delta^\dagger$, and otherwise, $a$ is a *bad* arm at time $t$. In $\mathcal{A}$, let $\bar{a}_1, \bar{a}_2, \ldots$, be a sequence of arms, which have i.i.d. mean rewards following (1). For analysis, *we consider abrupt change as sampling a new arm.* In other words, if a sudden change occurs to an arm $a$ by pulling the arm $a$, then the arm is considered to be two different arms; before and after the change. The type of abruptly rotted arms (good, near-good, or bad) after the change is determined by the rotted mean reward. Without loss of generality, we assume that the policy samples arms, which are pulled at least once, in the sequence of $\bar{a}_1, \bar{a}_2, \ldots,$.

Denote by $\mathcal{A}(i)$ the set of sampled arms, which are pulled at least once, in the $i$-th block, which satisfies $|\mathcal{A}(i)| \leq H$. We also define $\mathcal{A}_S(i)$ as a set of arms that have been rotted and pulled at least once in the $i$-th block, which satisfies $|\mathcal{A}_S(i)| \leq S_i$, where $S_i$ is defined as the number of abrupt changes in the $i$-th block. Let $\bar{\mu}_{[t_1, t_2]}(a) = \sum_{t=t_1}^{t_2} \mu_t(a)/n_{[t_1, t_2]}(a)$. We define the event $E_1 = \{|\hat{\mu}_{[s_1, s_2]}(a) - \bar{\mu}_{[s_1, s_2]}(a)| \leq \sqrt{12 \log(H)/n_{[s_1, s_2]}(a)}$ for all $1 \leq s_1 \leq s_2 \leq H, a \in \mathcal{A}(i)\}$. From Lemma A.30, as in (6), we have

$$\mathbb{P}(E_1^c) \leq 2/H^2.$$

For the case that $E_1$ does not hold, the regret is $\mathbb{E}[R^\pi(H)|E_1^c]\mathbb{P}(E_1^c) = O(1)$, which is negligible comparing with the regret when $E_1$ holds true which we show later. Therefore, in the rest of the proof we assume that $E_1$ holds true.

In the following, we first provide a regret bound over the first block.

For regret analysis, we divide $R_1^{\pi_1(\delta^\dagger)}(H)$ into two parts, $R^{\mathcal{G}}(H)$ and $R^{\mathcal{B}}(H)$ corresponding to regret of good or near-good arms, and bad arms over time $T$, respectively, such that $R_1^\pi(H) = R^{\mathcal{G}}(H) + R^{\mathcal{B}}(H)$. We can easily obtain that

$$\mathbb{E}[R^{\mathcal{G}}(H)] = O(\delta^\dagger H), \qquad (42)$$

from $\Delta(a) \le 2\delta^\dagger$ for any good or near-good arms $a$.

Now we analyze $R^\mathcal{B}(H)$. We divide regret $R^\mathcal{B}(H)$ into two regret from bad arms in $\mathcal{A}(1)/\mathcal{A}_S(1)$, denoted by $R^{\mathcal{B},1}(H)$, and regret from bad arms in $\mathcal{A}_S(1)$, denoted by $R^{\mathcal{B},2}(H)$ such that $R^\mathcal{B}(H) = R^{\mathcal{B},1}(H) + R^{\mathcal{B},2}(H)$. We first analyze $R^{\mathcal{B},1}(H)$ in the following. We consider arms in $\mathcal{A}(1)/\mathcal{A}_S(1)$. For the proof, we adopt the episodic approach in Kim et al. [13] for regret analysis. In the following, we introduce some notation. *Here we only consider arms in $\mathcal{A}(1)/\mathcal{A}_S(1)$ so that the following notation is defined without considering (rotted) arms in $\mathcal{A}_S(1)$.* Given a policy sampling arms in the sequence order, let $m^\mathcal{G}$ be the number of samples of distinct good arms and $m_i^\mathcal{B}$ be the number of consecutive samples of distinct bad arms between the $i-1$-st and $i$-th sample of a good arm among $m^\mathcal{G}$ good arms. We refer to the period starting from sampling the $i-1$-st good arm before sampling the $i$-th good arm as the $i$-th *episode*. Observe that $m_1^\mathcal{B}, \ldots, m_{m^\mathcal{G}}^\mathcal{B}$'s are i.i.d. random variables with geometric distribution with parameter $C(2\delta^\dagger)^\beta$ for some constant $C > 0$, conditional on the value of $m^\mathcal{G}$. Therefore, $\mathbb{P}(m_i^\mathcal{B} = k) = (1 - C(2\delta^\dagger)^\beta)^k C(2\delta^\dagger)^\beta$, for $i = 1, \ldots, m^\mathcal{G}$.

Define $\tilde{m}_H^\mathcal{G}$ to be the total number of samples of a good arm by the policy $\pi_1(\delta^\dagger)$ over the horizon $H$ and $\tilde{m}_{i,H}^\mathcal{B}$ to be the number of selections of a bad arm in the $i$-th episode by the policy $\pi$ over the horizon $H$. For $i \in [\tilde{m}_H^\mathcal{G}], j \in [\tilde{m}_{i,H}^\mathcal{B}]$, let $\tilde{n}_i^\mathcal{G}$ be the number of pulls of the good arm in the $i$-th episode and $\tilde{n}_{i,j}^\mathcal{B}$ be the number of pulls of the $j$-th bad arm in the $i$-th episode by the policy $\pi_1(\delta^\dagger)$ over the horizon $H$. Let $\tilde{a}$ be the last sampled arm over time horizon $H$ by $\pi_1(\delta^\dagger)$.

With a slight abuse of notation, we use $\pi_1(\delta^\dagger)$ for a modified strategy after $H$. Under a policy $\pi_1(\delta^\dagger)$, let $R_{i,j}^\mathcal{B}$ be the regret (summation of mean reward gaps) contributed by pulling the $j$-th bad arm in the $i$-th episode. Then let $R_{m^\mathcal{G}}^\mathcal{B} = \sum_{i=1}^{m^\mathcal{G}} \sum_{j \in [m_i^\mathcal{B}]} R_{i,j}^\mathcal{B}$, which is the regret from initially bad arms over the period of $m^\mathcal{G}$ episodes. For getting $R_{m^\mathcal{G}}^\mathcal{B}$, here we define the policy $\pi_1(\delta^\dagger)$ after $H$ such that it pulls $H$ amounts for a good arm and zero for a bad arm. After $H$ we can assume that there are no abrupt changes. For the last arm $\tilde{a}$ over the horizon $H$, it pulls the arm up to $H$ amounts if $\tilde{a}$ is a good arm and $\tilde{n}_{\tilde{m}_H^\mathcal{G}}^\mathcal{G} < H$. For $i \in [m^\mathcal{G}], j \in [m_i^\mathcal{B}]$ let $n_i^\mathcal{G}$ and $n_{i,j}^\mathcal{B}$ be the number of pulling the good arm in $i$-th episode and $j$-th bad arm in $i$-th episode under $\pi$, respectively. Here we define $n_i^\mathcal{G}$'s and $n_{i,j}^\mathcal{B}$'s as follows:

If $\tilde{a}$ is a good arm,

$$n_i^\mathcal{G} = \begin{cases} \tilde{n}_i^\mathcal{G} & \text{for } i \in [\tilde{m}_H^\mathcal{G} - 1] \\ H & \text{for } i = \tilde{m}_H^\mathcal{G} \\ 0 & \text{for } i \in [m^\mathcal{G}]/[\tilde{m}_H^\mathcal{G}] \end{cases}, n_{i,j}^\mathcal{B} = \begin{cases} \tilde{n}_{i,j}^\mathcal{B} & \text{for } i \in [\tilde{m}_H^\mathcal{G}], j \in [\tilde{m}_{i,H}^\mathcal{B}] \\ 0 & \text{for } i \in [m^\mathcal{G}]/[\tilde{m}_H^\mathcal{G}], j \in [m_i^\mathcal{B}]/[\tilde{m}_{i,H}^\mathcal{B}] \end{cases}.$$

Otherwise,

$$n_i^\mathcal{G} = \begin{cases} \tilde{n}_i^\mathcal{G} & \text{for } i \in [\tilde{m}_H^\mathcal{G}] \\ H & \text{for } i = \tilde{m}_H^\mathcal{G} + 1 \\ 0 & \text{for } i \in [m^\mathcal{G}]/[\tilde{m}_H^\mathcal{G} + 1] \end{cases}, n_{i,j}^\mathcal{B} = \begin{cases} \tilde{n}_{i,j}^\mathcal{B} & \text{for } i \in [\tilde{m}_H^\mathcal{G}], j \in [\tilde{m}_{i,H}^\mathcal{B}] \\ 0 & \text{for } i \in [m^\mathcal{G}]/[\tilde{m}_H^\mathcal{G} - 1], j \in [m_i^\mathcal{B}]/[\tilde{m}_{i,H}^\mathcal{B}] \end{cases}.$$

With a slight abuse of notation, we define $S_i$ to be the number of abrupt changes in $i$-th block. Then, we show that if $m^\mathcal{G} = S_1$, then $R^{\mathcal{B},1}(H) \le R_{m^\mathcal{G}}^\mathcal{B}$.

**Lemma A.24.** *Under $E_1$, when $m^\mathcal{G} = S_1$ we have*

$$R^{\mathcal{B},1}(H) \le R_{m^\mathcal{G}}^\mathcal{B}.$$

*Proof.* There are at most $S_1 - 1$ number of abrupt changes over the first block $H$. We consider two cases; there are $S_1 - 1$ abrupt changes before sampling $S_1$-th good arm or not. For the first case, if $\pi_1(\delta^\dagger)$ samples the $S_1$-th good arm and there are $S_1 - 1$ number of abrupt changes before sampling the good arm, then it continues to pull the good arm for $H$ rounds from $E_1$ and the definition of $\pi_1(\delta^\dagger)$ after $H$.

Now we consider the second case. If $\pi_1(\delta^\dagger)$ samples the $S_1$-th good arm before $T$ and there is at least one abrupt change after sampling the arm, then before sampling the $S_1$-th good arm, there must exist two consecutive good arms such that there is no abrupt change between sampling the two good arms.

This is a contraction because $\pi_1(\delta^\dagger)$ must pull the first good arm up to $H$ under $E_1$ and $S_1 - 1$-st abrupt change must occur after $H$.

Therefore, considering the first case, when $m^{\mathcal{G}} = S_1 + 1$, we have

$$\sum_{i \in [m^{\mathcal{G}}]} n_i^{\mathcal{G}} \geq H,$$

which implies $R^{\mathcal{B}}(H) \leq R^{\mathcal{B}}_{m^{\mathcal{G}}}$. □

From the above lemma, we set $m^{\mathcal{G}} = S_1$ and analyze $R^{\mathcal{B}}_{m^{\mathcal{G}}}$ to get a bound for $R^{\mathcal{B},1}(H)$ in the following lemma.

**Lemma A.25.** *Under $E_1$ and policy $\pi_1(\delta^\dagger)$, we have*

$$\mathbb{E}[R^{\mathcal{B}}_{m^{\mathcal{G}}}] = \tilde{O}\left(\mathbb{E}[S_1 \max\{1/(\delta^\dagger)^\beta, 1/\delta^\dagger\}]\right).$$

*Proof.* We can show this theorem by following the proof steps in Lemma A.8.

□

Now we analyze $R^{\mathcal{B},2}(H)$ in the following lemma. We denote by $V_H$ a cumulative amount of rotting rates in the first block.

**Lemma A.26.** *Under $E_1$ and policy $\pi$, we have*

$$\mathbb{E}\left[R^{\mathcal{B},2}(H)\right] = \tilde{O}\left(\mathbb{E}\left[\max\{S_1/\delta^\dagger, \sum_{s=1}^{S_1} \rho_{t(s)}\}\right]\right).$$

*Proof.* We can show this theorem by following the proof steps in Lemma A.9. □

From Lemmas A.24, A.25, A.26, we have

$$\mathbb{E}[R^{\mathcal{B}}(H)] = \mathbb{E}[R^{\mathcal{B},1}(H)] + \mathbb{E}[R^{\mathcal{B},2}(H)] = \tilde{O}\left(\mathbb{E}\left[S_1 \max\{1/(\delta^\dagger)^\beta, 1/\delta^\dagger\} + \sum_{s=1}^{S_1} \rho_{t(s)}\right]\right) \quad (43)$$

From $R_1^\pi(H) = R^{\mathcal{G}}(H) + R^{\mathcal{B}}(H)$, (42), and (43), we have

$$\mathbb{E}[R^{\pi_1(\delta^\dagger)}_{m^{\mathcal{G}}}] = \tilde{O}\left(H\delta^\dagger + \mathbb{E}\left[S_1 \max\{1/(\delta^\dagger)^\beta, 1/\delta^\dagger\} + \sum_{s=1}^{S_1} \rho_{t(s)}\right]\right).$$

The above regret is for the first block. Therefore, by summing regrets over $\lceil T/H \rceil$ number of blocks, we have shown that

$$\mathbb{E}[R_1^\pi(T)] = \tilde{O}(T\delta^\dagger + (T/H + S_T)\max\{1/(\delta^\dagger)^\beta, 1/\delta^\dagger\} + \sum_{s=1}^{S_T} \mathbb{E}[\rho_{t(s)}]). \quad (44)$$

**Upper bounding** $\mathbb{E}[R_2^\pi(T)]$. By following the proof steps in Theorem A.15, we have

$$\mathbb{E}[R_2^\pi(T)] = \tilde{O}\left(\sqrt{HT}\right). \quad (45)$$

Finally, from (41), (44), and (45), with the fact that $\sum_{s=1}^{S_T}\mathbb{E}[\rho_{t(s)}]\le V_T$, $H=T^{1/2}$, and $\delta^\dagger=\Theta(\max\{(S_T/T)^{1/(\beta+1)},1/H^{1/(\beta+1)},(S_T/T)^{1/2},1/H^{1/2}\})$, we have

$$\mathbb{E}[R^\pi(T)]=\tilde{O}\left(T\delta^\dagger+(T/H+S_T)\max\{1/(\delta^\dagger)^\beta,1/\delta^\dagger\}+\sqrt{HT}+\sum_{s=1}^{S_T}\mathbb{E}[\rho_{t(s)}]\right)$$

$$=\tilde{O}\left(T\delta^\dagger+\max\{T/H,S_T\}\max\{1/(\delta^\dagger)^\beta,1/\delta^\dagger\}+\sqrt{HT}+\sum_{s=1}^{S_T}\mathbb{E}[\rho_{t(s)}]\right)$$

$$=\tilde{O}\left(2T\delta^\dagger+\sqrt{HT}+\sum_{s=1}^{S_T}\mathbb{E}[\rho_{t(s)}]\right)$$

$$=\tilde{O}\left(\max\{S_T^{1/(\beta+1)}T^{\beta/(\beta+1)}+T^{(2\beta+1)/(2\beta+2)},\sqrt{S_T T}+T^{3/4},V_T\}\right),$$

which concludes the proof.

### A.8 Proof of Theorem 4.1: Regret Lower Bound for Slowly Rotting Rewards

We first consider the case when $V_T=\Theta(T)$. Recall that $\Delta_1(a)=1-\mu_1(a)$. Then for any randomly sampled $a\in\mathcal{A}$, we have $\mathbb{E}[\mu_1(a)]\ge y\mathbb{P}(\mu_1(a)\ge y)=y\mathbb{P}(\Delta_1(a)<1-y)$ for $y\in[0,1]$. Then with $y=1/2$, we have $\mathbb{E}[\mu_1(a)]\ge(1/2)\mathbb{P}(\Delta_1(a)<(1/2))=\Theta(1)$ from constant $\beta>0$ and (1). Then with $\mathbb{E}[\mu_1(a)]\le1$, we have $\mathbb{E}[\mu_1(a)]=\Theta(1)$. We then think of a policy $\pi'$ that randomly samples a new arm and pulls it once every round. Since $\mathbb{E}[\mu_1(a)]=\Theta(1)$ for any randomly sampled $a$, we have $\mathbb{E}[R^{\pi'}(T)]=\Theta(T)$. Next, we think of any policy $\pi''$ except $\pi'$. Then any policy $\pi''$ must pull an arm $a$ at least twice. Let $t'$ and $t''$ be the rounds when the policy pulls arm $a$. If we consider $\rho_{t'}=V_T$ then such policy has $\Omega(V_T)$ regret bound. Since $V_T=\Theta(T)$, any algorithm has $\Omega(T)$ in the worst case. Therefore we can conclude that any algorithm including $\pi'$ has a regret bound of $\Omega(T)$ in the worst case, which concludes the proof for $V_T=\Theta(T)$.

Now we think of the case where $V_T=o(T)$. For the lower bound, we adopt the proof methodology of Theorem 1 in Kim et al. [13] by making necessary adjustments to accommodate $V_T$ and $\beta$. We note that since higher $\beta$ implies a reduced chance of sampling a near-optimal arm, the criteria for defining the mean rewards of near-optimal arms becomes less stringent for higher $\beta$, which does not appear in the previous work. We first categorize arms as either bad or good according to their initial mean reward values. For the categorization, we utilize two thresholds in the proof as follows. Consider $0<\gamma<c<1$ for $\gamma$, which will be specified, and a constant $c$. Then the value of $1-\gamma$ represents a threshold value for identifying good arms, while $1-c$ serves as the threshold for identifying bad arms. We refer to arms $a$ satisfying $\mu_1(a)\le1-c$ as 'bad' arms and arms $a$ satisfying $\mu_1(a)>1-\gamma$ as 'good' arms. We also consider a sequence of arms in $\mathcal{A}$ denoted by $\bar{a}_1,\bar{a}_2,\ldots$. Given a policy $\pi$, without loss of generality, we can assume that $\pi$ selects arms according to the order of $\bar{a}_1,\bar{a}_2,\ldots$. For the rotting rates, we define $\varrho=V_T/(T-1)$. Then we consider $\rho_t=\varrho$ for all $t\in[T-1]$ so that $\sum_{t=1}^{T-1}\rho_t=V_T$.

**Case of $V_T=O(1/T^{1/(\beta+1)})$:** When $V_T=O(1/T^{1/(\beta+1)})$, the lower bound of order $T^{\frac{\beta}{\beta+1}}$ for the stationary case, from Theorem 3 in Wang et al. [24], is tight enough for the non-stationary case. From Theorem 3 in Wang et al. [24], we have

$$\mathbb{E}[R^\pi(T)]=\Omega(T^{\frac{\beta}{\beta+1}}). \tag{46}$$

We note that even though the mean rewards are rotting in our setting, Theorem 3 in Wang et al. [24] remains applicable without requiring any alterations in the proofs providing a tight regret bound for the near-stationary case. For the sake of completeness, we provide the proof of the theorem in the following. Let $K_1$ denote the number of bad arms $a$ that satisfy $\mu_1(a)\le1-c$ before sampling the first good arm, which satisfies $\mu_1(a)>1-\gamma$, in the sequence of arms $\bar{a}_1,\bar{a}_2,\ldots$. Let $\overline{\mu}$ be the initial mean reward of the best arm among the sampled arms by $\pi$ over time horizon $T$. Then for some $\kappa>0$, we have

$$R^\pi(T)=R^\pi(T)\mathbb{1}(\overline{\mu}\le1-\gamma)+R^\pi(T)\mathbb{1}(\overline{\mu}>1-\gamma)$$
$$\ge T\gamma\mathbb{1}(\overline{\mu}\le1-\gamma)+K_1c\mathbb{1}(\overline{\mu}>1-\gamma)$$
$$\ge T\gamma\mathbb{1}(\overline{\mu}\le1-\gamma)+\kappa c\mathbb{1}(\overline{\mu}>1-\gamma,K_1\ge\kappa). \tag{47}$$

By taking expectations on the both sides in (47) and setting $\kappa = T\gamma/c$, we have

$$\mathbb{E}[R^\pi(T)] \geq T\gamma\mathbb{P}(\overline{\mu} \leq 1 - \gamma) + \kappa c(\mathbb{P}(\overline{\mu} > 1 - \gamma) - \mathbb{P}(K_1 < \kappa)) = c\kappa\mathbb{P}(K_1 \geq \kappa).$$

We observe that $K_1$ follows a geometric distribution with success probability $\mathbb{P}(\mu_1(a) > 1 - \gamma)/p(\mu_1(a) \notin (1 - c, 1 - \gamma]) = \overline{\gamma} \leq C_1\gamma^\beta/(1 + C_2\gamma^\beta - C_3c^\beta)$ for some constants $C_1, C_2, C_3 > 0$ from (1), in which the success probability is the probability of sampling a good arm given that the arm is either a good or bad arm. Here we set a constant $0 < c < 1$ satisfying $1 - C_3c^\beta > 0$. Then by setting $\gamma = 1/T^{\frac{1}{\beta+1}}$ with $\kappa = T^{\frac{\beta}{\beta+1}}/c$, for some constant $C > 0$ we have

$$\mathbb{E}[R^\pi(T)] \geq c\kappa(1 - \overline{\gamma})^\kappa = \Omega\left(T^{\frac{\beta}{\beta+1}}(1 - C\gamma^\beta)^{T^{\frac{\beta}{\beta+1}}/c}\right) = \Omega(T^{\frac{\beta}{\beta+1}}),$$

where the last equality is obtained from $\log x \geq 1 - 1/x$ for all $x > 0$.

**Case of $V_T = \omega(1/T^{1/(\beta+1)})$ and $V_T = o(T)$:** When $V_T = \omega(1/T^{1/(\beta+1)})$, however, the lower bound of the stationary case is not tight enough. Here we provide the proof for the lower bound of $V_T^{1/(\beta+2)}T^{(\beta+1)/(\beta+2)}$ for the case of $V_T = \omega(1/T^{1/(\beta+1)})$. Let $K_m$ denote the number of "bad" arms $a$ that satisfy $\mu_1(a) \leq 1 - c$ before sampling $m$-th "good" arm, which satisfies $\mu_1(a) > 1 - \gamma$, in the sequence of arms $\overline{a}_1, \overline{a}_2, \dots$. Let $N_T$ be the number of sampled good arms $a$ such that $\mu_1(a) > 1 - \gamma$ until $T$.

We can decompose $R^\pi(T)$ into two parts as follows:

$$R^\pi(T) = R^\pi(T)\mathbb{1}(N_T < m) + R^\pi(T)\mathbb{1}(N_T \geq m). \tag{48}$$

We set $m = \lceil(1/2)T^{1/(\beta+2)}V_T^{(\beta+1)/(\beta+2)}\rceil$ and $\gamma = (V_T/T)^{1/(\beta+2)}$ with $V_T = o(T)$. For the first term in (48), $R^\pi(T)\mathbb{1}(N_T < m)$, we consider the fact that the minimal regret is obtained from the situation where there are $m - 1$ arms whose mean rewards are 1. In such a case, the optimal policy must sample the best $m - 1$ arms until their mean rewards become below the threshold $1 - \gamma$ (step 1) and then samples the best arm at each time for the remaining time steps (step 2). The number of times each arm needs to be pulled for the best $m - 1$ arms until their mean reward falls below $1 - \gamma$ is bounded from above by $\gamma/\rho + 1 = \gamma((T - 1)/V_T) + 1$. Therefore, the regret from step 2 is $R = \Omega((T - m\gamma(T/V_T))\gamma) = \Omega(T^{(\beta+1)/(\beta+2)}V_T^{1/(\beta+2)})$ in which the optimal policy pulls arms which mean rewards are below $1 - \gamma$ for the remaining time after step 1. Therefore, we have

$$R^\pi(T)\mathbb{1}(N_T < m) = \Omega(R\mathbb{1}(N_T < m)) = \Omega(T^{(\beta+1)/(\beta+2)}V_T^{1/(\beta+2)}\mathbb{1}(N_T < m)). \tag{49}$$

For getting a lower bound of the second term in (48), $R^\pi(T)\mathbb{1}(N_T \geq m)$, we use the minimum number of sampled arms $a$ that satisfy $\mu_1(a) \leq 1 - c$. When $N_T \geq m$ and $K_m \geq \kappa$, the policy samples at least $\kappa$ number of distinct arms $a$ satisfying $\mu_1(a) \leq 1 - c$ until $T$. Therefore, we have

$$R^\pi(T)\mathbb{1}(N_T \geq m) \geq c\kappa\mathbb{1}(N_T \geq m, K_m \geq \kappa). \tag{50}$$

We have $\overline{\gamma} = \Theta(\gamma^\beta)$ from (1) with constant $\beta > 0$. By setting $\kappa = m/\overline{\gamma} - m - \sqrt{m}/\overline{\gamma}$, with $V_T = o(T)$ and constant $\beta > 0$, we have

$$\kappa = \Theta(T^{(\beta+1)/(\beta+2)}V_T^{1/(\beta+2)}). \tag{51}$$

Then from (49), (50), and (51), we have

$$\mathbb{E}[R^\pi(T)] = \Omega(T^{(\beta+1)/(\beta+2)}V_T^{1/(\beta+2)}\mathbb{P}(N_T < m) + T^{(\beta+1)/(\beta+2)}V_T^{1/(\beta+2)}\mathbb{P}(N_T \geq m, K_m \geq \kappa))$$
$$\geq \Omega(T^{(\beta+1)/(\beta+2)}V_T^{1/(\beta+2)}\mathbb{P}(K_m \geq \kappa)). \tag{52}$$

Next we provide a lower bound for $\mathbb{P}(K_m \geq \kappa)$. Observe that $K_m$ follows a negative binomial distribution with $m$ successes and the success probability $\mathbb{P}(\mu_1(a) > 1 - \gamma)/\mathbb{P}(\mu_1(a) \notin (1 - c, 1 - \gamma]) = \overline{\gamma}$, in which the success probability is the probability of sampling a good arm given that the arm is either a good or bad arm. In the following lemma, we provide a concentration inequality for $K_m$.

**Lemma A.27.** *For any $1/2 + \overline{\gamma}/m < \alpha < 1$,*

$$\mathbb{P}(K_m \geq \alpha m(1/\overline{\gamma}) - m) \geq 1 - \exp(-(1/3)(1 - 1/\alpha)^2(\alpha m - \overline{\gamma})).$$

*Proof.* Let $X_i$ for $i > 0$ be i.i.d. Bernoulli random variables with success probability $\overline{\gamma}$. From Section 2 in Brown [9], we have

$$\mathbb{P}\left(K_m \leq \left\lfloor \alpha m \frac{1}{\overline{\gamma}} \right\rfloor - m\right) = \mathbb{P}\left(\sum_{i=1}^{\left\lfloor \alpha m \frac{1}{\overline{\gamma}} \right\rfloor} X_i \geq m\right). \tag{53}$$

From (53) and Lemma A.29, for any $1/2 + \overline{\gamma}/m < \alpha < 1$ we have

$$\mathbb{P}\left(K_m \leq \alpha m \frac{1}{\overline{\gamma}} - m\right) = \mathbb{P}\left(K_m \leq \left\lfloor \alpha m \frac{1}{\overline{\gamma}} \right\rfloor - m\right)$$

$$= \mathbb{P}\left(\sum_{i=1}^{\left\lfloor \alpha m \frac{1}{\overline{\gamma}} \right\rfloor} X_i \geq m\right)$$

$$\leq \exp\left(-\frac{(1 - 1/\alpha)^2}{3} \left\lfloor \alpha m \frac{1}{\overline{\gamma}} \right\rfloor \overline{\gamma}\right)$$

$$\leq \exp\left(-\frac{(1 - 1/\alpha)^2}{3} (\alpha m - \overline{\gamma})\right),$$

in which the first inequality comes from Lemma A.29, which concludes the proof. $\qquad\square$

From Lemma A.27 with $\alpha = 1 - 1/\sqrt{m}$ and large enough $T$, we have

$$\mathbb{P}(K_m \geq \kappa) \geq 1 - \exp\left(-\frac{1}{3}(m - \sqrt{m} - \overline{\gamma})\left(\frac{1}{\sqrt{m} - 1}\right)^2\right)$$

$$\geq 1 - \exp\left(-\frac{1}{6}(m - \sqrt{m})\left(\frac{1}{\sqrt{m} - 1}\right)^2\right)$$

$$= 1 - \exp\left(-\frac{1}{6}\frac{\sqrt{m}}{\sqrt{m} - 1}\right)$$

$$\geq 1 - \exp(-1/6). \tag{54}$$

Therefore, from (52) and (54), we have

$$\mathbb{E}[R^\pi(T)] = \Omega(T^{(\beta+1)/(\beta+2)} V_T^{1/(\beta+2)}). \tag{55}$$

Finally, from (46) and (55), we conclude that for any policy $\pi$, we have

$$\mathbb{E}[R^\pi(T)] = \Omega\left(\max\left\{T^{(\beta+1)/(\beta+2)} V_T^{1/(\beta+2)}, T^{\frac{\beta}{\beta+1}}\right\}\right).$$

### A.9 Proof of Theorem 4.2: Regret Lower Bound for Abruptly Rotting Rewards

First, we deal with the case when $S_T = 1$ or $S_T = \Theta(T)$. When $S_T = 1$ (implying $V_T = 0$), from the definition, the problem becomes stationary without rotting instances, which implies $\mathbb{E}[R^\pi(T)] = \Omega(\sqrt{T})$ from Theorem 3 in Wang et al. [24]. When $S_T = \Theta(T)$, we consider that rotting occurs for the first $S_T - 1$ rounds with $\rho_t = 1$ for all $t \in [S_T - 1]$. Then it is always beneficial to pull new arms every round until $S_T - 1$ rounds because the mean rewards of rotted arms are below 0 and those of non-rotted arms lie in $[0, 1]$. This means that any ideal policy samples a new arm and pulls it every round until $S_T - 1$. Then for any randomly sampled $a \in \mathcal{A}$, we have $\mathbb{E}[\mu_1(a)] \geq y\mathbb{P}(\mu_1(a) \geq y) = y\mathbb{P}(\Delta_1(a) < 1 - y)$ for $y \in [0, 1]$. Then with $y = 1/2$, we have $\mathbb{E}[\mu_1(a)] \geq (1/2)\mathbb{P}(\Delta_1(a) < (1/2)) = \Theta(1)$ from constant $\beta > 0$ and (1). Then with $\mathbb{E}[\mu_1(a)] \leq 1$, we have $\mathbb{E}[\mu_1(a)] = \Theta(1)$. Since $\mathbb{E}[\mu_1(a)] = \Theta(1)$ for any randomly sampled $a \in \mathcal{A}$, any ideal policy has $\mathbb{E}[R^\pi(T)] \geq \sum_{i=1}^{S_T} \mathbb{E}[\mu_1(a)] = \Omega(S_T) = \Omega(T)$, which concludes the proof for $S_T = \Theta(T)$.

Now we consider the case of $S_T = o(T)$ and $S_T \geq 2$. We initially provide a regret bound with respect to the cumulative rotting amount of $\bar{V}_T$. We first think of a policy $\pi$ that randomly samples a new arm and pulls it once every round. Then for any randomly sampled $a \in \mathcal{A}$, we have $\mathbb{E}[\mu_1(a)] = \Theta(1)$. Then from constant $\beta > 0$, $\mathbb{E}[R^\pi(T)] = \Omega(T)$. Then there always exists $\rho_t$'s satisfying $\sum_{t=1}^{T-1} \rho_t = T$, which implies $\mathbb{E}[R^\pi(T)] = \Omega(T) = \Omega(\bar{V}_T)$.

Now we think of any nontrivial algorithm which must pull an arm $a$ at least twice. Let $t'$ and $t''$ be the rounds when the policy pulls arm $a$ ($t' < t''$). If we consider $\rho_{t'} > 0$ and $\rho_t = 0$ for $t \in [T-1]/\{t'\}$ in which $\rho_{t'} = \sum_{t=1}^{T-1} \rho_t$ and $1 + \sum_{t=1}^{T-1} \rho_t \mathbb{1}(\rho_t \neq 0) \leq S_T$, then such policy has $R^\pi(T) = \Omega(\sum_{t=1}^{T-1} \rho_t)$ regret bound because, at time $t''$, it pulls the arm $a$ rotted by $\rho_{t'}$. Therefore, for any policy $\pi$, there always exist a rotting rate adversary satisfying the following expected regret bound of

$$\mathbb{E}[R^\pi(T)] = \Omega(\bar{V}_T). \tag{56}$$

Next, for the regret bound with respect to $S_T$, we follow the proof steps in Theorem 4.1. However, the regret bound of $S_T$ does not depend on the magnitude of rotting rates but on the number of rotting instances. To address this, we need to design a new worst-case in which an adversary makes near-optimal arms rotted to be sub-optimal arms *abruptly* rather than gradually. We first categorize arms as either bad or good according to their initial mean reward values. For the categorization, we utilize two thresholds in the proof as follows. Consider $0 < \gamma < c < 1$ for $\gamma$, which will be specified, and a constant $c$. Then the value of $1 - \gamma$ represents a threshold value for identifying good arms, while $1 - c$ serves as the threshold for identifying bad arms. We refer to arms $a$ satisfying $\mu_1(a) \leq 1 - c$ as 'bad' arms and arms $a$ satisfying $\mu_1(a) > 1 - \gamma$ as 'good' arms. We also consider a sequence of arms in $\mathcal{A}$ denoted by $\bar{a}_1, \bar{a}_2, \ldots$. Given a policy $\pi$, without loss of generality, we can assume that $\pi$ selects arms according to the order of $\bar{a}_1, \bar{a}_2, \ldots$.

Let $K_m$ denote the number of bad arms $a$ that satisfy $\mu_1(a) \leq 1 - c$ before sampling $m$-th good arm, which satisfies $\mu_1(a) > 1 - \gamma$, in the sequence of arms $\bar{a}_1, \bar{a}_2, \ldots$. Let $N_T$ be the number of sampled good arms $a$ such that $\mu_1(a) > 1 - \gamma$ until $T$.

We can decompose $R^\pi(T)$ into two parts as follows:

$$R^\pi(T) = R^\pi(T)\mathbb{1}(N_T < m) + R^\pi(T)\mathbb{1}(N_T \geq m). \tag{57}$$

We set $m = S_T$ and $\gamma = (S_T/T)^{1/(\beta+1)}$ with $S_T = o(T)$. For getting a lower bound for the first term in (57), $R^\pi(T)\mathbb{1}(N_T < m)$, we consider the fact that the minimal regret is obtained from the situation where there are $m - 1$ arms whose mean rewards are 1. In such a case, the optimal policy must sample the best $m - 1$ arms until their mean rewards become equal to or below the threshold value of $1 - \gamma$ (step 1) and then samples the best arm at each time for the remaining time steps (step 2). In step 1, when the optimal policy pulls an optimal arm, we can think of the case when the mean reward of the arm is abruptly rotted to the value of $1 - \gamma$. This implies that the required number of rounds for step 1 is $m - 1$. The regret from step 2 is $R = \Omega((T - m + 1)\gamma) = \Omega(S_T^{1/(\beta+1)}T^{\beta/(\beta+1)})$, in which the optimal policy pulls arms which mean rewards are below or equal to $1 - \gamma$ for the remaining time after step 1. Therefore, we have

$$R^\pi(T)\mathbb{1}(N_T < m) = \Omega(R\mathbb{1}(N_T < m)) = \Omega(S_T^{1/(\beta+1)}T^{\beta/(\beta+1)}\mathbb{1}(N_T < m)). \tag{58}$$

For getting the above, we note that there always exists $\rho_t$'s satisfying $\sum_{t=1}^{T-1} \rho_t = O(\gamma m) = o(T)$, which implies $\sum_{t=1}^{T-1} \rho_t \leq T$. Such $\rho_t$'s can be considered for the below. For getting a lower bound of the second term in (57), $R^\pi(T)\mathbb{1}(N_T \geq m)$, we use the minimum number of sampled arms $a$ that satisfy $\mu_1(a) \leq 1 - c$. When $N_T \geq m$ and $K_m \geq \kappa$, the policy samples at least $\kappa$ number of distinct arms $a$ satisfying $\mu_1(a) \leq 1 - c$ until $T$. Therefore, we have

$$R^\pi(T)\mathbb{1}(N_T \geq m) \geq c\kappa\mathbb{1}(N_T \geq m, K_m \geq \kappa). \tag{59}$$

We set $\bar{\gamma} = \mathbb{P}(\mu_1(a) > 1 - \gamma)/p(\mu_1(a) \notin (1 - c, 1 - \gamma])$. Then we have $\bar{\gamma} = \Theta(\gamma^\beta)$ from (1) with constant $\beta > 0$. By setting $\kappa = m/\bar{\gamma} - m - m/(\bar{\gamma}\sqrt{m+3})$, with $S_T = o(T)$ and constant $\beta > 0$, we have

$$\kappa = \Theta(S_T^{1/(\beta+1)}T^{\beta/(\beta+1)}). \tag{60}$$

Then from (58), (59), and (60), we have

$$\mathbb{E}[R^\pi(T)] = \Omega(S_T^{1/(\beta+1)}T^{\beta/(\beta+1)}\mathbb{P}(N_T < m) + S_T^{1/(\beta+1)}T^{\beta/(\beta+1)}\mathbb{P}(N_T \geq m, K_m \geq \kappa))$$

$$\geq \Omega(S_T^{1/(\beta+1)}T^{\beta/(\beta+1)}\mathbb{P}(K_m \geq \kappa)). \tag{61}$$

Next we provide a lower bound for $\mathbb{P}(K_m \geq \kappa)$. Observe that $K_m$ follows a negative binomial distribution with $m$ successes and the success probability $\mathbb{P}(\mu_1(a) > 1 - \gamma)/\mathbb{P}(\mu_1(a) \notin (1 - c, 1 - \gamma]) = \overline{\gamma}$, in which the success probability is the probability of sampling a good arm given that the arm is either a good or bad arm. We recall Lemma A.27 for a concentration inequality for $K_m$ in the following.

**Lemma A.28.** *For any $1/2 + \overline{\gamma}/m < \alpha < 1$,*

$$\mathbb{P}(K_m \geq \alpha m(1/\overline{\gamma}) - m) \geq 1 - \exp(-(1/3)(1 - 1/\alpha)^2(\alpha m - \overline{\gamma})).$$

From Lemma A.28 with $\alpha = 1 - 1/\sqrt{m+3}$ and large enough $T$, we have

$$\mathbb{P}(K_m \geq \kappa) \geq 1 - \exp\left(-\frac{1}{3}(m - \frac{m}{\sqrt{m+3}} - \overline{\gamma})\left(\frac{1}{\sqrt{m+3}-1}\right)^2\right)$$

$$\geq 1 - \exp\left(-\frac{1}{6}(m - \frac{m}{\sqrt{m+3}})\left(\frac{1}{\sqrt{m+3}-1}\right)^2\right)$$

$$= 1 - \exp\left(-\frac{1}{6}\frac{m}{m+3}\frac{\sqrt{m+3}}{\sqrt{m+3}-1}\right)$$

$$\geq 1 - \exp(-1/24), \tag{62}$$

where the last inequality comes from $m/(m+3) = (S_T)/(S_T+3) \geq 1/4$ and $\sqrt{m+3}/(\sqrt{m+3}-1) \geq 1$. Therefore, from (61) and (62), we have

$$\mathbb{E}[R^\pi(T)] = \Omega(S_T^{1/(\beta+1)}T^{\beta/(\beta+1)}). \tag{63}$$

Overall from (56) and (63), for any $\pi$, there exist $\rho_t$'s such that $\mathbb{E}[R^\pi(T)] = \Omega(\max\{S_T^{1/(\beta+1)}T^{\beta/(\beta+1)}, V_T\})$.

### A.10 Additional Experiments

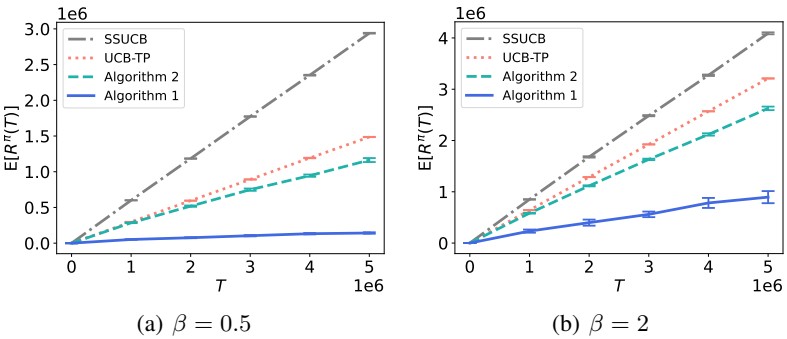

(a) $\beta = 0.5$        (b) $\beta = 2$

Figure 5: Regret Performance comparison between our algorithms and benchmarks.

We compare the performance of our Algorithms with benchmarks for smaller or larger $\beta$. In Figure 5 (a,b), we can observe that our algorithms outperform the benchmarks for $\beta = 0.5$ and $\beta = 2$.

### A.11 Lemmas for Concentration Inequalities

**Lemma A.29** (Theorem 6.2.35 in Tsun [22]). *Let $X_1, \ldots, X_n$ be identical independent Bernoulli random variables. Then, for $0 < \nu < 1$, we have*

$$\mathbb{P}\left(\sum_{i=1}^n X_i \geq (1+\nu)\mathbb{E}\left[\sum_{i=1}^n X_i\right]\right) \leq \exp\left(-\frac{\nu^2\mathbb{E}[\sum_{i=1}^n X_i]}{3}\right).$$

**Lemma A.30** (Corollary 1.7 in Rigollet and Hütter [18]). *Let $X_1, \ldots, X_n$ be independent random variables with $\sigma$-sub-Gaussian distributions. Then, for any $a = (a_1, \ldots, a_n)^\top \in \mathbb{R}^n$ and $t \geq 0$, we have*

$$\mathbb{P}\left(\sum_{i=1}^{n} a_i X_i > t\right) \leq \exp\left(-\frac{t^2}{2\sigma^2 \|a\|_2^2}\right) \text{ and } \mathbb{P}\left(\sum_{i=1}^{n} a_i X_i < -t\right) \leq \exp\left(-\frac{t^2}{2\sigma^2 \|a\|_2^2}\right).$$

