# OpenReview forum: "An Adaptive Approach for Infinitely Many-armed Bandits under Generalized Rotting Constraints"
_NeurIPS.cc/2024/Conference — NeurIPS 2024 poster_

### Official Review · Reviewer_WSKL · 2024-07-01

**Soundness:** 3
**Presentation:** 3
**Contribution:** 2
**Rating:** 6
**Confidence:** 4

**Summary:**

This paper studied an extension of multi-armed bandit problem by introducing infinitely many arms and generalized rotting constraints. It provides explicit regret lower bounds and proposes an algorithm with regret upper bound matching the lower bound when $\beta \geq 1$. It has been claimed that closing the gap between the lower and upper bounds when $\beta \in (0,1) $ remains an open problem.

**Strengths:**

- Extending the multi-armed bandit problem to infinitely many arms with rotting mean rewards is feasible in many real-world applications. Solid theoretical analyses and empirical results are presented to justify the proposed solution.

- The paper in general is well-written and easy to follow.  Since I have not checked the supplementary material step by step, I can not guarantee the correctness of the proofs, but the theoretical results make sense to me.

**Weaknesses:**

- The paper could be viewed as an extension of [1], and the impact of the paper might not be extremely significant. As mentioned in the paper, when $\beta \in (0,1)$, the proposed algorithm can not be proved near optimal at the current stage. The theoretical result could be more impactful if this issue can be solved.

[1] Jung-hun Kim, Milan Vojnovic, and Se-Young Yun. Rotting infinitely many-armed bandits. ICML, pages 11229–11254. PMLR, 2022.

- When the environment parameters $\beta, V_T,$ and $S_T$ are unknown, a significant amount of additional regret is generated with proposed Algorithm 2.

- The paper could benefit from a more extensive experiment study. For example, the algorithm performances can be compared under different rotting processes (perhaps by varying the rotting rate and adding randomness).

**Questions:**

The following questions are raised simply for discussion. There is no need to address them in the paper.

- Why the rotting budget $V_T$ and $S_T$ are defined for all selected arms? I would imagine it can fit better with real-world problems to treat the rotting process for each arm independently. Saying if an arm is selected $N$ times, the rotting budget is $f(N)$ where $f$ is a nondecreasing function.

- Corresponding to my second point in Weakness, is it possible to address the unknown parameters without incurring large additional regret? Can the methodology in the following paper [2] be applied in this paper to address the nonstationary rotting rewards? Can $\beta$ be estimated by selecting multiple new arms?

[2] Chen, Yifang, et al. "A new algorithm for non-stationary contextual bandits: Efficient, optimal and parameter-free." Conference on Learning Theory. PMLR, 2019.

**Limitations:**

The authors adequately addressed the limitations.

---

> ### Author Rebuttal · Authors · 2024-08-07
>
> We appreciate your feedback and positive evaluation. Below, we address each comment.
>
> **The paper is an extension of [1]. When $\beta\in(0,1)$, the proposed algorithm can not be proved optimal:**
>
> First of all, we highlight that we consider rotting constraints with $V_T$ or $S_T$ and initial mean rewards with $\beta>0$, both of which are not considered in [1]. As mentioned in Remark 2.2, the rotting constraint with $V_T$ ($\sum_t \rho_t \le V_T$) is more general than the maximum rotting constraint in [1]. Furthermore, the constraint with $S_T$ ($1+\sum_t 1(\rho_t \neq 0) \le S_T$) is fundamentally different from that in [1] because $S_T$ considers the total number of rotting instances rather than the magnitude of the rotting rate.
>
> While, in our work, we have demonstrated optimality only for $\beta \ge 1$, we believe that the suboptimality for $\beta \in (0,1)$ arises from the looseness of our lower bounds rather than the regret upper bounds. The gap between lower and upper bounds comes from that when $\beta(<1)$ decreases, the regret lower bounds (in Theorems 4.1, 4.2) also decrease while the upper bounds (in Theorems 3.1, 3.3) remain the same. As discussed in Appendix A.1, the phenomenon where the regret upper bound remains the same as $\beta$ decreases has also been observed in many infinitely many armed bandits [5, 24, 10]. This is because, as mentioned in [10], although there are likely many good arms when $\beta$ is small, it is not possible to avoid a certain amount of regret from estimating mean rewards. Therefore, we believe that our regret upper bounds are near-optimal across the entire range of $\beta$, and achieving tighter lower bounds for $\beta < 1$, considering such unavoidable regret, is left for future research. Notably, the optimality proven only for $\beta \ge 1$ has also been observed in stationary infinitely many armed bandits [5, 24].
>
> **When $\beta,V_T,S_T$ are unknown, additional regret is generated with Algorithm 2:**
>
> While our algorithm (Algorithm 2) incurs an additional regret to estimate the threshold parameter when parameters are unknown, if $V_T$, $S_T$ are large enough (e.g. $V_T\ge T^{1/4}$, $S_T\ge \sqrt{T}$ for $\beta=1$), then the regret bound of Algorithm 2 matches that of Algorithm 1 in Corollary 3.5 (for known parameters). This is because the additional term becomes negligible compared to the main term involving $V_T$ and $S_T$. We leave it as future work to obtain a tighter regret bound for the entire range of $V_T$ and $S_T$ when the parameters are unknown.
>
> **Extensive experiment study by varying the rotting rate with randomness:**
>
>  The main purpose of our experiments is to validate some claims of our theoretical analysis such as Remark 3.2. Based on your comments, we have conducted an additional experiment for random rotting rates in Figure 3 in the attached pdf. Following the rotting rates used in the experiment section in our paper, which rotting rates are convenient to compare the theoretical results of algorithms, we introduced random rotting rate  $\rho_t$ uniformly sampled from $[0,3/(t\log(T))]$ for each time $t$. In Figure 3, our algorithms outperform other benchmarks. We will include further experiments regarding rotting rates in our final version.
>
>  **The rotting budget  $V_T$ and $S_T$ are defined for all selected arms. How about treating the rotting process for each arm independently with the budget of each arm, $f(N)$, for selected $N$ times:**
>
> In our setting, the constraints on $V_T$ and $S_T$ quantify the rotting rate of all selected arms for all $t$, as you mentioned. We note that similar quantities related to overall nonstationarity across all arms have also been considered in standard non-stationary bandits [7, 19, 4]. These quantities help quantify the overall difficulty of the problem in our regret analysis. As you suggested, addressing individual budgets for each arm would be an interesting research question.
>
> We believe that our algorithm and analysis provide helpful insights for solving such a problem. This is because the core of our algorithm involves determining whether a selected arm is good or bad based on a threshold parameter tuned by the value of the constraint budget. If we know $f(N)$ for each arm, we can use our method to determine the status of the arm using a threshold adjusted by the budget information.
>
>  **Addressing the unknown parameters without incurring large additional regret. Can the methodology in [2] be applied in this paper? Can $\beta$ be estimated?**
>
> Here, we provide our thoughts regarding the applicability of [2]. In our problem, it is required to determine whether a selected arm is bad or not. In our algorithm, using a threshold, we can determine this with some confidence. Although we assume that nonstationarity can be detected using techniques from [2], it may not be clear how to determine the status of an arm whose initial mean reward is bad (small). In our setting, which includes infinitely many arms and many near-optimal ones, an absolute threshold (rather than a relative value between arms) may be required, as used in our algorithm to determine the status of arms. Without knowledge of parameters of  $V_T$, $S_T$, and $\beta$, how to determine such a threshold combined with the detection method is an unresolved problem.
>
> Another issue is that our setting allows for potentially large negative rewards from rotting with sub-Gaussian noise, whereas [2] considers rewards within $[0,1]$. This difference makes it challenging to apply the concentration inequality of Lemma 12 in [2] to our problem. As you suggested, applying the detection-based algorithm [2] to our setting while addressing those issues would be an interesting direction for future work.
>
> Regarding the unknown $\beta$, estimating $\beta$ directly was considered in Algorithm 3 in [10]. However, it requires additional information on a (non-zero) lower bound of $\beta$, and the performance depends on the tightness of this lower bound.

---

> > ### Comment · Reviewer_WSKL · 2024-08-08
> > **Response to rebuttal**
> >
> > Thank you to the authors for their reply. I am writing to confirm that I've read the author's rebuttal and my scores remain unchanged.

---

> > > ### Author Response · Authors · 2024-08-08
> > > **Thank you for your comment.**
> > >
> > > Thank you for maintaining the positive rating of our paper. We will incorporate the discussion, including additional experimental results, into our final version. If there are any other questions, please let us know.

---

### Official Review · Reviewer_jk3J · 2024-07-02

**Soundness:** 3
**Presentation:** 4
**Contribution:** 3
**Rating:** 6
**Confidence:** 4

**Summary:**

This paper considers the extension of the classic stochastic multi-armed bandit problem to the case where there is a) an infinite set of arms and b) rotting of the arm means. Specifically the rotting behaviour is of the 'rested bandit' variety, where the mean reward of an arm may fall as an immediate consequence of playing it, but the mean of an unplayed arm will not change, in contrast to the restless bandit. The infinite set of arms have i.i.d. mean rewards (no structural assumptions on the set of arms or existence of a reward function over them etc.) and reward observations are sub-Gaussian.

This paper considers a more general rotting behaviour than Kim et al. (2022, ICML). In Kim et al. (2022) the magnitude of rotting in each round was bounded, in the present paper the more general setting where the total amount of rotting in T rounds is bounded is considered, and the somewhat different setting where the number of rounds in which rotting occurs is considered. These two settings are referred to as the 'slow' and 'abrupt' rotting cases respectively.

An important parameter in the infinitely-many-armed bandit problem is $\beta>0$ which controls the probability of an arm being $\delta$-near optimal. All regret bounds in the paper for mean rewards in [0,1] and consider dependence on the number of rounds $T$ and parameter $\beta$. $\beta=1$ corresponds to a uniform distribution on arm means.

The paper derives a sliding-window-UCB-based approach, which can be tuned to both the slow and abrupt rotting cases when the bounds on rotting behaviour are known, and an adaptive version which aims to learn the rotting behaviour when these parameters are not known. The paper shows tight (in terms of order) regret guarantees on these algorithms, and improved empirical performance over sensible competitors.

**Strengths:**

The paper studies an interesting extension of prior work to allow for more general rotting constraints. This is likely to be of interest to the multi-armed bandit community and both the methodological and theoretical work present some novelty and careful algorithmic design that merit publication.

I was pleased to see a treatment of the case where key problem parameters were not known, and that largely comparable bounds were achievable in this setting.

Generally, the paper is written well and concisely, with useful clarifications made around the most important details and a clear relationship to the most closely related prior work.

**Weaknesses:**

There are three main weaknesses which I would like to see commented on the rebuttal phase. I think the first is difficult to address fully in a rebuttal phase, but I would like to be reassured that potential issues do not limit the scope of the contribution.

1. All of the theoretical results are ultimately order results only, without identification of the constants in the appendices, and some being somewhat vaguely described as needing to be 'large enough'. The paper would be stronger if these constants (or upper bounds on them) could be identified, to guarantee for which values of $T$ the bounds are meaningful. It would be helpful if the bounds could thus be realised for the cases considered in the experiments and compared to the actual regret of Algorithms 1 and 2.

2. If I am not mistaken the experiments only consider $\beta=1$? It would be interesting to see if the convergence of the algorithm is replicated for various $\beta \neq 1$.

3. The motivation in terms of real applications is not especially strong. It is difficult to imagine a setting where the precise assumptions of the paper are realistic. In a recommendation setting, is it likely that there will be sufficient information to be confident that the click through rate of an item is always non-increasing, yet there is no contextual information available to supplement the model or exploit similar click through rates across similar items? In the clinical setting, is it likely that the rested assumption is realistic (i.e. any loss in efficacy is only determined by the actions of a single decision maker) alongside the immediacy of feedback and time horizons being of a scale such that the algorithms actually exhibit non-linear regret? If it is the case that the benefit of the paper is more fundamental - it presents an understanding of a simpler problem and its insights serve as a foundation to tackling these more complex real-world challenges, then perhaps it would be more helpful to make that clear?

**Questions:**

1. Can you provide clarity on whether your constants in the regret analysis are identifiable, and suggest the values of these for some simple instance of the problem? How would the resultant bounds compare to the empirical performance of Algorithms 1 and 2?

2. What would the experimental results look like for $\beta \neq 1$?

3. Is Prop 2.3 missing some kind of additional condition, or clarity over what is meant by worst-case in line 453? It would seem to be that you could construct an example where $\rho_1=T+1$ and all other $\rho=0$ and one could achieve $\sqrt{T}$ regret while satisfying $\sum{t=1}^{T-1}\rho_t >T$?

4. Can you put the WUCB definition in an equation display and reference its equation number in the statement of Algorithm 1?

**Limitations:**

Yes

---

> ### Author Rebuttal · Authors · 2024-08-07
>
> We appreciate your feedback and positive evaluation of our work. Below, we address each comment.
>
> **All of the theoretical results are ultimately order results only, without identification of the constants in the appendices. It would be helpful if the bounds could thus be realised for the cases considered in the experiments and compared to the actual regret of Algorithms 1 and 2:**
>
> In our work, we provide regret bounds in terms of the horizon time $T$ and the rotting constraint parameters $V_T$ and $S_T$ (to the power of $\beta$ terms),  which hold up to constant factors whose values are not asserted in the statements of our theorems.
>
> Based on your feedback, we will include more details regarding the values of these constant factors in the final version of our paper. For instance, in Lemma A.5, instead of using $m^{\mathcal{G}} = C_3$ for some sufficiently large constant $C_3 > 0$, we can specify $m^\mathcal{G}=3$ with $V_T\le  \max\\{1/T^{1/(\beta+1)},1/\sqrt{T}\\}$ and $\delta=\max\\{1/T^{1/(\beta+1)},1/\sqrt{T}\\}$. Then, we have $m^{\mathcal{G}}\min\\{\delta/2,V_T\\}=3\min\\{\delta/2,V_T\\}>V_T$, which can conclude the lemma.
>
> We will also provide additional experimental results related to this. For now, we present an experiment in Figure 1 in the attached pdf that compares the performance of our algorithms with theoretical regret upper bounds with a constant of 1. We observe that the performance of our algorithms (blue and green solid lines) is better than the theoretical regret upper bounds (light blue and light green dashed lines), respectively, because the theoretical bounds represent worst-case regret regarding rotting rates, while the experiment is conducted under a specific instance of rotting rates.
>
> It is noteworthy that providing regret bounds that hold up to constant factors is an important first step to establishing fundamental bounds for the underlying learning problem. Note that previous work on stationary infinitely-armed bandits (e.g. [24, 10]) also focused on providing regret bounds up to constant factors, whose values are not identified explicitly.
>
> **What would the experimental results look like for $\beta\neq 1$:**
>
> We appreciate your suggestion. We have included additional experiments in which we varied the value of $\beta$ in Figure 2 of the attached pdf. In Figure 2, our algorithms outperform other benchmarks for various $\beta$. We will incorporate this into our final version.
>
>
> **The motivation in terms of real applications is not especially strong:**
>
> We appreciate your comments and suggestions. We highlight that (rested) rotting rewards in bandits have been studied [16, 20, 21, 13], motivated by real-world applications such as recommendations and clinical trials. For instance, in recommender systems, the click rate for each item may diminish due to user boredom with repeated exposure to the same content. Similarly, in clinical trials, the efficacy of a medication can decline due to drug tolerance induced by repeated administration. However, the previous work regarding rotting bandits, except for [13], focused on MAB with finite arms, which have limitations when the number of items is large, as in recommender systems. In contrast, we consider the case with infinitely many arms without contextual information.
>
> We believe our algorithm, as you mentioned, provides insights into handling rotting cases in real-world scenarios. Specifically, our adaptive SW-UCB and threshold approach offers valuable insights for designing bandit algorithms regarding when to explore new arms and how to estimate rotting mean rewards and determine the status (good or bad) of each arm when rotting occurs. Furthermore, our study on optimizing the threshold parameter provides practical guidance for setting threshold values in real-world applications.
>
> **Is Prop 2.3 missing some kind of additional condition, or clarity over what is meant by worst-case in line 453?:**
>
> Our regret analysis focuses on the *worst-case* regret concerning rotting rates. As outlined in Assumption 2.1, we consider an adaptive adversary that determines the rotting rate $\rho_t$ arbitrarily immediately after the agent pulls an action $a_t$ at each time step $t$. In Proposition 2.3, we show that there always exists a rotting adversary (or instances of rotting rates over $T$) with $\sum_t\rho_t>T$ such that it incurs at least $T$ regret for any algorithm, implying an $\Omega(T)$ regret lower bound. This does *not* imply that any rotting adversary (or any rotting instances) occur $T$ regret. This worst-case lower bound can be demonstrated by providing *a* rotting adversary (or rotting rate instances) where any algorithm will incur at least $T$ regret. Additionally, we show that this lower bound can be achieved by a simple algorithm that pulls a new arm every round. Therefore, Proposition 2.3, which is regarding the worst case, does not contradict the example you mentioned. We appreciate your helpful comments to improve clarification. We will include the explanation to clarify Proposition 2.3 in our final version.
>
>  **Can you put the WUCB definition in an equation display and reference its equation number in the statement of Algorithm 1?:**
>
> We appreciate your helpful comments for improving the clarity of our paper. We will display the WUCB definition in an equation and reference its number in Algorithm 1.

---

> > ### Comment · Reviewer_jk3J · 2024-08-10
> > **Response to Rebuttal**
> >
> > Thank you for your detailed reply to my comments. I appreciate the commitment to make modifications and am in agreement with your response. I have also read the other reviews and your responses and consider these to be suitably well addressed. As such I will retain my score and increase my confidence. (Justification: I agree with the remarks that finding results without constants and algorithms that can lead to further application-specific work are important fundamentals, but I feel for a score of 7+ the paper would have made more progress in one of these areas.)

---

> > > ### Author Response · Authors · 2024-08-10
> > > **Thank you for your comment.**
> > >
> > > Thank you for maintaining a positive rating on our paper and increasing your confidence score. We appreciate your detailed feedback. We will incorporate the discussion into our final version, including more details regarding constant factors, a detailed explanation of Prop 2.3, and additional experimental results.

---

### Official Review · Reviewer_ALsC · 2024-07-11

**Soundness:** 3
**Presentation:** 3
**Contribution:** 3
**Rating:** 6
**Confidence:** 2

**Summary:**

Authors investigated an adaptive approach for the rotting bandits problem under the infinitely arms assumption.

**Strengths:**

Authors introduced a new ucb like policy for the mentioned problem, additionally a lower bound analysis has been carried on.

**Weaknesses:**

No clear weaknesses

**Questions:**

I wonder how the regret bound behaves with respect to the effective rotting instead of the V_T or S_T. Assuming their values to be large and far from the real rotting, would it be possible to extend this analysis and solution to propose an adaptive result? Additionally, I wonder if also the regret lower bound would be tight in that case.

Similarly, I was wondering how good these results would be for the infinitely many afrmed bandit with no rotting.

**Limitations:**

No clear limitations have been found.

---

> ### Author Rebuttal · Authors · 2024-08-06
>
> We appreciate your feedback and positive evaluation of our work. Below, we address each comment.
>
> **I wonder how the regret bound behaves with respect to the effective rotting instead of the $V_T$ or $S_T$:**
>
> In this problem, as mentioned in Assumption 2.1, we consider an adaptive adversary who determines $\rho_t$ arbitrarily, subject to the constraint $\sum_t \rho_t \le V_T$ or $1 + \sum_t 1(\rho_t \neq 0) \le S_T$ for given $V_T$ or $S_T$.  As a side remark, these constraints are more general than the stricter conditions $\sum_t \rho_t = V_T$ or $1 + \sum_t 1(\rho_t \neq 0) = S_T$. Our regret analysis, for both upper and lower bounds, addresses the *worst-case* scenario concerning the rotting rates under the general constraint of $V_T$ or $S_T$. Hence, our regret bounds are expressed in terms of $V_T$ or $S_T$.  We also note that standard nonstationary bandits have been studied under a similar concept of a nonstationary budget upper bounded by $V_T$ [7,19].
>
>
> We can consider a scenario where an adversary determines $\rho_t$ under the stricter constraints of $\sum_t \rho_t = V_T$ or $1 + \sum_t 1(\rho_t \neq 0) = S_T$. Our (upper and lower) regret bounds apply with these values of $V_T$ and $S_T$, respectively corresponding to the effective rotting amounts of $\sum_t\rho_t$ and $1 + \sum_t 1(\rho_t \neq 0)$. When there is no rotting such that $V_T=0$ and $S_T=1$, then our bounds are the same as the bounds for the stationary case (e.g. $\sqrt{T}$ for $\beta=1$ as in [24, 5]). Furthermore, if $V_T$ and $S_T$ are unknown, Algorithm 2 (in Appendix A.6) can achieve regret bounds that depend on the values of $V_T$ and $S_T$ (Theorem A.15).

---

> > ### Comment · Reviewer_ALsC · 2024-08-12
> >
> > Thank you to the authors for their reply. I am writing to confirm that I've read the author's rebuttal and my scores remain unchanged.

---

> > > ### Author Response · Authors · 2024-08-12
> > > **Thank you for your comment**
> > >
> > > Thank you for maintaining a positive rating on our paper. We will incorporate the discussion into our final version.

---

### Official Review · Reviewer_g8GP · 2024-07-12

**Soundness:** 3
**Presentation:** 3
**Contribution:** 2
**Rating:** 4
**Confidence:** 4

**Summary:**

The paper studies infinite-armed bandits with rotting rewards. They show a lower bound on regret in terms of the total-variation and number of abrupt changepoints in the change in rewards. They also provide regret upper bounds: (1) a UCB-like algorithm tuned with knowledge of problem parameters gets optimal regret in some regimes and (2) a parameter-free bandit-over-bandit algorithm can attain optimal regret in some regimes when the level of non-stationarity is large.

**Strengths:**

* The infinite-armed bandit problem is well motivated and the problem and results are clearly presented.
* There are matching upper and lower bounds on regret, at least for the better-understood $\beta \geq 1$ setting.

**Weaknesses:**

* The definition of $V_T,S_T$ are a bit unclear and may not be fully rigorous. They are bounds on the total amount and count of rotting, but the rotting $\rho_t$ depends on the chosen arm $\rho_t$ and is thus random (in fact, determined by an adaptive adversary according to Assumption 2.1). But, meanwhile, $V_T,S_T$ seem to be treated as deterministic constants in the whole paper given that they are, for example, used to tune Algorithm 1 to get the best regret bounds. In fact, it would not even make sense for $V_T,S_T$ to _appear_ in the expected regret bounds if they were random. So, the only way the results of this work can rigorously hold is if $V_T,S_T$ bound the respective random quantities $\sum_{t=1}^{T_1} \rho_t$ and $1 + \sum_{t=1}^{T-1} 1\{\rho_t\neq 0\}$ for all realizations of the randomness, which then means there are some missing assumptions for the results of this paper. For instance, even for a "nice" algorithm, $\sum_{t=1}^{T_1} \rho_t$ could be very large with some probability in which case $V_T$ is forced to be large as well because of a "bad realization". The only other way around this is to make further assumptions about the adversary/design of non-stationarity. In either case, there should be more discussion as the rigor of results presented versus the given assumptions seems unclear to me.
*  There are no optimal regret upper bounds without parameter knowledge. Also, the bandit-over-bandit result for parameter-free result requires further assumptions on the non-stationarity and so does not apply as generally as the other results of this paper.

**Questions:**

Please refer to strengths and weaknesses above.

**Limitations:**

No broader impact concerns.

---

> ### Author Rebuttal · Authors · 2024-08-06
>
> We appreciate your time to review our paper and comments. Below, we address each comment.
>
> **The definition of $V_T$, $S_T$
>  are a bit unclear and may not be fully rigorous. They are bounds on the total amount and count of rotting, but the rotting
>  depends on the chosen arm
>  and is thus random**:
>
>
>  We appreciate your detailed comments. However, we argue that the definitions of $V_T$ and $S_T$ are rigorous in our context, as they serve as *constraints* on the rotting adversary. These quantities are not random outcomes based on the adversary's behavior but rather imposed conditions. To clarify this, we restate our Assumption 2.1 concerning the adaptive adversary as it is.
>
> ---
>
>  **Assumption 2.1.** At each time $t\in[T]$, the value of rotting rate $\rho_t>0$ is arbitrarily determined immediately after the agent pulls $a_t$, subject to the constraint of either slow rotting for a given $V_T$ or abrupt rotting for a given $S_T$.
>
> ---
>
> According to this assumption, we consider $V_T$ and $S_T$ to be given a priori to the adversary, and the adversary determines $\rho_t$ subject to these constraints. $V_T$ and $S_T$ are deterministic quantities that act as *constraints*, which are determined before the game begins. This is why we refer to the two scenarios as the slow rotting constraint ($V_T$) and the abrupt rotting constraint ($S_T$). In our setting, if $V_T$ or $S_T$ is large, then, as we have shown in our regret lower bounds, any algorithm naturally cannot avoid a large regret bound in the worst case.
>
>
> **There are no optimal regret upper bounds without parameter knowledge. Also, the bandit-over-bandit result for parameter-free result requires further assumptions on the non-stationarity and so does not apply as generally as the other results of this paper:**
>
>
>
>
> In the case when the parameters are unknown, there is an additional regret for Algorithm 2, which stems from learning the threshold parameter. However, if $V_T$ and $S_T$ are large enough (e.g. $V_T\ge T^{1/4}$, $S_T\ge \sqrt{T}$ for $\beta=1$), then the regret bound in Theorem A.15 for Algorithm 2 matches that in Corollary 3.5 for Algorithm 1 (for known parameters). It is an open problem to achieve tight regret bounds for entire ranges of $V_T$ and $S_T$.
>
>
>  Now we discuss the further assumptions.
> For the unknown parameter case, we consider  Assumptions A.10 and A.12 rather than Assumptions 2.1 and 2.4 (for the known parameter case). In Assumption A.10, we still consider an *adaptive adversary* for the rotting rates, such that $0 \le \rho_t \le \varrho_t$ for given $\varrho_t$ to the adversary, where $\sum_t \varrho_t \le V_T$ and $1 + \sum_t 1(\varrho_t \neq 0) \le S_T$. Here $\varrho_t$'s are assumed to be determined before the algorithm is run.
> As we describe in Remark A.11, this assumption is more general than that in [13], where $\varrho_t = \rho$ with a maximum rotting rate $\rho$ for all $t$. In the special case where $\rho_t = \varrho_t$, the assumption represents an oblivious adversary.
>
>
> In Assumption A.12, we consider $\sum\_{t \in \mathcal{T}\_i} \rho\_t \le H$ for all $i \in [ \lceil T/H \rceil]$, where $\mathcal{T}\_i=[(i-1)H+1,iH]$ representing the $i$-th block of times of length $H$ within horizon time $T$. As we mention in Remark A.13, this assumption is satisfied when mean rewards are constrained by $0 \le \mu_t(a_t) \le 1$ for all $t$ because $0\le\rho_t\le1$. The positive mean reward constraint is frequently encountered in real-world applications such as click rates in recommendation systems. We also note that, as mentioned in Remark A.14, the assumption is still more general than the one with a maximum rotting rate constraint of $\rho_t \le \rho = o(1)$ in [13]. This is because, in our setting, each $\rho_t$ is not necessarily bounded by $o(1)$ but $\sum_{t \in \mathcal{T}_i} \rho_t \le H$.
>
>
>  Lastly, we emphasize that our study is the first work to consider $V_T$, $S_T$, and $\beta$ in the context of rotting bandits with infinite arms, which is fundamentally different from finite-armed bandits due to the necessity of exploring new arms (details are described in lines 58–70). Notably, when the parameters are known, we achieve tight results using a novel approach and provide regret lower bounds.

---

> ### Author Response · Authors · 2024-08-14
> **Official Comment by Authors**
>
> Dear Reviewer g8GP,
>
> Thank you again for taking the time to review our paper.
>
> We sincerely hope our responses have adequately addressed your questions and comments. If they have, we appreciate it if you could reconsider your evaluation. If you have any last-minute questions, please let us know.
>
> Sincerely,
>
> Authors

---

> ### Comment · Reviewer_g8GP · 2024-08-14
>
> * **On $V_T/S_T$**: thank you for the clarification. It seems one assumes the adversary is not fully adaptive then and there are limitations on how much rotting it can force on the rewards. I would say wording this as a "adaptive adversary" can be a bit misleading then as it is really a constrained adversary.
>
> * **optimal regret upper bounds without parameter knowledge**: I would carefully explain in the presentation why the further Assumptions A.10 is needed. It seems like it is an artifact of the bandit-over-bandit approach as you cannot have the adversary respond in an overly strong way to the master's choice of base over $H$ rounds.
>
> Even under the broader above mentioned constrained adversary there seem to be no parameter-free results without this A.10. As such, I feel the scope of result remains limited in this work even while being the first to study this particular setting/parametrization.

---

> ### Author Response · Authors · 2024-08-14
> **Official Comment by Authors**
>
> Thank you for your comment. We would like to address a few points regarding your comments:
>
> **On $V_T$, $S_T$:** Based on your comment, we believe your initial concern regarding $V_T$ and $S_T$ has been resolved. However, we would like to provide additional explanation for clarity. We strongly believe that the adaptive adversary under the slow ($V_T$) or abrupt rotting constraint ($S_T$) is a natural and general assumption in our adversary rotting scenario. The adaptive adversary determines an arbitrary rotting rate at each time, immediately after the agent's action is determined, under either $V_T$ or $S_T$. From the values of $S_T$ and $V_T$, we can appropriately quantify the difficulty of our problems, as demonstrated by our regret lower bounds.
>
> Without such constraints (i.e., fully adaptive according to your comment), the adaptive adversary could easily make the problem trivial in the 'worst-case' scenario of our setting because whenever an algorithm finds a good arm, the adversary could cause that arm to rot and become a bad arm or even have a negative mean reward adaptively, without the constraints. We also note that similar quantities of $V_T$ and $S_T$ have also been considered in standard nonstationary bandit literature [7,19,4], where they are treated as determined quantities, not random variables, as in our setting. If necessary, we are more than happy to clarify this further in our final version to prevent any potential confusion.
>
> **Assumption A.10 for parameter-free:** The additional constraint in Assumption A.10 is required due to our adaptive adversary, which selects the rotting rate arbitrarily and adaptively in response to the selected action at each time, within the bandit-over-bandit framework. If we consider an oblivious adversary instead of an adaptive one, where the values of rotting rates $\rho_t$ are predetermined such that $\sum_t \rho_t\le V_T$ and $1+\sum_t 1(\rho_t\neq 0)\le S_T$ before the game begins, then this satisfies Assumption A.10 with $\rho_t=\varrho_t$. In other words, as we mentioned in Remark A.11, Assumption A.10 is a more general assumption than this oblivious adversary and that in [13].
>
>  As mentioned in lines 831~834, the well-known black-box framework proposed for addressing nonstationarity [25] is not applicable to this problem, and attaining the optimal regret bound under a parameter-free algorithm for all ranges of $V_T$ and $S_T$ remains an open problem. However, we respectfully disagree with your comment that the results remain limited. We again highlight that our study is the first work to examine  $V_T$, $S_T$, and $\beta$ in the context of rotting bandits with infinite arms, which is fundamentally different from finite-armed bandits (details are described in lines 58–70). Notably, we achieve tight results through a novel approach when the parameters are known, and we establish regret lower bounds. We also examine the case of unknown parameters. We believe our work is crucial for the community.

---

### Author Rebuttal · Authors · 2024-08-06

We appreciate you taking the time to review our paper. We are encouraged by the feedback indicating that our problem is well motivated by many real-world applications, solid theoretical analyses and empirical results are presented, and the paper is written well and concisely.  We have attached a pdf with additional experimental results to address the reviewers' comments. In the following, we provide detailed responses to each comment from the reviewers.

---

### Decision · Program_Chairs · 2024-09-25

**Decision:**

Accept (poster)

**Comment:**

This paper considers the many-armed bandit problem where the arms rewards may drop as one draws them. The paper considers two regimes (slow and abrupt rottings). The problem setting of this paper generalizes Kim et al. [13] for general beta (top-arm ratio) and variation budget. They propose sliding-UCB-based algorithms and analyze them for the two regimes. They also provided a lower bound that is tight for $\beta > 1$. The main strength of this paper is its adaptivity against a general setting with a single algorithm. Regarding the motivation based on real-world, sentiment is mixed (generally positive). The weaknesses of this paper, suggested by the reviewers, are mainly due to the requirement for the variation size ($S_T$ or $V_T$), which is partially addressed in Algorithm 2 with additional assumptions). The score of this paper is on the margin. The overall sentiment is positive about the paper. In the discussion phase, some of the reviewers suggest consideration of existing work (such as the replay phase method by Chen et al. [2019] or Auer et al. [2018]). Even though the complexity of the method on Chen et al. is relatively high.